# Modeling the evolution of pulse-like perturbations in atmospheric carbon and carbon isotopes: the role of weathering-sedimentation imbalances

Aurich Jeltsch-Thömmes and Fortunat Joos

Climate and Environmental Physics, Physics Institute and Oeschger Centre for Climate Change Research, University of Bern, Bern, Switzerland

**Correspondence:** Aurich Jeltsch-Thömmes (aurich.jeltsch-thoemmes@climate.unibe.ch)

**Abstract.** Measurements of carbon isotope variations in climate archives and isotope-enabled climate modeling foster the understanding of the carbon cycle. Perturbations in atmospheric $CO_2$ and in its isotopic ratios ($\delta^{13}C$, $\Delta^{14}C$) are removed by different processes acting on different timescales. We investigate these differences on timescales of up to 100,000 years in pulse release experiments with the Bern3D-LPX Earth system model of intermediate complexity and by analytical solutions from a box model. On timescales from years to many centuries, the atmospheric perturbations in $CO_2$ and $\delta^{13}CO_2$ are reduced by air-sea gas exchange and physical transport from the surface to the deep ocean and by the land biosphere. Isotopic perturbations are initially removed much faster from the atmosphere than perturbations in $CO_2$ as explained by aquatic carbonate chemistry. On multi-millennial timescales, the $CO_2$ perturbation is removed by carbonate compensation and silicate rock weathering. In contrast, the $\delta^{13}C$ perturbation is removed by the relentless flux of organic and calcium carbonate particles buried in sediments. The associated removal rate is significantly modified by spatial $\delta^{13}C$ gradients within the ocean influencing the isotopic perturbation of the burial flux. Space-time variations in ocean $\delta^{13}C$ perturbations are captured by Principal Components and Empirical Orthogonal Functions. Analytical impulse response functions for atmospheric $CO_2$ and $\delta^{13}CO_2$ are provided. Results suggest that changes in terrestrial carbon storage were not the sole cause for the abrupt, centennial $CO_2$ and $\delta^{13}CO_2$ variations recorded in ice during Heinrich Stadials HS1 and HS4, though model and data uncertainties prevent a firm conclusion. The $\delta^{13}C$ offset between the Penultimate and Last Glacial Maximum reconstructed for the ocean and atmosphere is most likely caused by imbalances between weathering, volcanism and burial fluxes. Our study highlights the importance of isotopic fluxes connected to weathering-sedimentation imbalances, which so far have been often neglected on glacial-interglacial timescales.

## 1 Introduction

Research efforts to understand the past and future evolution of atmospheric $CO_2$ and of its isotopic signature are ongoing since many decades. Yet many open questions remain, including for example the offsets in the stable isotope signature of $CO_2$ ($\delta^{13}C_{atm}$) and of dissolved inorganic carbon ($\delta^{13}C_{DIC}$) between the Penultimate (PGM) and Last Glacial Maximum (LGM) and between the two recent interglacials, the Eem and and the Holocene (Schneider et al., 2013; Eggleston et al.,

2016; Hoogakker et al., 2006; Oliver et al., 2010). Could this PGM-LGM $\delta^{13}C$ offset result from a change in terrestrial carbon storage, or rather from changes in carbon and carbon isotope fluxes connected to the sedimentary weathering-burial cycle? Further questions remain regarding $CO_2$ and $\delta^{13}C_{atm}$ variations on centennial and millennial timescales and a possible role of changes in the land biosphere carbon inventory (e.g. Köhler et al., 2010; Skinner et al., 2010; Schmitt et al., 2012; Skinner et al., 2017; Bauska et al., 2018). Reconstructed variations in $CO_2$ and $\delta^{13}C_{atm}$ during Heinrich Stadial (HS) 4 and 1, two northern hemisphere cold periods during the last glacial, have been attributed to changes in terrestrial carbon storage (e.g. Bauska et al., 2016, 2018), however other scenarios might be possible.

The perturbations in $CO_2$ and $\delta^{13}CO_2$ due to a carbon input (or removal) from the land biosphere or from fossil fuel burning are eventually removed by a few principal processes. In the case of $CO_2$, over the first decades and centuries, it is ocean- and land uptake; on millennial timescales, $CaCO_3$ compensation; and finally, on timescales of hundreds of thousands of years the remaining atmospheric $CO_2$ perturbation is removed by enhanced silicate rock weathering (e.g. Archer et al., 1998; Joos et al., 2001, 2013; Colbourn et al., 2015; Lord et al., 2016). In the case of $\delta^{13}C$, the perturbation is mixed rather quickly throughout the atmosphere-land-ocean reservoir and is removed on multi-millennial timescales by changes in the isotopic signature of burial fluxes to the lithosphere.

Simulations of climate and $CO_2$ over thousands of years with dynamic, 3-dimensional models remain challenging. Over the last decades, computational power has continuously increased and first results from simulations with Earth System Models of Intermediate Complexity (EMICs) covering glacial-interglacial cycles have been published (e.g. Brovkin et al., 2012; Menviel et al., 2012; Ganopolski and Brovkin, 2017). However, to date these long timescales prevented large ensemble simulations with spatially resolved models as well as glacial-interglacial simulations using state-of-the-art Earth System Models. A possible tool to overcome these computational obstacles is the use of reduced form models invoking Impulse Response Functions (IRFs) and Principal Component - Empirical Orthogonal Function (PC-EOF) analysis. IRFs and PC-EOF analysis are instrumental to characterize the fundamental time and spatial scales of model response to a perturbation.

The response of a linear system to a perturbation such as carbon emissions to the atmosphere can be fully characterized by its IRF. The atmosphere-ocean $CO_2$ cycle can be approximated as a nearly linear system, as long as concentrations do not vary much, i.e. less than a doubling of pre-industrial $CO_2$ concentrations (Hooss et al., 2001). The perturbation in the atmosphere or ocean is then given by the convolution integral of the emission history and the corresponding IRF (e.g. Joos and Bruno, 1996; Joos et al., 1996). IRFs can easily be described by a sum of exponentials, yielding a cost efficient substitute model for describing the temporal evolution of an atmospheric $CO_2$ perturbation. IRFs have been widely used to calculate remaining atmospheric fractions of anthropogenic $CO_2$ emissions on different timescales and to characterize model behaviour and their characteristic response time scales (e.g. Siegenthaler and Oeschger, 1978; Maier-Reimer and Hasselmann, 1987; Sarmiento et al., 1992; Siegenthaler and Joos, 1992; Enting et al., 1994; Joos and Bruno, 1996; Archer et al., 1997, 1998; Ridgwell and Hargreaves, 2007; Archer and Brovkin, 2008; Archer et al., 2009; Joos et al., 2013; Colbourn et al., 2015; Lord et al., 2016). However, peer-reviewed studies that quantify the IRF for carbon isotopes of $CO_2$ are scarce (Schimel et al., 1996; Joos and Bruno, 1996) and, to our knowledge, not available for multi-millennial time scales.

Principal Component analysis is widely used to capture the information of multi-dimensional data. The atmospheric perturbation in $CO_2$ and its isotopes in response to carbon input external to the ocean-atmosphere system is propagated to the ocean. The resulting spatio-temporal (4-D) evolution of marine tracers may be conveniently approximated using PC-EOF analysis. The perturbation is described as a superposition of spatial empirical orthogonal functions ($EOF(x)$) multiplied with the corresponding time dependent principal component ($PC(t)$). Combining IRF and PC-EOF then allows to compute the perturbation in space and time with minimal computational resources (Hooss et al., 2001; Joos et al., 2001)

The main goal of this study is to investigate the time scales and underlying processes for the removal of an atmospheric perturbation in $CO_2$ and its $\delta^{13}C$ signature in response to an external carbon input. To this end, we performed pulse-release (uptake) experiments over 100,000 years with the Bern3D-LPX EMIC. We determine characteristic Impulse Response (or Green's) Functions (IRF) and describe marine spatio-temporal variations using PC-EOF analysis. Responses in atmospheric radiocarbon signature ($\Delta^{14}C$) are briefly addressed. Implications of our results for the PGM-LGM $\delta^{13}C$ difference and for past centennial $CO_2$ variations are discussed. Simple expressions that illuminate the fundamental differences in the removal time scales for an atmospheric perturbation in $CO_2$ versus the associated perturbation in $\delta^{13}CO_2$ are developed in the appendix. The pulse experiments represent (i) the sudden carbon uptake from or release to the atmosphere by the land biosphere, (ii) the release of old organic carbon, e.g., from ancient peat or permafrost material, (iii) the burning of all conventional fossil fuel resources, and (iv) a variation in $^{14}C$ production by cosmic radiation or atomic bomb tests.

## 2 Model and experimental set-up

### 2.1 The Bern3D-LPX model

For this study, the Bern3D v2.0s Earth System Model of Intermediate Complexity (EMIC) is used coupled to the dynamic vegetation model LPX-Bern v1.4. The Bern3D consists of a three dimensional geostrophic ocean (Edwards et al., 1998; Müller et al., 2006) with an isopycnal diffusion scheme and Gent-McWilliams parameterization for eddy-induced transport (Griffies, 1998), and a single layer energy-moisture balance atmosphere to which a thermodynamic sea-ice component is coupled (Ritz et al., 2011). We use the ocean model coupled to a 10 layer sediment module (Tschumi et al., 2011; Heinze et al., 1999) with updated sediment parameters (Jeltsch-Thömmes et al., 2019) and to a global representation of rock weathering (Colbourn et al., 2013). The resolution of the Bern3D model components is 41x40 horizontal grid cells with 32 logarithmically spaced depth layers in the ocean. Wind stress at the ocean surface is prescribed from the NCEP/NCAR monthly wind stress climatology (Kalnay et al., 1996) and air-sea gas exchange and carbonate chemistry follow OCMIP-2 protocols (Najjar and Orr, 1999; Orr et al., 1999; Wanninkhof, 2014; Orr and Epitalon, 2015; Orr et al., 2017). The global mean air-sea gas transfer rate is reduced by 19 % to match observational radiocarbon estimates (Müller et al., 2006) and gas transfer velocity scales linearly with wind speed (Krakauer et al., 2006).

Marine productivity is calculated as a function of light availability, temperature, and nutrient concentrations (P, Fe, Si), and is restricted to the euphotic zone in the uppermost 75 m. Tracers, including dissolved inorganic carbon and semi-labile organic carbon (DIC, DOC), the corresponding isotopic forms, as well as alkalinity (Alk), phosphate ($PO_4$), oxygen ($O_2$), iron

(Fe), silica (Si), and an ideal age tracer, are transported by advection, diffusion, and convection. Biogeochemical cycling is described in detail in Tschumi et al. (2011) and Parekh et al. (2008) and subsequent studies (e.g. Menviel and Joos, 2012; Menviel et al., 2012; Roth and Joos, 2012; Roth et al., 2014; Menviel et al., 2015; Battaglia et al., 2016; Battaglia and Joos, 2018a, b; Jeltsch-Thömmes et al., 2019). Isotopic fractionation between model components is documented in Jeltsch-Thömmes
et al. (2019).

The sediment module dynamically calculates the transport, redissolution/remineralization, and bioturbation of solid material, the pore water chemistry, and diffusion in the top 10 cm of the sediment. The governing equations of the sediment model are given in the appendix A of Tschumi et al. (2011). A comparison of simulated versus observation-based sediment composition for the setup used in this study is given in appendix B of Jeltsch-Thömmes et al. (2019). A table with all parameters applied
here and by Jeltsch-Thömmes et al. (2019) is given in the appendix (Table B1). Four solid tracers ($CaCO_3$, opal, POC, clay) and seven tracers in the porewater (DIC, $DI^{13}C$, $DI^{14}C$, alkalinity, phosphate, oxygen, and silicic acid) are modeled. The dissolution of $CaCO_3$ and POC depends on the respective weight fraction of $CaCO_3$ and POC in the solid phase of the sediment and the pore-water $CO_3^{2-}$ and $O_2$ concentration, respectively. Denitrification is not considered in this model version, but $O_2$ is not consumed below a threshold, somewhat reflecting the process of denitrification without modeling $NO_3^-$. The respective reaction
rate parameters for $CaCO_3$ dissolution and POC oxidation are global constants (see Roth et al., 2014; Jeltsch-Thömmes et al., 2019). The model assumes conservation of volume, i.e. the entire column of the sediments is pushed downward if deposition exceeds redissolution into pore waters. Any solid material that is pushed out of the diagenetic zone (top 10 cm) disappears into the subjacent diagenetically consolidated zone (burial or loss flux) (see Tschumi et al., 2011, for more details). The burial flux at preindustrial steady state is 0.22 $GtC\,yr^{-1}$ in the form of $CaCO_3$, 0.24 $GtC\,yr^{-1}$ in the form of POC, and 6.96 $TmolSi\,yr^{-1}$
in the form of opal, all within the observational range. The global inventories in the interactive sediment layers amount to 916 GtC for $CaCO_3$, to 510 GtC for POC, and to 21,460 Tmol for opal. Carbonate chemistry within sediment pore waters is calculated as in the ocean by using the MOCSY routine of Orr and Epitalon (2015).

Input fluxes by "weathering" of P, ALK, DIC, $DI^{13}C$, and Si are added uniformly to the coastal surface ocean. At the beginning of transient simulations, the global input fluxes of these compounds are set equal to the burial fluxes diagnosed at
the end of the model spin-up. These input fluxes are jointly denoted as "weathering flux". These fluxes are further attributed to weathering of organic material, of $CaCO_3$, of $CaSiO_3$, and to volcanic $CO_2$ outgassing. The flux of P is assigned to weathering of organic material and related C and ALK fluxes are computed by multiplication of the P flux with the Redfield ratios P:C:Alk = 1:117:-17 for organic material. Similarly, the Si flux is assigned to $CaSiO_3$-weathering and the related ALK flux is computed using Si:ALK=1:2 based on the simplified equation for $CaSiO_3$-weathering: $2CO_2 + H_2O + CaSiO_3 \rightarrow Ca^{2+} + 2HCO_3^- + SiO_2$
(Colbourn et al., 2013). The remaining ALK flux is attributed to $CaCO_3$-weathering with the stoichiometric ratio C:ALK=1:2 following from $CO_2 + H_2O + CaCO_3 \rightarrow Ca^{2+} + 2HCO_3^-$. The volcanic flux is the remaining flux needed to balance the C input flux. The diagnosed fluxes at the end of the spin up are 0.24 $GtC\,yr^{-1}$ for weathering of organic material, 0.14 $GtC\,yr^{-1}$ for $CaCO_3$ weathering, 0.08 $GtC\,yr^{-1}$ for volcanic $CO_2$ outgassing, and 6.96 $Tmol\,Si\,yr^{-1}$ for $CaSiO_3$-weathering. The isotopic signature of the weathering carbon corresponds to the respective signature of the burial fluxes and amounts to $\delta^{13}C$ = -9.2
‰, intermediate between isotopically light organic carbon ($\delta^{13}C$ = -20.5 ‰) and heavier $CaCO_3$ ($\delta^{13}C$ = 2.9 ‰). During

experiments, weathering fluxes of $CaCO_3$ and $CaSiO_3$, and accordingly DIC, $DI^{13}C$, and alkalinity, are allowed to vary as a function of surface air temperature, runoff, and net primary productivity (NPP). The parameterization of this weathering feedback follows equations described in detail in Colbourn et al. (2013). The runoff dependence is parameterized as a function of temperature and for the productivity feedback the NPP from the coupled LPX-Bern model is used. We apply weathering

feedbacks in the global average 0-D version (Colbourn et al., 2013, 2015), the equations are given in appendix B.

The Bern3D model is coupled to the Land surface Processes and eXchanges (LPX) model v1.4 (Lienert and Joos, 2018) which is based on the Lund-Potsdam-Jena (LPJ) model (Sitch et al., 2003). In LPX, coupled nitrogen, water, and carbon cycles are simulated and vegetation composition is determined dynamically and represented with 20 (15 on natural and 5 on anthropogenically used land) plant functional types (PFTs). The PFTs compete within their bioclimatic limits for resources.

$CO_2$ assimilation of plants is implemented following Farquhar et al. (1980) and Haxeltine and Prentice (1996a, b) and isotopic discrimination during photosynthesis is calculated according to Lloyd and Farquhar (1994). The carbon cycles of Bern3D and LPX-Bern are coupled through carbon and isotope exchange fluxes between the land and atmosphere and between the ocean and atmosphere. Climate change information from the Bern3D is passed on to LPX-Bern via a pattern scaling approach (Stocker et al., 2013). Spatial anomaly patterns in monthly temperature and precipitation, derived from a 21st century simu-

lation with the Community Climate System Model (CCSM4), are scaled by the anomaly in global monthly mean surface air temperature as computed interactively by the Bern3D. These anomaly fields are added to the monthly baseline (1901 to 1931 AD) climatology of the Climate Research Unit (CRU) (Harris et al., 2014). A more detailed description of the LPX-Bern model can be found in Keller et al. (2017) and Lienert and Joos (2018).

## 2.2 Experimental protocol

The Bern3D model is spun-up over 60 thousand years ($\mathrm{kyr}$) to a pre-industrial steady state with 1850 AD boundary conditions. Atmospheric $CO_2$ is prescribed to 284.7 $\mathrm{ppm}$, $\delta^{13}C$ to -6.305 ‰, and $\Delta^{14}C$ to 0 ‰. For LPX, the land area under anthropogenic use is fixed to its 1850 AD extent and kept constant in all simulations. The LPX-Bern model is spun-up uncoupled under identical 1850 AD boundary conditions over 2,500 years. Bern3D and LPX-Bern are then coupled and equilibrated for 500 years, again under fixed 1850 AD boundary conditions. After the spin-up, atmospheric $CO_2$ and its isotopic ratio are calculated

interactively, with enabled carbon–climate feedbacks. Pulse experiments are started from the 1850 AD equilibrium state at nominal year -100. After 100 years, i.e., during nominal year 0 of the simulations, an external flux of carbon with characteristic isotopic signatures is removed from or added to the model atmosphere at a constant rate over the year. Runs are continued for 100 $\mathrm{kyr}$ to simulate the redistribution of the added carbon and isotopes in the Earth system.

Table 1 summarizes run names and key characteristics of all simulations. Pulse sizes are varied between -250 $\mathrm{GtC}$ (removal)

to +500 $\mathrm{GtC}$ (addition) with $\delta^{13}C$ of -24 ‰ and $\Delta^{14}C$ of 0 ‰. This corresponds to a hypothetical, sudden uptake or release of carbon from the land biosphere with a typical C3-like $\delta^{13}C$ signature. For simplicity and to ease interpretation, we set the $\Delta^{14}C$ of these land biosphere carbon pulses to the atmospheric signature; thus, we assume that the released material has been recently assimilated and neglect the small depletion in $^{14}C$ of "young" plant, litter, and soil material. In sensitivity experiments, $\Delta^{14}C$ is set to -500 ‰ and -1,000 ‰ to mimic the release of old, dead organic carbon, for example from buried peat or

**Table 1.** Overview of simulations.

| Run name | Pulse size (GtC) | $\delta^{13}$C (‰) | $\Delta^{14}$C (‰) | model setup |
|---|---|---|---|---|
| *ctrl* | 0 | – | – | std. |
| *land biosphere C source* | | | | |
| $WEA_{-250}$ | -250 | -24 | 0 | std. |
| $WEA_{-100}$ | -100 | -24 | 0 | std. |
| $WEA_{100}$ | 100 | -24 | 0 | std. |
| $WEA_{250}$ | 250 | -24 | 0 | std. |
| $WEA_{500}$ | 500 | -24 | 0 | std. |
| *old organic C source (old soils / fossil fuels)* | | | | |
| $^{14}C_{-500}$ | 100 | -24 | -500 | std. |
| $^{14}C_{dead}$ | 100 | -24 | -1000 | std. |
| $WEA_{5000}$ | 5000 | -28 | -1000 | std. |
| *radiocarbon production anomaly* | | | | |
| $^{14}C_{only}$ | – | – | (100 GtC $\times^{14}$R$_{std}$)$^{a}$ | std. |
| *sensitivity to model setup* | | | | |
| $SED_{500}$ | 500 | -24 | 0 | no wea.[b] |
| $SED_{ctrl}$ | 0 | – | – | no wea.[b] |
| $CLO_{500}$ | 500 | -24 | 0 | closed[c] |
| $CLO_{ctrl}$ | 0 | – | – | closed[c] |
| $4box_{500}$ | 500 | -24 | 0 | 4-box land[d] |
| $4box_{ctrl}$ | 0 | – | – | 4-box land[d] |
| $LGM_{500}$ | 500 | -24 | 0 | LGM[e] |
| $LGM_{100}$ | 100 | -24 | 0 | LGM[e] |
| $LGM_{ctrl}$ | 0 | – | – | LGM[e] |

[a]The number of $^{14}$C atoms corresponding to 100 GtC of carbon with $\Delta^{14}$C=0 ‰ is added.

[b] Setup without CaCO$_3$ and CaSiO$_3$ weathering feedbacks.

[c]"Closed system": atm-ocean-land without ocean sediment module and CaCO$_3$ and CaSiO$_3$ weathering feedbacks.

[d] Setup with 4-box land biosphere instead of LPX

[e] Setup with 4-box land biosphere instead of LPX under LGM conditions.

permafrost soils instead of young plant-derived material. In another simulation, the isotopic signatures are set to the signature for the modern mix of fossil fuels. Finally, we also performed a run where only $^{14}$C, but no $^{12}$C and $^{13}$C is added to represent a change in radiocarbon production in the atmosphere. All these simulations were performed with fully interactive ocean sediments and enabled $CaCO_3$ and $CaSiO_3$ weathering feedbacks. The influence of the weathering feedback and of ocean-sediment interactions is quantified using two additional simulations. First, the weathering input flux is kept time invariant ("no weathering feedback"), Second, the sediment module is, in addition, not included in a so-called "closed" atmosphere-ocean-land model setup.

We also assess the influence of glacial climate boundary conditions and of a different land biosphere model in two additional simulations, $LGM_{500}$ and $4box_{500}$. In $4box_{500}$ the Bern3D setup and spin-up is as in $WEA_{500}$, except that an atmospheric $CO_2$ concentration of 278 ppm instead of 284.7 ppm is used and the pulse is released in the first time-step of simulation year 100, rather than being distributed over year 100. The model configuration includes reactive sediments and weathering feedbacks. A four-box terrestrial biosphere model (Siegenthaler and Oeschger, 1987), that allows for $CO_2$ fertilization, instead of LPX-Bern is coupled to the Bern3D in these runs. In $LGM_{500}$ the model is forced by glacial boundary conditions instead of preindustrial conditions as in $4box_{500}$ and in $WEA_{500}$. $CO_2$ is set to 180 ppm and northern hemisphere ice sheets are set to Last Glacial Maximum coverage (Peltier, 1994). It has to be noted, however, that by forcing the model with 180 ppm during spin-up leads to less carbon stored in the ocean under LGM than PI conditions and results from $LGM_{500}$ should be treated with some caution. Differences in results between $WEA_{500}$ and $4box_{500}$ are due to differences in the two land biosphere models. Differences between $4box_{500}$ and $LGM_{500}$ are exclusively due to differences in climatic boundary conditions.

## 2.3 Data reduction

The remaining fraction of a pulse-like perturbation or IRF at time $t$ after the pulse release is defined for atmospheric $CO_2$ as

$$\text{IRF}(CO_{2,a})(t) = \frac{\Delta CO_{2,a}(t)}{\Delta CO_{2,a}^{ini}} = \frac{CO_{2,a}(t) - CO_{2,a}^{ctrl}(t)}{P/\left(2.12 \text{ GtC ppm}^{-1}\right)}. \tag{1}$$

$\Delta$ indicates a perturbation, here evaluated as difference between a simulation with pulse release and a corresponding control simulation. $P$ indicates the pulse size in GtC which yields the initial atmospheric $CO_2$ perturbation in ppm when divided by $2.12 \text{ GtC ppm}^{-1}$. The superscript $ctrl$ refers to the control simulation without pulse, $ini$ to the (maximum) initial perturbation assuming an instantaneous carbon release, and subscript $a$ to the atmosphere. The IRF for $\delta^{13}C_a$ is:

$$\text{IRF}(\delta^{13}C_a)(t) = \frac{\delta^{13}C_a(t) - \delta^{13}C_a^{ctrl}(t)}{\Delta\delta^{13}C_a^{ini}}. \tag{2}$$

$\Delta\delta^{13}C_a^{ini}$ denotes the initial isotopic perturbation in the atmosphere and equals:

$$\Delta\delta^{13}C_a^{ini} = \frac{P}{N_{a,0} + P} \cdot (\delta^{13}C_P - \delta^{13}C_{a,0}). \tag{3}$$

$\delta^{13}C_P$ indicates the isotopic signature of the pulse, the subscript 0 the value prior to the pulse, and $N_a$ the atmospheric carbon inventory. Analogous equations to Eqs. 2 and 3 hold for $\Delta^{14}C$.

An analytical expression for the IRF is calculated from experiment $WEA_{500}$. The IRF is fitted by the sum of five exponential terms to

$$\text{IRF}(\text{CO}_{2,a}(t)) = a_0 + \sum_{i=1}^{5} a_i \cdot exp\left(\frac{-t}{\tau_i}\right) \; for \; t \geq 0 \tag{4}$$

using a least-squares optimization routine in Python. The coefficients $a_i$ are fractions of the perturbation, each associated with a timescale $\tau_i$, and $a_0$ the fraction of the perturbation remaining constantly in the atmosphere ($\tau_0 = \infty$). The sum over all six coefficients $a_i$ equals 1. Modeled values are well reproduced with n=5 exponents. Differences between the original model output and the fit are insignificant for practical applications. Finally, we note that the IRF or remaining atmospheric fraction is in our experimental setup always smaller than one: the carbon is added at a constant rate during year 0 in the pulse simulations and part of this carbon is already taken up by the ocean during this initial year. Similarly, the IRF for $\delta^{13}\text{C}_a$ and $\Delta^{14}\text{C}_a$ is fitted with n=4 exponents.

Principal component empirical orthogonal function (PC-EOF) analysis is used to fit the $^{13}\text{C}$ isotopic perturbations of dissolved inorganic carbon (DIC) in the ocean ($\Delta\delta^{13}\text{C}_{\text{DIC}}$). To this end, the Python package $eofs$ (Dawson, 2016) is used to calculate principal component timeseries ($PC(t)$) and the spatial empirical orthogonal functions ($EOF(\boldsymbol{x})$) from the volume-weighted 4-D field of $\Delta\delta^{13}\text{C}_{\text{DIC}}(t,\boldsymbol{x})$. PCs and EOFs are then used to build a cost-efficient substitute model of the spatio-temporal evolution of $\Delta\delta^{13}\text{C}_{\text{DIC}}$ in response to an atmospheric pulse perturbation. The output frequency for marine 3-D tracer fields after the perturbation is every 10 years during the first 1 kyr, every 200 years until 10 kyr and every 1 kyr thereafter. In the PC-EOF calculation this gives more weight to the first part of the simulation, where changes in the perturbation are larger.

## 3 Results

### 3.1 Earth system response to a pulse release of carbon

**Impulse response functions for atmospheric CO$_2$ and $\delta^{13}$C:** We first describe the IRF for the atmospheric CO$_2$ perturbation using experiment $WEA_{250}$ (Fig. 1a). The initial spike in response to the carbon release is followed by a substantial decline over the first decades due to carbon uptake by the upper ocean and land biosphere. The IRF is reduced to less than 40 % within the first century and decreases further on the timescale of ocean mixing to less than 16 % by year 2,000. The fit by exponentials (Table 2) yields time scales of 6, 47, and 362 years for this initial decline. They represent the continuum of carbon overturning time scales within the ocean and the land biosphere as well as time scales governing air-sea and air-land carbon exchange.

CaCO$_3$ compensation and weathering-burial imbalances remove the atmospheric perturbation on multi-millennial timescales such that after 10 kyr less than 8 % and after 100 kyr less than 2 % of the CO$_2$ perturbation remain airborne (Fig. 1a). The fit by exponentials shows that a fraction of 11.2 % of the initial perturbation is removed with a time scale of 5.5 kyr. We associate these values with the process of CaCO$_3$ compensation. A remaining fraction of 4 to 5 % is removed with a time scale of 69 kyr (Table 2), linked to enhanced silicate rock weathering. The jumps visible at around 20 kyr in Fig. 1 arise from a sea ice-albedo feedback in the control run. They appear most pronounced in the experiments with small perturbations. The change

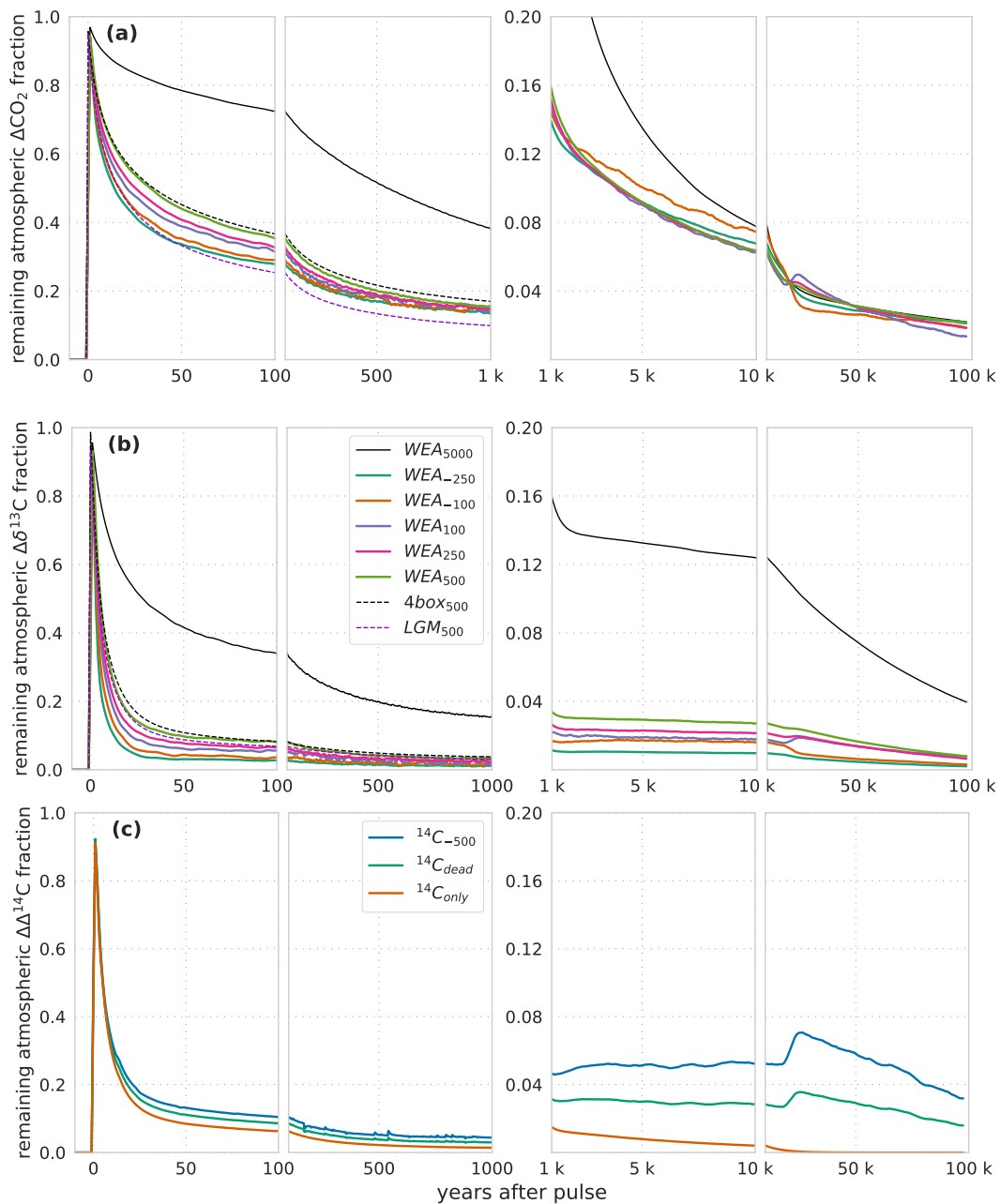

**Figure 1.** (a) $CO_2$, (b) $\delta^{13}C$, and (c) $\Delta^{14}C$ perturbation in the atmosphere for different simulations with different pulse sizes and model configurations. The isotopic signature is $\delta^{13}C$=-24 ‰ for all pulse sizes except the 5,000 GtC pulse with $\delta^{13}C$=-28 ‰. See Table 1 and section 2.2 for details of the experimental setup of the runs. Data are filtered with a moving average of 1,000 years in the interval 1 kyr - 10 kyr and 5,000 years in the interval 10 kyr - 100 kyr for better visibility. Factorial simulations ($WEA_{500}$, $SED_{500}$, $CLO_{500}$, $4box_{500}$, and $LGM_{500}$) for the time interval 1 kyr to 100 kyr are shown in Fig. 2 for better visibility.

**Table 2.** Fit parameters for IRF($\Delta CO_2(t)$) and IRF($\Delta \delta^{13}C_a(t)$) of $WEA_{500}$ and IRF($\Delta\Delta^{14}C_{a,dead}(t)$), and IRF($\Delta\Delta^{14}C_{a,only}(t)$). Note that in the case of IRF($\Delta\Delta^{14}C_{a,dead}(t)$) the actual fitting was done only with data until simulation year 15,000 to avoid complications from the sea-ice feedback in the control run (see main text and Fig. 1c).

| | CO$_2$ | | $\delta^{13}C_a$ | | $\Delta^{14}C_{dead}$ | | $\Delta^{14}C_{only}$ | |
|---|---|---|---|---|---|---|---|---|
| $i$ | $a_i$ | $\tau_i$ (yr) | $a_i$ | $\tau_i$ (yr) | $a_i$ | $\tau_i$ (yr) | $a_i$ | $\tau_i$ (yr) |
| 0 | 0.008 | - | 0.001 | - | 0.006 | - | 0.001 | - |
| 1 | 0.044 | 68,521 | 0.034 | 74,781 | 0.03 | 146,4130 | 0.017 | 7,510 |
| 2 | 0.112 | 5,312 | 0.034 | 436 | 0.0004 | 1,243 | 0.082 | 209 |
| 3 | 0.224 | 362 | 0.092 | 75 | 0.173 | 85 | 0.274 | 15 |
| 4 | 0.310 | 47 | 0.840 | 6 | 0.790 | 5 | 0.629 | 4 |
| 5 | 0.297 | 6 | | | | | | |

in the control run in CO$_2$ is less than 1 ppm and less than 0.03 ‰ in $\delta^{13}CO_2$. These jumps are not of further relevance for our discussion. In comparison to others, our results show, up to a few percent, lower remaining fractions for the 5,000 GtC pulse ($WEA_{5000}$) than results presented by Colbourn et al. (2015) and Lord et al. (2016). The remaining airborne fraction of the CO$_2$ perturbation is 5 % and less after 100 kyr in all studies.

Compared to CO$_2$, the atmospheric $\delta^{13}C$ perturbation is initially removed much faster (Fig. 1b). Within the first decades, IRF($\delta^{13}C_a$) is reduced to below 10 %. The further decline is slower and 1 kyr after the pulse, less than 3 % of the initial perturbation remains airborne. The remaining atmospheric perturbation is finally reduced to ≤1 % of the initial perturbation at the end of the simulation (Fig. 1b). The fit by exponentials for IRF($\delta^{13}C_a$) (Table 2) shows that 84 % of the perturbation are removed with a timescale of 6 years, reflecting the initial fast decline seen in Fig. 1b when the perturbation is taken up by the

upper ocean and living vegetation. About 9 % and 3 % are removed on timescales of 75 and 436 years, respectively, reflecting the decadal and centennial uptake of the perturbation by the land biosphere and deeper ocean. ∼3 % of the perturbation are finally removed with a timescale of ∼74 kyr by weathering-sedimentation processes, in agreement with the e-folding timescale of the decline (∼70 kyr; see further below). However, it has to be kept in mind, that these removal timescales result from a fit and thus other equally accurate solutions are possible to approximate the continuum of removal timescales.

**Processes:** Next, we address the processes behind the different temporal evolution of CO$_{2,a}$ and $\delta^{13}C_a$. We focus on the ocean as ocean uptake is thought to dominate on long time scales. A perturbation in atmospheric CO$_2$ or in $^{13}CO_2$ is communicated by air-sea gas exchange to the surface ocean and further transported, mainly by physical processes, to the deep ocean. The fundamental difference between the IRF for CO$_{2,a}$ and $\delta^{13}C_a$ is linked to the aquatic carbonate chemistry and the associated equilibrium between dissolved CO$_2$, bicarbonate and carbonate ions (CO$_2$+ H$_2$O $\Leftrightarrow$ HCO$_3^-$ + H$^+$ $\Leftrightarrow$ CO$_3^{2-}$+2H$^+$)

(Dickson et al., 2007). The removal of an atmospheric CO$_2$ perturbation is co-controlled by this acid-base carbonate chemistry (Revelle and Suess, 1957). In contrast, the removal of a perturbation in the isotopic ratio $^{13}CO_2/^{12}CO_2$ is hardly affected by this chemical buffering, because $^{13}CO_2$ and $^{12}CO_2$ are affected approximately equally by the acid-base chemistry reactions. The chemical buffering diminishes the uptake capacity of the ocean for CO$_2$, but not for the isotopic ratio. Correspondingly, the

IRF for $CO_{2,a}$ decreases less rapidly than the IRF for $\delta^{13}C_a$ and IRF($\delta^{13}C_a$) is smaller than IRF($CO_{2,a}$) over the simulation period.

For an illustrative, semi-quantitative analysis, we developed approximate expressions for the IRF for $CO_{2,a}$ and $\delta^{13}C_a$ for time scales up to 2 kyr as explained in the appendix. Simplifying assumptions are that equilibrium between the atmosphere
and a fraction of the ocean with the initial carbon inventory $N_{o,0}$ is assumed, while land biosphere carbon uptake and other climate-carbon cycle feedbacks are neglected. Changes in land biosphere carbon storage can be considerable, in particular during the first century (see Fig. 3a). However, the role of the land carbon stock changes becomes smaller, the more carbon is taken up by the ocean and is thus for simplicity not considered in the analytical expression for $CO_2$. For $CO_2$ the expression reads:

$$\mathrm{IRF}_\infty(CO_{2,a}) = \frac{N_{a,0}}{N_{a,0} + \frac{1}{\xi}N_{o,0}}, \quad \text{for } 0 \leq t \leq 2 \text{ kyr}, \tag{5}$$

The subscript $a$ indicates the atmosphere and $o$ the ocean, 0 the time prior to the pulse and $\infty$ the time when an effective ocean volume with the initial carbon inventory $N_{o,0}$ is equilibrated with the atmosphere. $\xi$ describes the influence of the acid-base carbonate chemistry on the relationship between the relative perturbation in the $CO_2$ partial pressure and in dissolved inorganic carbon (DIC) (Revelle and Suess, 1957). $\xi$ varies with environmental conditions and increases with the size of the
$CO_2$ perturbation. $\xi$ is on the order of 10 for small pulse sizes (here -250 to 500 GtC). The dependency of IRF($CO_2$) on the size of the carbon pulse is implicitly captured in the variable $\xi$. The key information of eq. 5 is that the magnitude of the IRF is determined by the ratio of the (equilibrated) ocean carbon inventory and the buffer factor, $N_{o,0}/\xi$. The smaller this ratio, the larger is IRF($CO_{2,a}$).

For $\delta^{13}C_a$, we also consider the initial carbon inventory of the land biosphere ($N_{b,0}$), as a substantial amount of the isotopic
perturbation is contained in the land biosphere on multi-centennial to millennial timescales (Fig. 3b). The corresponding expression reads (for the derivation see the appendix):

$$\mathrm{IRF}_\infty(\delta^{13}C_a) = \frac{N_{a,0} + P}{N_{a,0} + N_{o,0} + N_{b,0} + P} \cdot \frac{(\delta^{13}C_P - \delta^{13}C_{mean})}{(\delta^{13}C_P - \delta^{13}C_{a,0})}, \quad \text{for } 0 \leq t \leq 2 \text{ kyr}. \tag{6}$$

Equation 6 shows that IRF($\delta^{13}C_a$) is described by the product of two ratios. The first ratio corresponds to the ratio of the carbon inventory in the atmosphere immediately after the pulse release to the total carbon inventory in the ("equilibrated")
atmosphere-ocean-land biosphere system. It tells us that the pulse perturbation is mixed uniformly (see also appendix eq. A20) and in proportion to the carbon inventories of the different reservoirs (atmosphere, ocean, land biosphere). The second fraction is about 1.2 and thus a correction term of order 20 % for small pulses sizes as typically applied in this study. It arises as the isotopic signature of the ocean and land biosphere is different from that of the atmosphere. $\delta^{13}C_{mean}$ corresponds to the weighted mean signature of the atmosphere, ocean and land biosphere, with the fraction of the pulse found in each of these
reservoirs as weights (appendix eq. A27). Remarkably, the buffer factor $\xi$ does not directly show up in eq. 6. The reason is, as noted already above, that $CO_2$ and $^{13}CO_2$ are both about equally affected by the aquatic chemistry. In turn, their ratio, i.e., $\delta^{13}C$, is hardly affected by this acid-base buffering.

Equation 5 and eq. 6 reveal the fundamental difference between a perturbation in $CO_2$ (and $^{13}CO_2$) and in the isotopic ratio, $\delta^{13}CO_2$. For small pulse sizes ($P \to 0$), neglecting the influence of the land biosphere and the correction term in eq. 6 for $\delta^{13}C$, eq. 6 and eq. 5 are formally equal, except that a buffer factor $\xi$ of 1, instead of $\sim 12$ applies for $\delta^{13}C_a$. Equations 6 and 5 show: the pulse perturbation is distributed between the atmosphere and ocean proportionally to their initial carbon inventories $N_{a,0}$

and $N_{o,0}$ in the case of $\delta^{13}C_a$, but proportionally to $N_{a,0}$ and $N_{o,0}/\xi$ in the case of $CO_2$. As a consequence, the perturbation in the ratio $\delta^{13}C_a$ is apparently removed much faster than the perturbation in the concentration $CO_{2,a}$, despite that the time scales to transport a perturbation from the surface ocean to the deep ocean by advection, convection, and diffusion are the same.

We may illustrate this difference in the removal rate numerically. $\xi$ is about 12 for a small pulse, $N_{a,0}$ is 600 GtC, $N_{o,0}$ is 37,400 GtC for the whole ocean. Further, the top 300 m with a carbon inventory of 2,600 GtC are ventilated within

approximately a decade. through air-sea gas exchange and upper ocean physical transport processes (circulation, mixing). Then, the evaluation of eq. 5 yields for a small pulse that around 73 % and 16 % of the initial $CO_2$ perturbation are still found in the atmosphere after about 10 years and 2 kyr, respectively. It has to be noted here that we assume $\xi$ constant for small perturbations. With increasing pulse sizes, also $\xi$ increases, leaving larger airborne fractions (c.f. $WEA_{500}$ versus $WEA_{5000}$ in Fig. 1a after 1-2 kyr). Regarding $\delta^{13}C$, the corresponding fractions (eq. 6) are much smaller and 12 % to 31 % and 1 % to 3

% for small pulse sizes (-250 GtC to 500 GtC; see appendix, text after eq. A14 for more details). We assume the perturbation to have mixed with 500 GtC and 2,700 GtC on land after 10 yr and 2 kyr, respectively. These values are roughly in agreement with the values shown in Fig. 1a for pulse sizes of up to 500 GtC.

Turning to intermediate time scales (2 kyr $\leq t \leq$ 20 kyr), $CaCO_3$ compensation further reduce the atmospheric $CO_2$ perturbation and removes the differences in IRF($CO_{2,a}$) arising from different pulse sizes (Fig. 1a). The process of $CaCO_3$ com-

pensation is briefly explained as follows. $CO_2$ is taken up by the ocean and partly reacts to form bicarbonate ($HCO_3^-$) and hydrogen ($H^+$) ions. This makes the water more acidic; the carbonate ion concentration ($[CO_3^{2-}]$) is reduced and the water becomes corrosive to $CaCO_3$. In turn, sedimentary $CaCO_3$ dissolves to $CO_3^{2-}$ and $Ca^{2+}$ ions, partly removing the perturbations in $[CO_3^{2-}]$ and $[H^+]$, a so-called seafloor neutralization. Under more acidic conditions, less alkalinity and carbon, in the form of $CaCO_3$, is buried in the lithosphere than delivered by weathering input; this excess input leads to an increase in alkalinity

and DIC further removing the perturbations in $[CO_3^{2-}]$ and $[H^+]$, a so-called terrestrial neutralization, and a new equilibrium between weathering and burial fluxes is established. This chain of processes raises the alkalinity twice as much as the DIC concentration in the ocean and additional $CO_2$ is taken up from the atmosphere.

Again, an approximate expression for the IRF($CO_{2,a}$) is developed. This applies for the time when $CaCO_3$ compensation for the pulse perturbation is completed (see appendix):

$$\text{IRF}_{CaCO_3}(CO_{2,a}) = 2 \cdot 1.5 \cdot \frac{N_{a,0}}{N_{o,0}}, \ \text{ for } t \sim 20 \text{ kyr} \tag{7}$$

The factor 2 in eq. 7 arises from the acid-base carbonate chemistry, the factor 1.5 from the change in the ocean carbon inventory due to excess dissolution (or burial) of $CaCO_3$, equivalent to $\sim 0.5 \times P$. The factor 1.5 is model dependent (see e.g. Archer et al., 1998) and also slightly different from previous Bern3D model versions. This equation shows that the airborne fraction after $CaCO_3$ compensation depends on the ratio of the initial atmosphere ($N_{a,0}$) to ocean carbon inventory ($N_{o,0}$). Therefore,

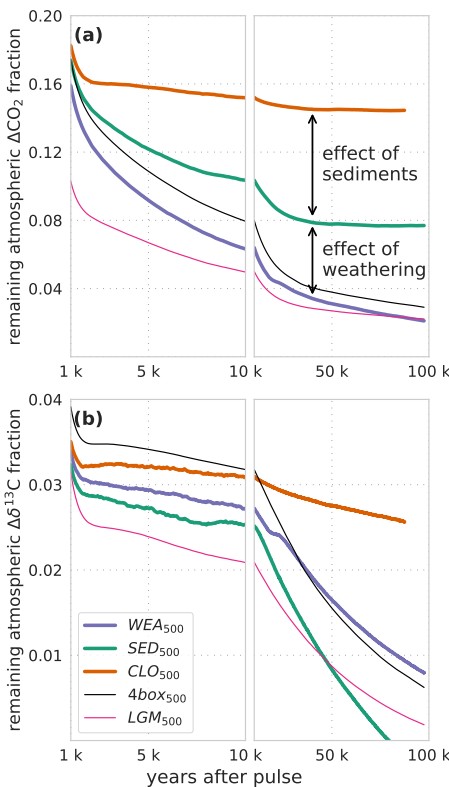

**Figure 2.** (a) $CO_2$ and (b) $\delta^{13}C$ perturbation in the atmosphere for the factorial simulations $WEA_{500}$, $SED_{500}$, $CLO_{500}$, $4box_{500}$, and $LGM_{500}$. Shown is the time interval 1 kyr to 100 kyr. Data are filtered with a moving average of 1,000 years in the interval 1 kyr - 10 kyr and 5,000 years in the interval 10 kyr - 100 kyr for better visibility. Note that experiment $CLO_{500}$ was run for 90 kyr instead of 100 kyr.

$IRF_{CaCO_3}$ is (approximately) independent of the pulse size. Equation 7 successfully explains why $IRF(CO_{2,a})$ for pulse sizes from -250 to +5,000 GtC take on a nearly identical value of 5 % around ∼20 kyr after the pulse.

$IRF(\delta^{13}C_a)$ stays nearly constant between 1 and 10 kyr (Fig. 1b). This implies that $CaCO_3$ compensation has little influence on the removal of the $\delta^{13}C$ perturbation. Reasons are that the change in ocean carbon inventory due to $CaCO_3$ dissolution
5 (burial) is small compared to the total ocean inventory and that the isotopic signature of $CaCO_3$ is similar to that of DIC.

On even longer time scales, the $CaSiO_3$ weathering feedback further removes the remaining perturbation in $CO_{2,a}$ with an e-folding time scale on the order of 70 kyr. The perturbation in $\delta^{13}C_a$ is removed by the burial flux of organic carbon and $CaCO_3$. This burial flux removes the isotopic perturbation to the lithosphere. Surprisingly, the removal rate is substantially faster than expected from the residence time of carbon and highly sensitive to the gradients in the perturbation of $\delta^{13}C$ ($\Delta\delta^{13}C$)
10 within the ocean as further discussed below in the context of the factorial simulations and in the appendix.

**Factorial simulations:** We now turn to the factorial simulations to further quantify the effect of $CaCO_3$ compensation, burial fluxes, and $CaCO_3$ and $CaSiO_3$ weathering feedbacks on the $CO_2$ and $\delta^{13}C$ perturbations. The evolution of $IRF(CO_{2,a})$ for experiment $SED_{500}$ (no terrestrial weathering feedback but with marine sediments) is comparable to $WEA_{500}$ until simulation

year 1,000 and differs thereafter due to varying weathering rates in $WEA_{500}$ (Figs. 1a and 2a). There is no addition of alkalinity to the ocean from enhanced terrestrial weathering in $SED_{500}$ and $\Delta CO_{2,a}$ levels out at a higher value than in $WEA_{500}$, with about 8 % of the perturbation remaining constantly airborne in $SED_{500}$. This value is in agreement with previous studies (e.g. Archer et al., 1998). It is somewhat higher than suggested by eq. 7 as this equation represents a simplification.

$CaCO_3$ compensation leads in both, $WEA_{500}$ and $SED_{500}$, to a further removal of $\Delta CO_{2,a}$ on multi-millennial timescales as seen between 1 and 10 kyr in Fig. 2a. After 10 kyr, IRF($CO_{2,a}$) decreases further as a result of weathering-burial imbalances in the $CaCO_3$ flux, also referred to as terrestrial neutralization (e.g. Archer et al., 1998; Colbourn et al., 2015). The e-folding timescale for this terrestrial neutralization from $SED_{500}$ is about 11 kyr, higher than the value provided by Archer et al. (1998) (8.5 kyr) but in the range (8 to 12 kyr) provided by Colbourn et al. (2015).

Comparing $\Delta\delta^{13}C_{atm}$ of $SED_{500}$ and $WEA_{500}$, the evolution is initially comparable but on multi-millennial timescales the isotopic perturbation is, somewhat surprisingly and in contrast to the $CO_2$ perturbation, removed slower in $WEA_{500}$ than in $SED_{500}$ (Fig. 2b). In the case of $SED_{500}$ the e-folding timescale is on the order of 20 to 30 kyr. This is much shorter than expected from the mean residence time of carbon in the atmosphere-ocean-land biosphere system of about 90 kyr (Carbon inventory: $\sim$40,000 GtC; burial flux: 0.46 GtC yr$^{-1}$). In simulation $WEA_{500}$, with weathering feedbacks enabled, the long-
term removal timescale of $\Delta\delta^{13}C_{atm}$ ($\sim$70 kyr) is also shorter than the residence time of carbon in the atmosphere-ocean-land biosphere system. The isotopic perturbation is larger in the surface ocean than in the ocean mean. In turn, the perturbation is removed more efficiently than expected from the residence time of carbon (see Fig. A1). The isotopic signature of the car-bon burial flux is mainly determined by the upper ocean isotopic signature which is more depleted than the ocean mean for $WEA_{500}$. In the case of $SED_{500}$ this gradient is even stronger and lasts longer (Fig. A1, see section 3.2 for further explana-
tion). These results highlight the importance of spatial patterns within the ocean for the removal of an isotopic perturbation even for these very long-time scales. However, the absolute perturbation in $\delta^{13}C_a$ for these small pulse sizes and with $\delta^{13}C$=-24 ‰ is rather small after a few kyr. For example, in the case of $SED_{500}$, $\Delta\delta^{13}C_a$ decreases to below 0.05 ‰ after $\sim$60 kyr. The interannual variability from the LPX-Bern is of the same magnitude and also model drift starts to play a role.

In scenario $CLO_{500}$, we exclude the effect of ocean sediment interactions and treat the atmosphere, ocean, and land bio-
sphere as a closed system. Considering ocean mixing times of 1-2 kyr, we would expect the $CO_2$ perturbation for experiment $CLO_{500}$ to reach a steady state within 2 kyr. The slow response timescale of peat carbon in the LPX-Bern model, however, leads to a further decline of $\Delta CO_{2,a}$ over multiple millennia (Fig. 2a). About 15 % of the initial perturbation remain constantly airborne in the closed system model setting as expected from eq. 5. $\delta^{13}C_a$ levels out around 2 kyr at around 3 ‰, as expected from eq. 6. The further decrease on multi-millennial timescales results from model drift in the LPX-Bern.

Next, we compare $WEA_{500}$, where LPX-Bern is used as land model, with $4box_{500}$, where a 4-box land biosphere is used. Differences in IRF($CO_{2,a}$) between $WEA_{500}$ and $4box_{500}$ are due to the different land biosphere models. The four-box land biosphere model takes up less of the $CO_2$ perturbation than LPX-Bern. The root mean squared deviation in IRF($CO_{2,a}$) from the two simulations is less than 1 % of the perturbation. Differences are most pronounced between simulation year 1 and 10,000 (Fig. 2a). Differences in IRF($\delta^{13}C_a$) (Fig. 2b) are largest in the first decades after the pulse and the root mean square deviation
in IRF($\delta^{13}C_a$) amounts to less than 1 % between the two simulations.

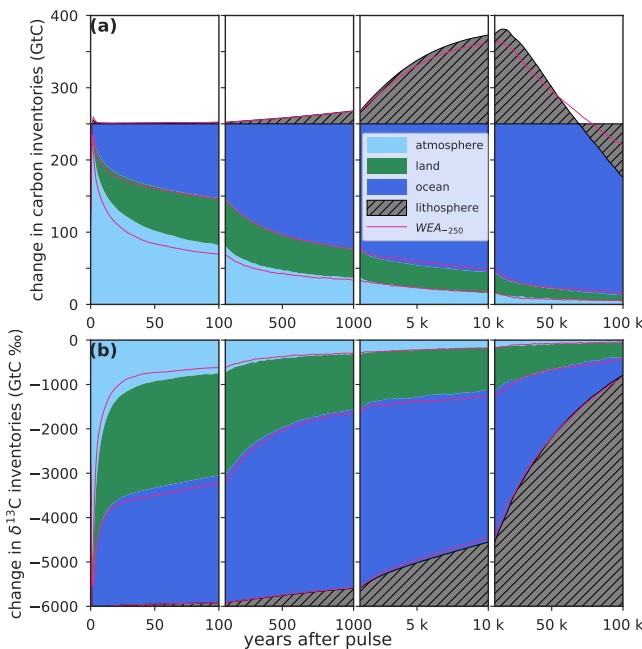

**Figure 3.** Inventories of the (a) carbon and (b) $\delta^{13}$C perturbation for experiment $WEA_{250}$. Shown are inventories of the perturbations in the atmosphere (light blue), the land biosphere (green), and the ocean (blue). Hatched areas indicate the influence of changes in the lithosphere (including sediments) due to $CaCO_3$ compensation and imbalances in the weathering and burial fluxes on the carbon and isotopic inventories of the ocean-atmosphere-land system. For carbon, this inventory is elevated above the pulse perturbation of 250 GtC until about 70 kyr and depleted thereafter. For $\delta^{13}$C, the atmosphere-ocean-land inventory is continuously reduced over the course of the simulation. Magenta lines show results for experiment $WEA_{-250}$ multiplied by -1.

Finally, we compare the simulations $LGM_{500}$, forced by glacial boundary conditions, and $4box_{500}$, forced by preindustrial conditions. Differences between the two simulations are solely due to the difference in climate and $CO_2$ forcing. IRF($CO_{2,a}$) is lower by a few percent in $LGM_{500}$ compared to $4box_{500}$ (Fig. 2a). The difference is largest in the first 10 kyr and up to 12 % of the perturbation. The root mean square deviation between the two simulations amounts to ~2 %. The larger

5 oceanic carbon uptake in $LGM_{500}$ than in $4box_{500}$ can be explained by the higher alkalinity simulated for LGM than for preindustrial conditions. A higher alkalinity implies lower values of the Revelle factor $\xi$ and a larger carbon uptake capacity of the ocean. IRF($\delta^{13}C_a$) shows a similar pattern with lower values for the LGM background compared to the preindustrial background (cf. black dotted and dark violet dotted lines in Fig. 1b). The difference can be understood with the help of eq. 6; inserting the small glacial value for the atmospheric carbon inventory (381.6 GtC) in the nominator of the first term yields

10 IRF$_\infty$($\delta^{13}C_a$)=0.027 compared to IRF$_\infty$($\delta^{13}C_a$)=0.032 when using the preindustrial value (589.4 GtC) in the equation. These estimates of IRF($\delta^{13}C_a$) are in good agreement with the results from $LGM_{500}$ and $4box_{500}$, respectively (Fig. 2b). In summary, the influence of glacial compared to preindustrial boundary conditions on IRF($\delta^{13}C_a$) appears modest in our model.

**The budgets of the carbon and carbon isotope perturbation:** Next, we discuss the budgets of the carbon and $^{13}$C perturbations. Figure 3 shows results for simulation $WEA_{250}$, with a pulse release of 250 GtC, and for simulation $WEA_{-250}$, with a pulse removal of 250 GtC. Differences between these two runs are clearly visible in Fig. 3, but the general evolution of the budgets is similar. In the following, we give numerical results for simulation $WEA_{250}$. After 100 years, about 82 GtC are still airborne and the land and the ocean carbon inventories have increased by about 65 GtC and 105 GtC, respectively (Fig. 3a). As time evolves, the oceanic carbon perturbation grows at the cost of the atmosphere and land. On multi-millennial timescales, carbonate compensation and enhanced terrestrial carbonate and silicate weathering increase ocean alkalinity and thereby the uptake capacity for atmospheric $CO_2$ and the remaining atmospheric $CO_2$ fraction decreases. Nevertheless, the carbon perturbation in the atmosphere-ocean-land system exceeds the 250 GtC of the pulse as more carbon is added to the ocean through increased $CaCO_3$ weathering and redissolution/reduced burial of marine sediments (hatched area in Fig. 3a). The maximum perturbation is reached after ∼16 kyr with a total of about 381 GtC in the atmosphere-ocean-land system. Thereafter, this inventory decreases due to weathering-burial imbalances. After 100 kyr, about 5 GtC of the perturbation remain airborne, 8 GtC are stored in the land biosphere, and the ocean contains 163 GtC more carbon than before the perturbation. Accordingly, 74 GtC of the initial perturbation are lost from the atmosphere-ocean-land system through increased marine burial compared to terrestrial weathering input (hatched area extends below 250 GtC line in Fig. 3a).

The isotopic perturbation budget is established in units of GtC ‰. Any isotopic inventory is defined as the product of the corresponding carbon inventory in GtC multiplied by its isotopic signature in ‰. The role of the land biosphere in removing the $\delta^{13}C_a$ perturbation is clearly visible (Fig. 3b). After 100 years, the ocean contains ∼50 % of the isotopic perturbation (in GtC ‰), the land biosphere about 35 % and the atmosphere less than 15 %. On millennial timescales, the perturbation is slowly removed from the atmosphere-ocean-land system. The $\delta^{13}C$ perturbation of the burial flux (combined POC and $CaCO_3$) follows the negative $\delta^{13}C$ perturbation in surface DIC. Therefore the burial flux is what ultimately removes the $\delta^{13}C$ perturbation from the atmosphere-ocean-land biosphere system (Fig. 3b).

We further illustrate the contribution of the burial and weathering fluxes to the carbon and isotope budgets for the simulation with ($WEA_{500}$) and without weathering feedbacks ($SED_{500}$) in Fig. 4. The inventory in the relatively fast exchanging pools – atmosphere, ocean, reactive sediments, land biosphere - increases by ∼350 GtC in addition to the pulse input of 500 GtC over the course of simulation $SED_{500}$, leading to a total increase of 850 GtC. The increase additional to the pulse is mainly due to a reduction in the burial of $CaCO_3$. In contrast, $CaCO_3$ burial is enhanced in simulation $WEA_{500}$ and ∼430 Gt of carbon are removed from the fast exchanging reservoirs. This removal is only partly compensated by enhanced weathering (∼380 GtC), leading to a reduction in the inventory of the fast exchanging pools from 500 GtC immediately after the pulse to 453 GtC at 100 kyr. Changes in POC burial fluxes are negligible for the carbon budget.

Turning to the isotopic pertubation budget, the initial perturbation due to the pulse input of -12,000 GtC ‰ is mainly removed by anomalies in the POC burial flux both in $SED_{500}$ (∼10,000 GtC ‰) and $WEA_{500}$ (∼7,000 GtC ‰). Changes in $CaCO_3$ burial and weathering mitigate about a quarter (∼2,900 GtC ‰) of the initial isotopic perturbation in the two simulations. This leaves a relatively small isotopic inventory perturbation of about +1,100 and -1,600 GtC ‰ in the fast exchanging reservoirs at 100 kyr in simulations $SED_{500}$ and $WEA_{500}$.

The contributions of the anomalies in the POC and $CaCO_3$ burial fluxes may be approximately attributed to changes in the signature ($\Delta\delta$) and to changes in flux ($\Delta F$) relative to the initial flux, $F_0$, and signature, $\delta_0$, before the perturbation:

$$\Delta(F \cdot \delta) = F_0 \cdot \Delta\delta + \Delta F \cdot \delta_0 \tag{8}$$

The isotopic perturbation of the POC burial flux is mainly mediated by a change in the signature of the organic carbon buried ($\Delta\delta^{13}C(POC)$). For example, the signature of the total POC burial flux is on average 0.29 ‰ more negative than the corresponding input from weathering in simulation $WEA_{500}$. This change in signature of the total POC burial flux leads to an effective reduction of the isotopic perturbation. The change in POC burial flux ($\Delta F(POC)$) is, as mentioned above, negligible. Both changes in the signature of the total $CaCO_3$ burial flux and changes in $CaCO_3$ burial contribute significantly to the isotopic perturbation. At the end of simulation $WEA_{500}$, the mean $\delta^{13}C$ signature of $CaCO_3$ burial is on average 0.14 ‰ more negative than the input from weathering. In turn, the term $F_0 \cdot \Delta\delta$ for $CaCO_3$ burial contributes $\sim$ -3,100 GtC ‰ to the mitigation of the initial isotopic perturbation. This is partly counteracted by excess burial of $CaCO_3$, with $\Delta F \cdot \delta_0$ equal $\sim$1,300 GtC ‰. This yields an overall burial contribution of $\sim$ -1,800 GtC ‰. The $\delta^{13}C$ perturbation is further removed by increased weathering input of $CaCO_3$ (see methods), contributing about 1,100 GtC ‰ over 100 in $WEA_{500}$.

In summary, the isotopic perturbation is mainly mitigated by a change in the signature of the mean POC burial flux. The smaller contributions from the $CaCO_3$ cycle to the isotopic budget is related to both a change in the signature of the mean $CaCO_3$ burial flux and to changes in burial and weathering carbon fluxes.

**Influence of pulse size:** The IRF for atmospheric $CO_2$ and $\delta^{13}C$ is sensitive to the magnitude of the pulse size (Fig. 1a,b) and we next address related differences between simulations. For $CO_2$, the remaining atmospheric fraction is higher for larger than for smaller pulse sizes. Differences between different small pulses (-250 GtC to 500 GtC) are most significant in the first decades; differences between the large ($WEA_{5000}$) and any small pulse (Fig. 1a) are most significant in the first millennia. Several processes contribute to these differences.

First, non-linearities in the carbonate chemistry contribute to differences in IRF($CO_{2,a}$) in the first centuries to millennia, where the perturbation is largely contained in the atmosphere-ocean-land biosphere system. We see from eq. 5 that a controlling factor is the oceanic buffer factor. As pulse size increases so does the buffer factor, leading to higher values of IRF($CO_{2,a}$) for $WEA_{5000}$ compared to smaller pulse sizes (Fig. 1a).

Second, productivity on land depends non-linearly on $CO_2$ (Farquhar et al., 1980) and the loss of carbon from the land biosphere in response to a negative pulse is larger than the uptake in response to a positive pulse of same size. This contributes to the initially smaller remaining atmospheric fraction for negative pulses (see e.g., $WEA_{250}$ and $WEA_{-250}$ in Fig. 1a and 3a). The difference is most strongly pronounced in the first centuries and millennia but prevails until the end of the simulation.

Third, the response in marine export production is also non-linear for pulses of same absolute size but different sign. The initial decrease in export production after a positive pulse is smaller than the increase after a negative pulse of same size, thereby affecting surface ocean $CO_2$ concentrations and thus air–sea gas exchange and partly leveling out the effect from the land biosphere. For $\delta^{13}C_a$, the perturbation is smaller for negative, versus positive, pulses of the same magnitude (Fig. 1). The

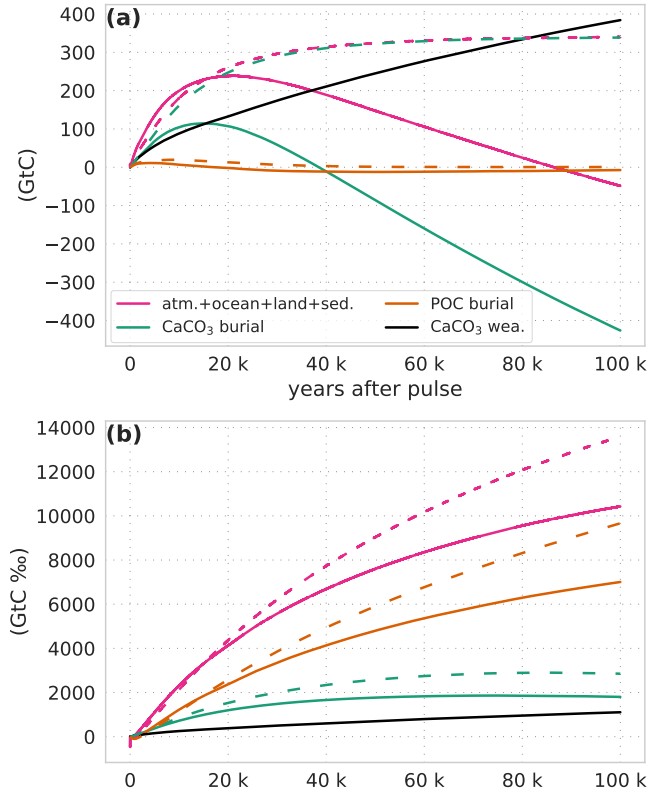

**Figure 4.** The perturbation budget for (a) carbon and (b) carbon isotopes. Solid lines show results from experiment $WEA_{500}$, dashed lines from experiment $SED_{500}$. In both panels, the pulse (500 GtC, -12,000 GtC ‰) is subtracted from the combined atmosphere-ocean-land-reactive sediment inventory. Burial fluxes are plotted inversely and the sum of burial (green, orange lines) and weathering (black) fluxes yields the combined inventory (pink).

difference can be understood with the help of eq. 6. Evaluation of eq. 6 (see appendix) yields an IRF($\delta^{13}C_a$) of ~2 % for the pulse addition of 250 GtC ($WEA_{250}$), but of only ~1 % for a corresponding removal ($WEA_{-250}$).

**Radiocarbon:** Finally, we discuss the $\Delta^{14}C$ perturbation experiments. In experiment $^{14}C_{only}$, a positive pulse of $^{14}C$ is added to the atmosphere but there is no perturbation in $CO_2$, analogous to a radiocarbon production pulse by atomic bomb

5  tests or cosmic rays. The initial evolution of the IRF($\Delta^{14}C_a$) is very similar to the one of IRF($\delta^{13}C_a$) (Fig. 1b,c). On longer timescales, radioactive decay additionally removes the perturbation, so that after ~20 kyr less than 0.1 % of the perturbation is left airborne. In simulations $^{14}C_{-500}$ and $^{14}C_{dead}$, a radiocarbon depleted carbon source of 100 GtC is added to the atmosphere causing a negative perturbation in $\Delta^{14}C$ and a positive perturbation in $CO_2$ in the atmosphere. As visible from Fig. 1a for $WEA_{100}$, a small percentage of the $CO_2$ perturbation remains airborne even after 100 kyr. This explains the persistence of

10  a small atmospheric perturbation in $\Delta^{14}C$, well beyond the life time of the initially added $^{14}C$. IRF($\Delta^{14}C_a$) is larger for $^{14}C_{-500}$ than for $^{14}C_{dead}$ on these very long time scales. This is because the perturbation in $CO_2$ and, in turn, the long-term

perturbation in $\Delta^{14}C_a$ is the same for both experiments. However, the initial perturbation in $\Delta^{14}C$ - the denominator of the response function (see eq. 2) - is smaller in $^{14}C_{-500}$ than $^{14}C_{dead}$.

## 3.2 The $\delta^{13}C$ perturbation of marine dissolved inorganic carbon and its approximation by PC-EOF

The ocean mean perturbation in $\delta^{13}C$ of DIC ($\Delta\delta^{13}C_{DIC}$) increases in absolute magnitude during the first millennium, as the atmospheric perturbation is propagated to the ocean (Fig. 5, thick gray line). The most negative mean perturbation (-0.23 ‰ for experiment $WEA_{500}$) occurs ~1,500 years after the pulse release (Fig. 5), reflecting ocean mixing timescales. Thereafter, the $\delta^{13}C_{DIC}$ perturbation decreases in magnitude as a result of weathering-burial imbalances (see section 3.1, Fig. 3 and 5). At the end of the simulation, less than 15 % of the maximum perturbation (-0.03 ‰) remains in the ocean.

Deep ocean invasion of the $\delta^{13}C_{DIC}$ perturbation is first accomplished in the North Atlantic Deep Water (NADW) region. After 100 years, the signal has already propagated to a depth of ~3.5 km in the North Atlantic, whereas it is confined to the upper ocean (>1 km) in the rest of the ocean (Fig. 6a). At year 1,000, the $\delta^{13}C_{DIC}$ perturbation is rather uniformly distributed in the ocean. Only the deep North Pacific (region of oldest waters; ideal age >1 kyr in the Bern3D) and waters at intermediate depths in the Southern Ocean show a smaller perturbation (Fig. 6c). On multi-millennial times, $\Delta\delta^{13}C_{DIC}$ decreases throughout the ocean. The perturbation, however, is not entirely uniformly distributed as expected from a completed ocean mixing by physical transport. More negative $\Delta\delta^{13}C_{DIC}$ values are present in the upper ocean and in North Atlantic Deep Water than in the intermediate and deep Southern Ocean, in Antarctic Bottom Water in the Atlantic, and the deep Pacific (Fig. 6e,g). This is due to changes in the fractionation during air-sea gas exchange at higher temperatures, enhanced by reduced fractionation during phytoplankton growth at elevated $CO_2$ concentrations. Both processes lead to more negative $\delta^{13}C$ signatures in the surface ocean. Additionally, gross air-sea fluxes are increased as a result of the higher $CO_2$ concentrations, decreasing the air-sea disequilibrium. This leads to more negative $\delta^{13}C$ signatures in mid- and low-latitude surface waters and less negative signatures in cold high-latitude waters (see e.g. Menviel et al., 2015). This surface to deep $\Delta\delta^{13}C_{DIC}$ gradient is dampened in the case of experiment $WEA_{500}$ by enhanced weathering of $CaCO_3$ on land, adding carbon with a positive $\delta^{13}C$ signature (~2.9 ‰ ) to the surface ocean. As discussed above, these spatial $\Delta\delta^{13}C$ gradients are important and co-govern the removal rate of the $\delta^{13}C$ perturbation. The larger than average $\delta^{13}C$ perturbation in the surface is communicated to phytoplankton and zooplankton, feeding the burial flux of biogenic material. This accelerates the removal of the $\delta^{13}C$ perturbation in comparison to a uniformly mixed ocean.

Next, the temporal evolution of the $\delta^{13}C_{DIC}$ perturbation and its spatial pattern are described by PC-EOF (Fig. 7). The first PC-EOF pair captures approximately the $\delta^{13}C_{DIC}$ evolution after 1 kyr, while the second and third PC-EOF pairs capture, toghether with the first pair, the decadal-to-century scale penetration of the perturbation into the ocean (Fig. 5). The first EOF pattern (Fig. 7a) shows negative values throughout the ocean, with a pattern similar to that modeled for year 50,000 (Fig. 6g). The corresponding PC timeseries increases during the first decades after the pulse, followed by a slow decline on multi-millennial timescales (Fig. 7b). This first PC captures the evolution of ocean mean $\Delta\delta^{13}C_{DIC}$ well after 1 kyr (Fig. 5 blue versus gray lines). In contrast to the first EOF, the second EOF has a clear dipole pattern with negative values in surface and intermediate waters and in the NADW region, and positive values below ~ 1 km depth (Fig. 7c). The corresponding PC

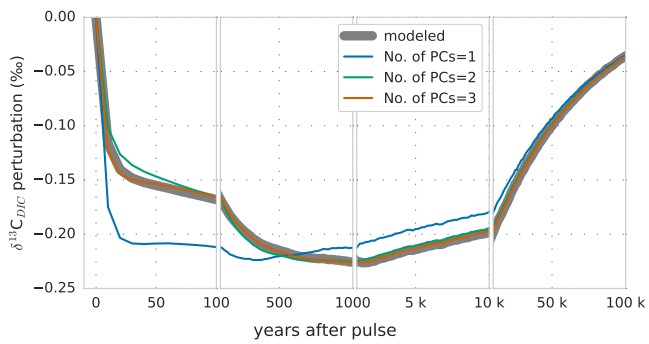

**Figure 5.** Modeled global-average perturbation of the isotopic signature of dissolved inorganic carbon in the ocean ($\delta^{13}C_{DIC}$; thick gray line) in response to a carbon pulse to the atmosphere of 500 GtC with a $\delta^{13}C$ signature of -24 ‰. The thin colored lines show the perturbation as reconstructed with one (blue), two (green), and three (orange) principal components.

timeseries has high positive values in the first decades of the simulation, declining over the next centuries, becoming negative around year 500, and approaching zero on multi-millennial timescales. In combination with the first PC-EOF pair this leads to strongly negative $\delta^{13}C$ values in surface and intermediate waters in the first centuries of the simulation, whereas on longer timescales the depth gradient in $\Delta\delta^{13}C_{DIC}$ disappears. The third PC-EOF, finally, is of importance during the first century (Fig. 7b) and captures regional features, e.g. related to North Atlantic Deep Water or Intermediate Waters (Fig. 7d).

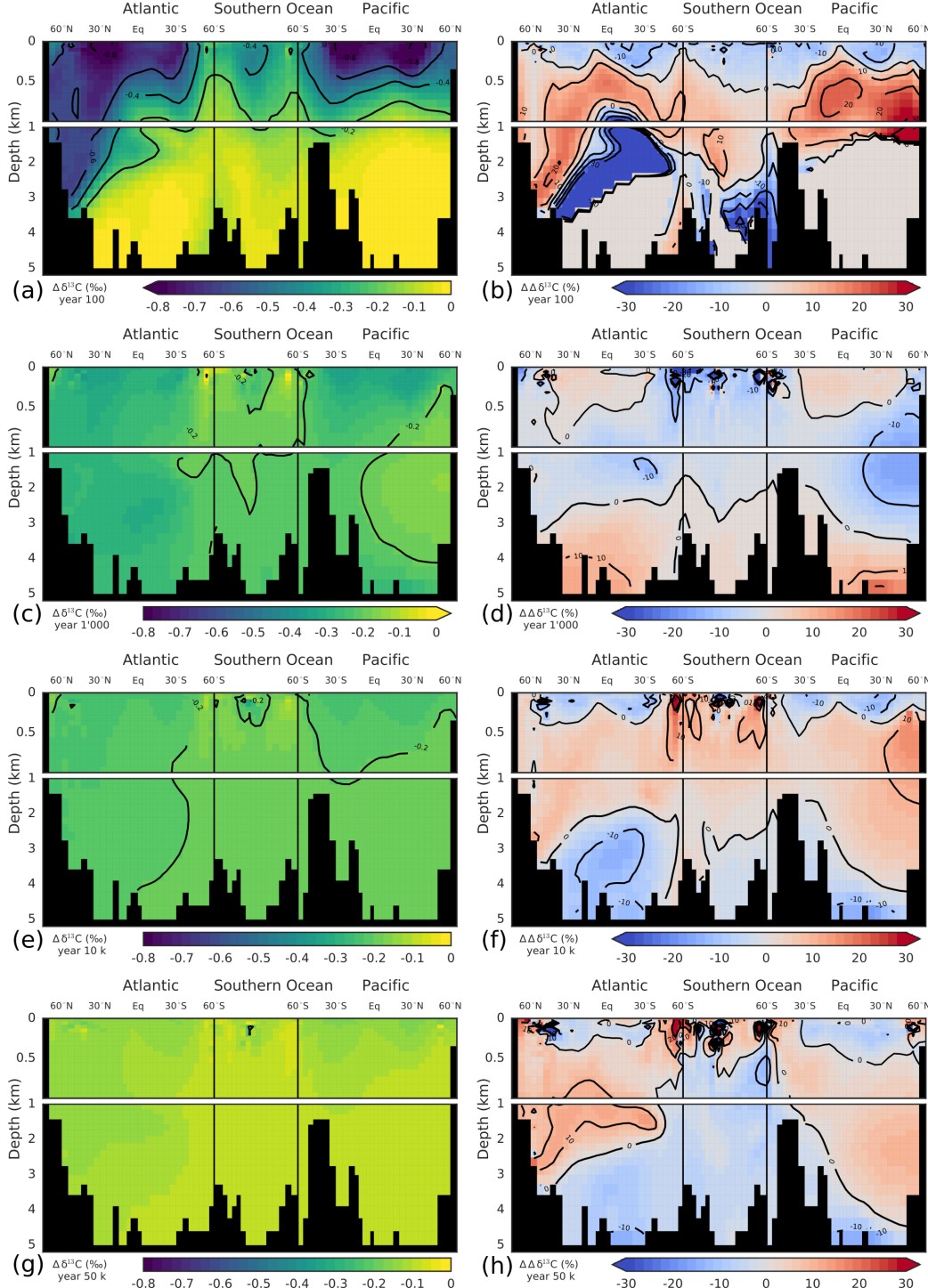

**Figure 6.** Modeled $\delta^{13}C_{DIC}$ perturbation in ‰ (a,c,e,g) and difference between modeled and reconstructed $\delta^{13}C_{DIC}$ perturbation using the first three principal components. These differences are shown in % of the modeled perturbation (b,d,f,h). Shown are time slices at: 100 yr (a,b); 1 kyr (c,d); 10 kyr (e,f); and 50 kyr (g,h) after the pulse release of carbon. Note that in panel (b) perturbation values smaller than |0.05| ‰ (abyssal Atlantic and Pacific) have been masked for better visibility as the perturbation has not yet propagated into these regions (see panel (a)).

The global mean $\Delta\delta^{13}\text{C}_{DIC}$ is well described by the first two PCs (Fig. 5, green vs gray line). The third PC-EOF pair mostly adds performance during the first century (cf. green and orange lines in Fig. 5) and reduces the root mean squared error (RMSE) between reconstructed and modeled global mean $\Delta\delta^{13}\text{C}_{DIC}$ from 0.003 ‰ with two PC-EOF pairs to 0.001 ‰ with three PC-EOF pairs used in the reconstruction.

For the reconstruction of the spatio-temporal evolution of $\Delta\delta^{13}\text{C}_{DIC}$, we will now use the first three PC-EOF pairs. The RMSE between the modeled and reconstructed 3-D field is highest (0.1 ‰) for the first 3-D output at year 10 after the pulse and decreases to 0.03 ‰ at year 100. As visible from Fig. 6a and b, largest deviations between modeled and reconstructed $\Delta\delta^{13}\text{C}_{DIC}$ are found in the thermocline of the Pacific and in the deep northern Atlantic. Further, the modeled perturbation has not yet propagated to the deep Pacific and Atlantic by year 100 (Fig. 6a), whereas the PC-EOF reconstruction displays a

$\Delta\delta^{13}\text{C}_{DIC}$ signal in these water masses. The performance of the reconstruction increases over the course of the simulation. The RMSE of the $\Delta\delta^{13}\text{C}_{DIC}$ fields amounts to 0.01 ‰ at 1 and 10 kyr and decreases to 0.007 ‰ 50 kyr after the pulse. Overall, reconstructing $\Delta\delta^{13}\text{C}_{DIC}$ with three PC-EOF pairs shows good results and deviations from the modeled $\Delta\delta^{13}\text{C}_{DIC}$ are of the order of 10 % of the perturbation and smaller on centennial to millennial timescales (Fig. 6d,f,h). To reconstruct the modeled evolution of $\Delta\delta^{13}\text{C}_{DIC}$ precisely during the first few centuries after a perturbation, more than three PC-EOF pairs

are necessary. In summary, the spatio-temporal evolution of $\Delta\delta^{13}\text{C}_{DIC}$ over ∼1,000 years to 100 kyr after a pulse input of carbon into the atmosphere is described accurately and conveniently by three PC-EOFs pairs.

## 4   Discussion and conclusion

The aim of this study is to investigate the evolution of a $\delta^{13}\text{C}$ perturbation in the Earth system in response to a carbon input to the atmosphere on time scales from centuries to 100 kyr. To this end, the response of the Bern3D-LPX model was probed by

pulse-like emissions of carbon to the atmosphere from an external source such as land biosphere carbon release or fossil fuel burning. The $\delta^{13}\text{C}$ response is compared to that of atmospheric $CO_2$ and radiocarbon.

     The impulse response (or Green's) functions (IRF) for atmospheric $CO_2$ and its $\delta^{13}\text{C}$ signature are fitted by a sum of exponential functions. Additionally, we show that the spatio-temporal evolution of a $\delta^{13}\text{C}_{DIC}$ perturbation is reasonably represented with three PC-EOF pairs. Deviations between the PC-EOF reconstruction and modeled $\delta^{13}\text{C}_{DIC}$ results are largest

in the first decades after the pulse and are on the order of 10 % of the perturbation on millennial time scales. This allows the construction of a computationally efficient substitute model which can be applied to investigate, e.g., reconstructed $\delta^{13}\text{C}_{DIC}$ variations from marine sediments in future studies. The evolution of the atmospheric $CO_2$ perturbation in the coupled Bern3D-LPX model is comparable to results from earlier studies addressing the pulse-like input of carbon into the atmosphere (e.g. Maier-Reimer and Hasselmann, 1987; Siegenthaler and Joos, 1992; Archer et al., 1998; Ridgwell and Hargreaves, 2007;

Archer et al., 2009; Colbourn et al., 2015; Lord et al., 2016). The inclusion of peat lands in the LPX-Bern model adds a (small) additional sink on multi-millennial timescales, previously not considered in such simulations. For the carbon isotopes, corresponding long-term pulse response simulations that consider ocean-sediment and weathering feedbacks are lacking in the literature. The "short-term" response from years to one or two millennia is consistent with earlier results (Joos et al., 1996).

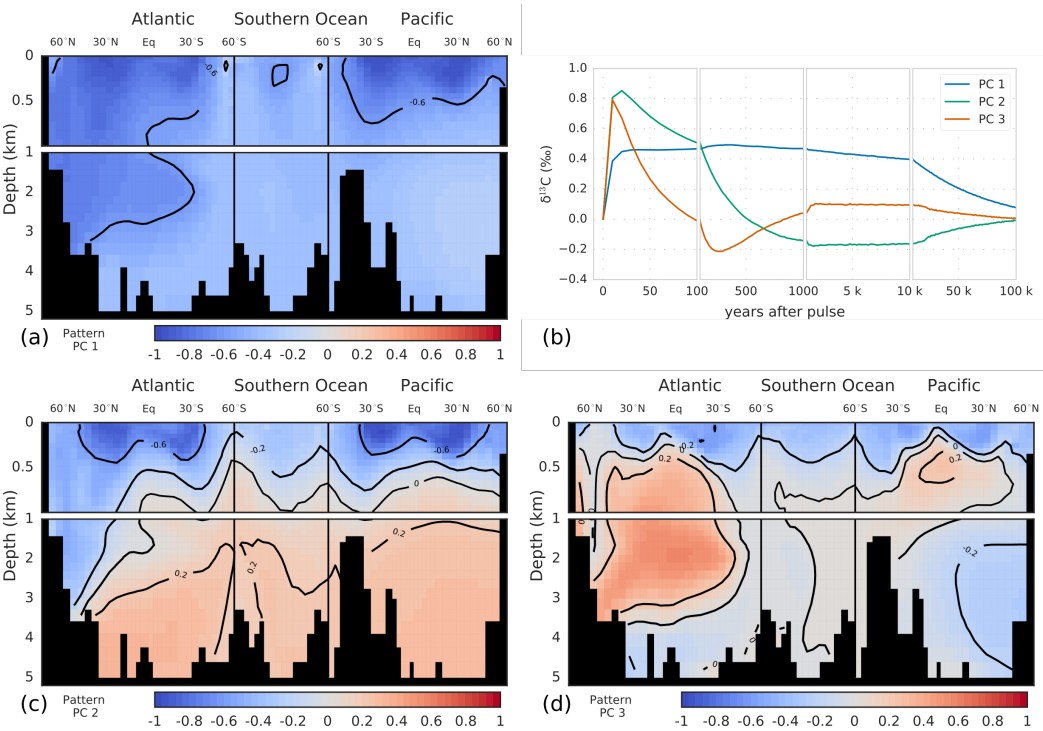

**Figure 7.** EOF pattern of the (a) first, (c) second, and (d) third principal component used to represent the spatial perturbation in $\Delta\delta^{13}$C of dissolved inorganic carbon in the ocean. Corresponding PC timeseries are shown in (b).

We quantify the processes leading to the different removal timescales of atmospheric $CO_2$ and $\delta^{13}C_{atm}$ perturbations with the help of factorial experiments. The role of different processes are further illustrated by analytical descriptions derived from a two box atmosphere-ocean and three box atmosphere-ocean-land model provided in the appendix. The removal of an atmospheric perturbation in the isotopic ratio $^{13}$C/$^{12}$C is not buffered by marine carbonate chemistry, unlike for $CO_2$. The atmospheric $\delta^{13}$C perturbation is partitioned among the ocean, land biosphere, and atmosphere roughly in proportion to their carbon inventories over about 2 kyr. This leaves only a very small fraction airborne (1 to 3 % for pulses of a few hundred GtC) after the signal is mixed within these three reservoirs. In contrast, a substantial fraction of carbon emissions and the $CO_2$ perturbation remains airborne for millennia. On time scales of a few millennia to 20 kyr, the initial atmospheric $CO_2$ perturbation is lowered by carbonate compensation to a fraction of about 8 %. In contrast, this chemical buffering hardly affects the atmospheric $\delta^{13}$C perturbation. On even longer time scales, the $CO_2$ and the isotopic perturbation is removed by different, though related processes. For $\delta^{13}$C, the continuous flux of biogenic calcium carbonate and organic carbon particles carries the isotopic perturbation from the surface ocean to the lithosphere. For $CO_2$, the excess weathering of silicate rocks in response to the $CO_2$ and climate perturbation adds alkalinity to the ocean which leads to a complete removal of the atmospheric $CO_2$ perturbation.

Gradients in the $\delta^{13}C_{DIC}$ perturbation strongly influence the long-term removal timescale of the isotopic perturbation. In the pulse experiments, $\Delta\delta^{13}C$ is enlarged in the surface relative to the deep ocean by temperature-dependent fractionation during air-sea gas exchange (Mook, 1986), $CO_2$-dependent fractionation by phytoplankton (Freeman and Hayes, 1992), and altered air-sea disequilibrium resulting from changes in gross exchange fluxes due to altered $CO_2$. Changes in $CaCO_3$ weathering on

the other hand, diminish the surface perturbation (e.g. experiment $WEA_{500}$ vs. $SED_{500}$ in Fig. 1a). All these processes lead to spatial gradients in the $\delta^{13}C$ perturbation in the ocean. As a result of these gradients in $\Delta\delta^{13}C$, the isotopic perturbation is removed faster by particle burial than expected from a uniformly mixed ocean. This highlights the importance to resolve spatial structures in the perturbation to represent its removal time scales. More generally and beyond pulse release experiments, any process that changes the $\delta^{13}C$ signature of the relentless flux of organic and calcium carbonate particles, will influence the

$\delta^{13}C$ loss flux from the ocean-atmosphere-land biosphere system to the lithosphere (for examples see Jeltsch-Thömmes et al. (2019); Roth et al. (2014); Tschumi et al. (2011)).

Our results have consequences for the interpretation of the difference in $\delta^{13}C$ between similar climate states such as the Penultimate Glacial Maximum (PGM) and the Last Glacial Maximum (LGM). Substantial temporal $\delta^{13}C$ differences between these periods are recorded in ice cores (Schneider et al., 2013; Eggleston et al., 2016) and in marine sediments (Hoogakker et al.,

2006; Oliver et al., 2010). Different mechanisms, such as changes in the $\delta^{13}C$ signature of weathering and burial fluxes, varying contribution of volcanic outgassing of $CO_2$, and changes in the amount of carbon stored in the land biosphere, especially in yedoma and permafrost soils, have been discussed for the PGM-LGM $\delta^{13}C$ offset (e.g. Lourantou et al., 2010; Schneider et al., 2013). An internal reorganization of the marine carbon cycle without considering changes in burial may not explain the offset. With no changes in the weathering-burial balance, the mass in the atmosphere-ocean-land system remains constant. Then, a

change in atmospheric $\delta^{13}C$ would require an opposing $\delta^{13}C$ change in the ocean. This appears in conflict with marine and ice core records which suggest that $\delta^{13}C$ increased both in the ocean and in atmosphere by about 0.3 ‰ between the PGM and LGM (Hoogakker et al., 2006; Oliver et al., 2010; Schneider et al., 2013; Eggleston et al., 2016). However, internal marine carbon cycle reorganizations, e.g., due to changes in circulation, may have altered $\delta^{13}C$ in the surface ocean, and in turn $\delta^{13}C$ of the burial flux, and thereby the balance between weathering input and burial.

The Bern3D results suggest that changes in organic carbon storage on land (or in the ocean) are not a likely explanation of the isotopic offset between the PGM and LGM. The $\delta^{13}C$ signal of a transfer of organic carbon of plausible magnitude to the atmosphere and ocean would have been attenuated too much over time to cause the observed PGM-LGM $\delta^{13}C$ difference. Schneider et al. (2013) estimated required changes in land carbon storage to match the ice core and marine temporal offsets in $\delta^{13}C$ using preliminary Bern3D results for pulse release simulations. Our results show, in agreement with these earlier results,

that the isotopic perturbation associated with a terrestrial carbon release is attenuated to less than 15 % within two millennia for pulse sizes of up to 5,000 GtC and declines thereafter (see black line in Fig. 1b). This would require several thousand GtC to have been stored additionally in the land biosphere during the LGM compared to the PGM in order to explain the $\delta^{13}C$ offset (see also Schneider et al., 2013). This amount seems large in light of estimated total carbon stocks of $\sim$1500 GtC in perennially frozen soils in Northern Hemisphere permafrost regions today (Tarnocai et al., 2009) and an even smaller inventory

at the LGM (Lindgren et al., 2018).

Taken together, these ad-hoc considerations suggest that long-term imbalances in the weathering (including volcanic emissions) and burial fluxes appear to be a plausible cause for the $\delta^{13}$C differences between PGM and LGM. However, further work, which considers the transient evolution of $CO_2$, $\delta^{13}$C, and other proxies, is required to gain further insight into the contributions of individual mechanisms to long term $\delta^{13}$C changes recorded in ice and marine cores.

Further, the different behavior of $CO_2$ and $\delta^{13}C_{atm}$ perturbations has also consequences for centennial scale $CO_2$ and $\delta^{13}C_{atm}$ variations such as during Heinrich Stadial (HS) 4 and 1. Variations in $CO_2$ and $\delta^{13}C_{atm}$ during HS4 and HS1 have been measured on Antarctic ice cores. The data show an increase in $CO_2$ on the order of $\sim$10 ppm over 200-300 years, accompanied by a decrease in $\delta^{13}C_{atm}$ of $\sim$0.2 ‰. This results in a value of roughly -0.02 ‰ per ppm for the ratio of the change in $\delta^{13}$C to the change in $CO_2$ ($r= \Delta\delta^{13}$C / $\Delta CO_2$). A release of terrestrial carbon has been discussed as a possible
cause of these events (Bauska et al., 2016, 2018).

We utilize our pulse response simulations under PI and LGM boundary conditions to estimate the changes in $r$ in response to a transient terrestrial carbon input with an isotopic signature of -24 ‰. The results for $r$ will depend on the evolution of the emissions into the atmosphere. For a pulse-like input at time zero the evolution of $r$ after the pulse is shown in Fig. 8a for the first 2,000 years. $r$ is about -0.08 (-0.06) ‰ ppm$^{-1}$ immediately after the release under LGM (PI) boundary conditions.
The difference between LGM and PI values of $r$ comes from the difference in the atmospheric C and $^{13}$C inventory to which the pulse is added. The magnitude of $r$ is reduced rapidly within the first 25 years and remains approximately constant at -0.015 (-0.01) ‰ ppm$^{-1}$ thereafter. Thus, $r$ recorded in ice after a hypothetical pulse release of carbon would be above -0.02 ‰ ppm$^{-1}$ if the age distribution of the ice core samples is wider than a few decades. However, emissions may occur more gradually. For simplicity, we assume an emission pulse sustained over period $T$ according to a two-sided Heaviside function;
terrestrial emissions, $e(t)$, suddenly increase to unity, remain elevated for period $T$, and are then immediately reduced again to zero. $r$ is then readily calculated from the IRFs ($r = \int_0^T dt \cdot e(t) \cdot$ IRF($\delta^{13}$C)($T-t$)$/ \int_0^T dt \cdot e(t) \cdot$ IRF($CO_2$)($T-t$)). The results for $r$ are again plotted in Fig. 8a. $r$ is decreasing with the duration $T$ of sustained emissions and falls below -0.02 ‰ ppm$^{-1}$ for $T$ larger than about 90 (30) years for LGM (PI) boundary conditions. Finally, we also performed simulations with the Bern3D-LPX under PI boundary conditions to account for non-linearity that may not be captured in the IRF representation.
The releases of varying amounts of terrestrial carbon with $\delta^{13}$C = -24 ‰ over 100 to 400 years yield a linear response in $\Delta CO_2$ and $\Delta\delta^{13}C_{atm}$ (Fig. 8b) with a slope $r$ of about -0.01 ‰ ppm$^{-1}$, in agreement with the IRF results.

There are uncertainties in these estimates of $r$. We assumed that the emitted carbon was about -18 ‰ depleted with respect to the atmosphere. The relative uncertainty in this value translates directly to the relative uncertainty in $r$. The release of material that is isotopically even more depleted would bring our model results in better agreement with the ice core estimate of $r$.
Observations and land model simulations suggest that the $\delta^{13}$C depletion in vegetation and soils may vary between -24 ‰ , for heavily discriminated C3 plant material, to -5 ‰ indicative of C4 plant material Keller et al. (2017). C4 plants were likely more abundant during glacials than interglacials (Collatz et al., 1998). Also, the values shown in Fig. 8b for century-scale emissions are 30-year means. A wider gas age distribution would reduce the perturbation in $\delta^{13}C_{atm}$ more than in $CO_2$ (see e.g. Köhler et al., 2011). On the other hand, the $CO_2$ increase may have occurred in less than 100 years (Bauska et al., 2018)
and correspondingly a more negative value of $r$ is expected for such a fast (<100 yr) increase. Given these uncertainties in our

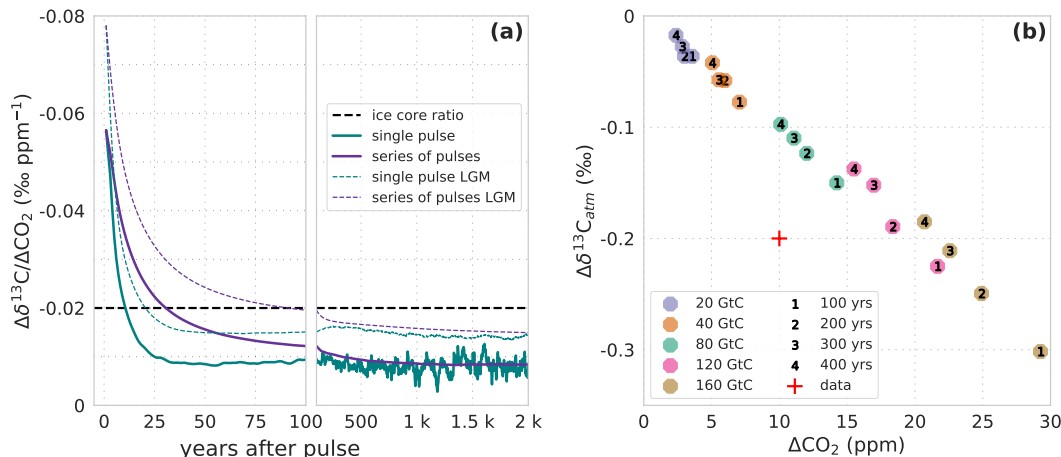

**Figure 8.** (a) Temporal evolution of the ratio of $\Delta\delta^{13}C$ to $\Delta CO_2$ in the atmosphere. Results are from simulations $WEA_{100}$ forced by preindustrial boundary conditions (solid lines) and $LGM_{100}$ forced by Last Glacial Maximum boundary conditions (thin dashed lines). Carbon is released as a single pulse at year 0 (green lines) or assumed to be released at a uniform rate from year 0 to the year plotted (purple lines; see main text). The dashed horizontal line indicates a $\Delta\delta^{13}C$ to $\Delta CO_2$ ratio of -0.2 ‰ to 10 ppm as seen in ice core records for HS1 and HS4. (b) Response in $\Delta CO_2$ and $\Delta\delta^{13}C_{atm}$ to a uniform release of 20, 40, 80, 120, and 160 $GtC$ (colors) of isotopically light carbon ($\delta^{13}C$ = -24 ‰) to the atmosphere over 100, 200, 300, and 400 years (numbers). Shown is the mean over 30 years starting in the last year of the release. Red cross indicates the approx. $\Delta\delta^{13}C$ to $\Delta CO_2$ change as seen in ice core records for HS1 and HS4 (Bauska et al., 2016, 2018).

model estimates as well as in the determination of $r$ from the ice core measurements, we are not in a position to firmly conclude whether a terrestrial carbon release was the sole mechanism for the HS1 and HS4 $CO_2$ events or not. Other mechanisms in addition to a terrestrial release may have contributed to the $CO_2$ and $\delta^{13}C_{atm}$ excursions during HS4 and HS1.

5    We acknowledge that most of the idealized simulations presented in this study were run under preindustrial boundary conditions. A limited set of sensitivity simulations suggest that the IRFs for $CO_2$ and $\delta^{13}C$ are similar under preindustrial and glacial $CO_2$ in our model. The direct influence of lower atmospheric $CO_2$ on IRF($\delta^{13}C$) appears limited. Further work is needed to better quantify the dependency of IRFs on the climate state.

In conclusion, the responses in atmospheric $CO_2$ and in $\delta^{13}C_{atm}$ to a carbon input (or removal) are different, with implications for the interpretation of paleo proxy data. The difference in response is related to the aquatic carbonate chemistry

10   and, on millennial to multi-millennial time scales, to the burial weathering cycle, which affect $CO_{2,a}$ and its isotopic ratio differently. Our study highlights the importance of carbon isotope fluxes to and from the lithosphere, on timescales where the ocean-atmosphere-land biosphere has so far often been assumed to represent a closed system. Perturbations in $\delta^{13}C$ are removed from or modified within the atmosphere-ocean-land biosphere system by the continuous flux of biogenic organic and calcium carbonate particles buried in marine sediments. This isotopic burial flux is substantially influenced by spatial gradients

15   in $\delta^{13}C$ within the ocean, calling for a spatially-resolved representation of the marine carbon cycle to address isotopic change.

Carbon isotope fluxes to and from the lithosphere need to be considered for the interpretation of changes in atmospheric and marine $\delta^{13}C$ during glacial-interglacial and longer periods.

*Data availability.* Pulse response data and EOF patterns and PC timeseries will be made available for download on http://www.climate. unibe.ch/research/research_groups/earth_system_modelling_biogeochemical_cycles/index_eng.html

5  Further data upon request to the corresponding author.

## Appendix A:  Analytical expressions for the Impulse Response Functions of $CO_{2,a}$ and $\delta^{13}C_a$

**Ocean invasion of an atmospheric $CO_2$ perturbation:** The difference in the IRF for carbon and the isotopic ratios can be understood in terms of the aquatic carbonate chemistry. We develop approximate expressions for the IRF of atmospheric $CO_2$ (eq. A7) and $\delta^{13}C$ (eq. A29), starting in this section with IRF($CO_2$).

10    For small pulses, changes in sea surface temperature and in $CO_2$ solubility, changes in ocean circulation, and changes in the marine biological cycle are small and neglected in the following. Changes in land biosphere carbon storage can be considerable, in particular during the first century. However, the role of land carbon stock changes becomes smaller, the more carbon is taken up by the ocean and is neglected for simplicity.

We consider a two box model of the atmosphere (index: $a$) and ocean (index: $o$) with area A=$A_a$=$A_o$ and heights $h_a$ and 15  $h_o$. All boxes are assumed to be well mixed. Before the pulse addition, the model is in equilibrium and the initial $CO_2$ partial pressure is $pCO_{2,0}$=$pCO_{2,a,0}$=$pCO_{2,o,0}$. The initial concentration of dissolved inorganic carbon (DIC) in the ocean is $DIC_0$ and the carbon inventory in the ocean is given by $N_o$=DIC$\cdot h_o \cdot$A. We relate the atmospheric carbon inventory, $N_a$, to a time-varying atmospheric concentration $C_a$ and to the fixed volume of the model atmosphere: $N_a$=$C_a \cdot h_a \cdot$A. We are free to select the units of the model-specific concentration $C_a$. Here, we express $C_a$ in units of $DIC_0$ and set the initial atmospheric 20  concentration $C_{a,0}$ equal to $DIC_0$. The atmospheric inventory is proportional to the atmospheric partial pressure, $pCO_{2,a}$ and, correspondingly, $C_a$ has to be proportional to $pCO_{2,a}$. This leads to the following definition of $C_a$:

$$C_a = \frac{pCO_{2,a}}{pCO_{2,0}} DIC_0, \tag{A1}$$

$C_a$ is specifically defined for our box model and should not be confused with the concentration or mixing ratio of $CO_2$ in the real-world atmosphere. The atmospheric scale height $h_a$ is given by the requirement that volume times concentration equals 25  inventory:

$$h_a = \frac{N_{a,0}}{A \cdot DIC_0}. \tag{A2}$$

The perturbation ($\Delta$) from the initial equilibrium in $pCO_2$, and similarly for other quantities, is defined by: $\Delta pCO_2(t)$= $pCO_2(t)$-$pCO_2(t_0)$. The perturbation in $pCO_{2,o}$ is related to that in DIC by the Revelle factor $\xi$ (Revelle and Suess, 1957):

$$\xi = \frac{\Delta pCO_{2,o}}{pCO_{2,0}} \cdot \frac{DIC_0}{\Delta DIC} \tag{A3}$$

We note that the carbonate chemistry is non-linear and $\xi$ varies with environmental conditions (DIC, alkalinity, temperature, salinity). Assuming a constant Revelle factor $\xi$ is therefore an approximation. In particular, $\xi$ increases with increasing DIC.

As a consequence, the "effective" value of $\xi$ increases with increasing carbon emission and is thus the larger the larger the magnitude of the carbon pulses. This yields for the perturbation in the net air-to-sea carbon flux per unit area, $\Delta f_{a \to o,net}$:

$$\Delta f_{a \to o,net} = k_g \cdot (\Delta pCO_{2,a} - \Delta pCO_{2,o}) = g \cdot (\Delta C_a - \xi \Delta DIC), \tag{A4}$$

$\Delta pCO_{2,a}$ and $\Delta pCO_{2,o}$ were replaced with the help of eq. A1 and A3 to obtain the right-hand side of eq. A4. $k_g$ is the gas transfer rate relative to the partial pressure (Wanninkhof, 1992) and $g$ the rate relative to DIC:

$$g = k_g \cdot \frac{pCO_{2,0}}{DIC_0}. \tag{A5}$$

The IRF, or equivalent the airborne fraction, is defined by:

$$IRF(CO_{2,a}(t)) = \frac{\Delta N_a(t)}{P} = \frac{\Delta N_a(t)}{\Delta N_a(t) + \Delta N_o(t)}, \qquad \text{for } t > t_0 \tag{A6}$$

$P$ is the magnitude of the initial carbon pulse at time $t_0$ and this released carbon is conserved within the model atmosphere and ocean. The net air-to-sea flux becomes zero, when a new equilibrium at time $t_\infty$ is reached; it follows from eq. A1 that

$C_{a,0} = DIC_0$ and from eq. A4 that $\Delta DIC_\infty = \Delta C_{a,\infty}/\xi$. Finally, using this and $\Delta N_a = \Delta C_a \cdot h_a \cdot A$ and $\Delta N_o = \Delta DIC \cdot h_o \cdot A = \Delta C_a/\xi \cdot h_o \cdot A$ yields for the IRF in an ocean-atmosphere system at equilibrium:

$$IRF_\infty(CO_{2,a}) = \frac{h_a}{h_a + \frac{1}{\xi} h_o} = \frac{N_{a,0}}{N_{a,0} + \frac{1}{\xi} N_{o,0}}. \tag{A7}$$

**Distribution of a $\delta^{13}C$ perturbation within the atmosphere-ocean-land biosphere system:** We now turn to the isotope $^{13}C$. The isotopic ratio is given by (see e.g. Mook, 1986)):

$$^{13}R(C) = ^{13}C/^{12}C, \tag{A8}$$

and the isotopic signature in permil by:

$$\delta^{13}C = \left( \frac{^{13}R}{^{13}R_{std}} - 1 \right) \cdot 1000. \tag{A9}$$

$^{13}R_{std}$ is the standard ratio. Isotopic discrimination is described by factors using the symbol $\alpha$ or by additive terms in $\delta$-notation using symbol $\varepsilon$ in permil. It holds:

$$\varepsilon = (\alpha - 1) \cdot 1000. \tag{A10}$$

In the following, we neglect for simplicity the order 1% difference between the concentration of $^{12}C$ and total C (this difference may be accounted for in the definition and values of the isotopic discrimination factors).

We address first the response during the first two millennia after the pulse. In the following paragraphs, we will develop a simple analytical expression for the IRF ($\delta^{13}C_a$) after a new equilibrium for the atmosphere-ocean-land biosphere system is reached. Thus, we neglect carbonate compensation and weathering and burial fluxes. We consider exchange between the atmosphere and ocean, and between the atmosphere and land biosphere reservoirs and assume that these three reservoirs are
well mixed.

*The box model:* The carbon and $^{13}$C budgets for the 3-box model are described by:

$$
\begin{aligned}
\frac{dN_o^i}{dt} &= F_{a\to o}^i - F_{o\to a}^i = F_{a\to o,net}^i \\
\frac{dN_b^i}{dt} &= F_{a\to b}^i - F_{b\to a}^i = F_{a\to b,net}^i \\
\frac{dN_a^i}{dt} &= -F_{a\to o,net}^i - F_{a\to b,net}^i + \delta(t)\cdot P^i
\end{aligned}
\tag{A11}
$$

$N$ is the reservoir size (e.g. in GtC), indices $a$, $o$, and $b$ denote the atmosphere, ocean, and land biosphere box, respectively. Index $i$ refers either to carbon or $^{13}$C. $F$ denotes fluxes between reservoirs. $\delta$ is the Kronecker symbol and $P^i$ the pulse released at time $t = t_0 = 0$ to the atmosphere.

*The initial atmospheric $\delta^{13}C$ perturbation:* The IRF($t$) for $\delta^{13}C_a$ is given by the perturbation in the isotopic signature, $\Delta\delta^{13}C_a(t)$, divided by the initial ($ini$) perturbation, $\Delta\delta^{13}C_a^{ini}$. $\Delta\delta^{13}C_a^{ini}$ is the perturbation immediately after a carbon input
of amount $P$ and with signature $^{13}R_P$ ($\delta^{13}C_P$). Mass balance implies that the amount of $^{13}$C before and after the pulse release is equal:

$$
N_{a,0}\cdot{}^{13}R_{a,0} + P\cdot{}^{13}R_P = (N_{a,0} + P)\cdot{}^{13}R_a^{ini}.
\tag{A12}
$$

We convert the ratios to $\delta$-units and subtract on both sides $(N_{a,0} + P)\cdot\delta^{13}C_{a,0}$ to get the initial perturbation:

$$
\Delta\delta^{13}C_a^{ini} = \frac{P}{N_{a,0}+P}\cdot(\delta^{13}C_P - \delta^{13}C_{a,0}).
\tag{A13}
$$

$\Delta\delta^{13}C_a^{ini}$ is proportional to the difference between the signature of the pulse and the initial atmospheric signature. Thus, the initial perturbation is relative to the atmospheric $\delta^{13}C$ signature.

*The relationship between $\delta^{13}C_a$, $\delta^{13}C_o$, and $\delta^{13}C_b$ at equilibrium:* In this subsection, relationships between the isotopic signatures are developed by considering equilibrium between the three boxes. This will allow us to simplify the isotopic mass balance equations further below .

The gross air-to-sea flux per unit area is:

$$
{}^{13}f_{a\to o} = {}^{13}k_g\cdot{}^{13}\text{pCO}_{2,a} = k_g\cdot\text{pCO}_{2,a}\cdot{}^{13}\alpha_{a\to o}\cdot{}^{13}R(\text{pCO}_{2,a}),
\tag{A14}
$$

where $^{13}\alpha_{a\to o} = {}^{13}k_g/k_g$ is the discrimination factor for the gross air-to-sea transfer. The gross sea-to-air flux is:

$$
\begin{aligned}
{}^{13}f_{o\to a} = {}^{13}k_g\cdot{}^{13}\text{pCO}_{2,o} &= k_g\cdot\text{pCO}_{2,o}\left(\frac{{}^{13}k_g}{k_g}\frac{{}^{13}R(\text{pCO}_{2,o})}{{}^{13}R(\text{DIC})}\right)\cdot{}^{13}R(\text{DIC}) \\
&= k_g\cdot\text{pCO}_{2,o}\cdot{}^{13}\alpha_{o\to a}\cdot{}^{13}R(\text{DIC}).
\end{aligned}
\tag{A15}
$$

The term in parentheses is the the discrimination factor for the transfer of carbon from the DIC pool to the atmosphere, $^{13}\alpha_{o \to a}$. At equilibrium ($eq$), the two gross fluxes cancel each other. It follows with $pCO_{2,a,eq} = pCO_{2,o,eq}$ and $^{13}pCO_{2,a,eq} = {}^{13}pCO_{2,o,eq}$:

$$^{13}R_{eq}(pCO_{2,a}) = \frac{^{13}\alpha_{o \to a}}{^{13}\alpha_{a \to o}} \cdot {}^{13}R_{eq}(\text{DIC}) = \frac{1}{\alpha_{a,o}} \cdot {}^{13}R_{eq}(\text{DIC}). \tag{A16}$$

5 This equation gives the equilibrium relationship between the atmospheric and oceanic signature. $\alpha_{a,o}$ denotes the equilibrium discrimination factor between $pCO_{2,a}$ and DIC. $\alpha_{a,o}$ depends on temperature and somewhat on carbonate chemistry (Mook, 1986) and is about 1.008 and thus close to 1.

A similar relationship is readily developed for the isotopic ratio of the total carbon in the land biosphere. The gross fluxes between the land and atmosphere are:

$$
\begin{aligned}
^{13}F_{a \to b} &= F_{a \to b} \cdot {}^{13}\alpha_{a \to b} \cdot {}^{13}R(pCO_{2,a}) \\
^{13}F_{b \to a} &= F_{b \to a} \cdot {}^{13}\alpha_{b \to a} \cdot {}^{13}R(N_b).
\end{aligned} \tag{A17}
$$

This yields at equilibrium:

$$^{13}R_{eq}(pCO_{2,a}) = \frac{1}{\alpha_{a,b}} \cdot {}^{13}R_{eq}(N_b) \tag{A18}$$

Eqs. A16 and A18 are readily converted into $\delta$ notation:

$$
\begin{aligned}
\delta^{13}C_{o,eq} &= \left(1 + \frac{\varepsilon_{a,o}}{1000}\right)\delta^{13}C_{a,eq} + \varepsilon_{a,o} \cong \delta^{13}C_{a,eq} + \varepsilon_{a,o}. \\
\delta^{13}C_{b,eq} &= \left(1 + \frac{\varepsilon_{a,b}}{1000}\right)\delta^{13}C_{a,eq} + \varepsilon_{a,b} \cong \delta^{13}C_{a,eq} + \varepsilon_{a,b}.
\end{aligned} \tag{A19}
$$

Eqs. A16, A18 and A19, allow us to express the isotopic signature of the land biosphere and the ocean by the isotopic signature of the atmosphere at equilibrium. These equations hold both for the initial equilibrium ($t_0$) before the pulse release and when a new equilibrium is reached at $t_\infty$. In practice $t_\infty$ is about 2 kyr. It follows immediately from eq. A19 that the 20 isotopic signatures at the new equilibrium are about equal for the three reservoirs:

$$\Delta\delta^{13}C_\infty = \Delta\delta^{13}C_{a,\infty} \cong \Delta\delta^{13}C_{o,\infty} \cong \Delta\delta^{13}C_{b,\infty}. \tag{A20}$$

In other words, the isotopic signal is mixed uniformly through the atmosphere, ocean, and land biosphere when assuming that these three reservoirs are well mixed as in this appendix section.

*Solving the mass balance equations of the isotopic pertubation for $\Delta\delta^{13}C$:* The mass balance equations for the ocean-land 25 biosphere-atmosphere system are given for carbon by:

$$
\begin{aligned}
N_{a,0} + N_{o,0} + N_{b,0} + P &= N_{a,\infty} + N_{o,\infty} + N_{b,\infty}, &&\text{for } t > t_0 \\
P &= \Delta N_a + \Delta N_o + \Delta N_b.
\end{aligned} \tag{A21}
$$

The second equation describes the perturbation only. We use eqs. A16 and A18 to replace $N_o \cdot {}^{13}R_o$ with $\alpha_{a,o} \cdot N_o \cdot {}^{13}R_a$ and similar for the land reservoir and write the mass balance for ${}^{13}$C:

$$(N_{a,0} + \alpha_{a,o} \cdot N_{o,0} + \alpha_{a,b} \cdot N_{b,0}) \cdot {}^{13}R_{a,0} + P \cdot {}^{13}R_P \qquad \text{for } t > t_0$$

$$= (N_{a,\infty} + \alpha_{a,o} \cdot N_{o,\infty} + \alpha_{a,b} \cdot N_{b,\infty}) \cdot {}^{13}R_{a,\infty}. \tag{A22}$$

Now we convert to $\delta$ units:

$$(N_{a,0} + \alpha_{a,o} \cdot N_{o,0} + \alpha_{a,b} \cdot N_{b,0} + P) \cdot 1000 + (N_{a,0} + \alpha_{a,o} \cdot N_{o,0} + \alpha_{a,b} \cdot N_{b,0}) \cdot \delta^{13}C_{a,0} + P \cdot \delta^{13}C_P$$

$$= (N_{a,\infty} + \alpha_{a,o} \cdot N_{o,\infty} + \alpha_{a,b} \cdot N_{b,\infty}) \cdot 1000 + (N_{a,\infty} + \alpha_{a,o} \cdot N_{o,\infty} + \alpha_{a,b} \cdot N_{b,\infty}) \cdot \delta^{13}C_{a,\infty}. \tag{A23}$$

We subtract on both side the mass balance equation for carbon (eq. A21) multiplied by factor 1000. Note that $\alpha_{a,o}$ and $\alpha_{a,b}$ are present in the carbon terms in eq. A23, but not in the mass balance for carbon (eq. A21). This difference results in the first

two terms in eq. A24 below.

$$-\Delta N_{o,\infty} \cdot \overbrace{(\alpha_{a,o} - 1) \cdot 1000}^{\delta^{13}C_{DIC,0} - \delta^{13}C_{a,0}} \quad -\Delta N_{b,\infty} \cdot \overbrace{(\alpha_{a,b} - 1) \cdot 1000}^{\delta^{13}C_{b,0} - \delta^{13}C_{a,0}}$$

$$+(N_{a,0} + \alpha_{a,o} \cdot N_{o,0} + \alpha_{a,b} \cdot N_{b,0}) \cdot \delta^{13}C_{a,0} + P \cdot \delta^{13}C_P$$

$$= (N_{a,\infty} + \alpha_{a,o} \cdot N_{o,\infty} + \alpha_{a,b} \cdot N_{b,\infty}) \cdot \delta^{13}C_{a,\infty}. \tag{A24}$$

Next, we subtract on both sides the third left hand side term, $(N_{a,0} + \alpha_{a,o} \cdot N_{o,0} + \alpha_{a,b} \cdot N_{b,0}) \cdot \delta^{13}C_{a,0}$, as well as the term

$P \cdot \delta^{13}C_{a,0}$. We rearrange and use again the mass balance for carbon to get:

$$-\Delta N_{o,\infty} \cdot (\delta^{13}C_{DIC,0} - \delta^{13}C_{a,0} \cdot (1 + \varepsilon_{a,o}/1000))$$

$$-\Delta N_{b,\infty} \cdot (\delta^{13}C_{b,0} - \delta^{13}C_{a,0} \cdot (1 + \varepsilon_{a,b}/1000)) - P \cdot (\delta^{13}C_a - \delta^{13}C_P)$$

$$= (N_{a,0} + N_{o,0} \cdot (1 + \varepsilon_{a,o}/1000) + N_{b,0} \cdot (1 + \varepsilon_{a,b}/1000) + P) \cdot \Delta\delta^{13}C_{a,\infty}$$

$$\tag{A25}$$

The terms with $\varepsilon/1000$ represent a small correction ($< 0.02$) and can be safely neglected. We replace $P$ with $\Delta N_{a,\infty} + \Delta N_{o,\infty} + \Delta N_{b,\infty}$ in the above equation and exchange the left and right side to get the following mass balance for the $\delta^{13}$C perturbation:

$$(N_{a,0} + N_{o,0} + N_{b,0} + P) \cdot \Delta\delta^{13}C_{a,\infty}$$

$$= \Delta N_{a,\infty} \cdot (\delta^{13}C_P - \delta^{13}C_{a,0}) + \Delta N_{o,\infty} \cdot (\delta^{13}C_P - \delta^{13}C_{DIC,0}) + \Delta N_{b,\infty} \cdot (\delta^{13}C_P - \delta^{13}C_{b,0})$$

$$= P \cdot (\delta^{13}C_P - \delta^{13}C_{mean}), \tag{A26}$$

with:

$$\delta^{13}C_{mean} = \frac{\Delta N_{a,\infty}}{P} \cdot \delta^{13}C_{a,0} + \frac{\Delta N_{o,\infty}}{P} \cdot \delta^{13}C_{o,0} + \frac{\Delta N_{b,\infty}}{P} \cdot \delta^{13}C_{b,0} \tag{A27}$$

Equation A26 represents the mass balance for the isotopic perturbation and its interpretation is as follows. The term on the left hand side equals the carbon mass in the system times the perturbation in the signature and therefore equals the total isotopic

perturbation in GtC ‰. In equilibrium, the $\delta^{13}$C perturbation, $\Delta\delta^{13}$C$_\infty$, is well mixed (eq. A20) within the total ocean-atmosphere carbon reservoir $(N_{a,0} + N_{o,0} + N_{b,0} + P)$. The terms after the first equal sign provide a different view of the total isotopic perturbation. In this picture, a fraction $\Delta N_{l,\infty}$ of the pulse is added to each reservoir (index $l$) with the perturbation in signature given by $(\delta^{13}$C$_P - \delta^{13}$C$_l)$. This is then summarized after the second equal sign by the product of the carbon input $P$ times the isotopic perturbation of the pulse relative to the mean signature of the reservoirs absorbing this additional carbon, $\delta^{13}$C$_P - \delta^{13}$C$_{mean}$.

By rearranging eq. A26 we get the perturbation in the isotopic signature of the atmosphere-ocean-land system:

$$\Delta\delta^{13}C_\infty = \frac{P}{N_{a,0} + N_{o,0} + N_{b,0} + P} \cdot \left(\delta^{13}C_P - \delta^{13}C_{mean}\right). \tag{A28}$$

In other words, the perturbation in the signature scales with the carbon mass ratio of the pulse input to the total mass in the system. This perturbation is also proportional to the difference between the signature of the pulse and the "mean" signature of the system as defined by eq. A27.

*IRF($\delta^{13}$C$_a$) at equilibrium (t $\sim$ 2 kyr):* Finally, we use the definition for the IRF and eqs. A13 and A28 to obtain the following expression for the IRF at a new equilibrium for the ocean-atmosphere-land biosphere system:

$$\begin{aligned} \text{IRF}_\infty(\delta^{13}C_a) &= \frac{\Delta\delta^{13}C_{a,\infty}}{\Delta\delta^{13}C_a^{ini}} \\[2mm] &= \frac{N_{a,0} + P}{N_{a,0} + N_{o,0} + N_{b,0} + P} \cdot \frac{(\delta^{13}C_P - \delta^{13}C_{mean})}{(\delta^{13}C_P - \delta^{13}C_{a,0})} \end{aligned} \tag{A29}$$

The fraction of the initial perturbation in the atmosphere that remains airborne corresponds roughly to the ratio of the carbon in the atmosphere immediately after the pulse release to the total carbon in the system. The second fraction on the right side modifies this ratio by about +20 % for typical negative isotopic pulse perturbation applied in this study, because the mean signature is slightly heavier than the atmospheric signature. The nominator of this modifier reflects that the pulse perturbation is distributed within the system at the new equilibrium, while the denominator reflects that the pulse is initially added to the atmosphere.

**Ocean invasion: numerical examples:** Equation A7 and eq. A29 provide simple expressions to approximate the IRF for the perturbation in atmospheric $CO_2$ and its istopic signature after a pulse-like carbon input. These equations illustrate the fundamental difference in the IRF for $CO_{2,a}$ and for $\delta^{13}$C$_a$. To ease interpretation, we neglect the land biosphere for the moment. Then, in the limit of $P \to 0$, eq. A7 and eq. A29 are formally identical for carbon and $^{13}$C (index: $i$):

$$IRF^i \overset{P\to 0}{=} \frac{N_{a,0}}{N_{a,0} + \frac{1}{\xi^i}N_{o,0}} \cdot M^i \tag{A30}$$

However, for $\delta^{13}$C$_a$ a Revelle Factor of 1 instead of about 12 applies, while the modifier $M$ is exactly 1 for $CO_{2,a}$ and around 1 for $\delta^{13}$C$_a$. The perturbation in the isotopic signature is diluted by the entire carbon inventory in the system, whereas the "available" ocean inventory is reduced by the Revelle Factor for the dilution of an atmospheric $CO_2$ pertubation.

Numerically, the preindustrial carbon inventory is 600 GtC in the atmosphere and 37,400 GtC in the ocean. These values are equivalent to a scale height of 69 m for the atmosphere and of 4,300 m for the ocean, when assuming a preindustrial (surface)

concentration of DIC of 24 g/m$^3$ and an ocean area of $3.62 \cdot 10^{14}$ m$^2$. The buffer factor is about 12 for small perturbations. This yields for IRF$_\infty$(CO$_{2,a}$) a value of about 0.16. In other words, a fraction of about 16 % of a (small) carbon input into the atmosphere is still airborne after about 1 to 2 kyr, when the ocean and atmosphere have approached a new equilibrium. The observed penetration of CFCs and bomb-produced radiocarbon suggests that an atmospheric perturbation penetrates about

the top 300 m of the ocean within a decade. Thus, we expect that about 70 % of the initial perturbation are still found in the atmosphere after a decade. These numbers are comparable in magnitude to the IRF of CO$_2$ shown in Fig. 1a for 10 years and 2 kyr after the pulse input. The buffer factor increases with the magnitude of the perturbation in DIC and pCO$_2$ and thus with pulse size $P$. Hence, IRF increases with increasing pulse size, as again shown in Fig. 1a. We note that any carbon uptake or release by the land biosphere is not taken into account in eq. A7 and in this discussion, in contrast to the results discussed in

the main text.

To determine the IRF for $\delta^{13}$C$_a$, we take into account that the isotopic signal is also entering the land biosphere and assume a total inventory in the atmosphere-ocean–land system of 40,000 GtC. This yields an IRF$_\infty$ of 1 % to 3.2 % for P varying between -250 to +500 GtC and of about 18 % for P equal 5,000 GtC and using $\delta^{13}$C$_{mean}$=4 ‰ (calculated with eq. A27). These values are in agreement with the estimates shown in Fig. 1b for year 2,000. Assuming that the isotopic perturbation has

mixed within a layer of 300 m in the ocean ($\sim$2,600 GtC) and with the living vegetation on land (500 GtC) yiels a "decadal" IRF of about 12-31 % for small pulse sizes (-250 GtC to 500 GtC), again in agreement with the Bern3D-LPX results.

**Carbonate compensation and terrestrial neutralization of the CO$_2$ perturbation:** Next, we address CaCO$_3$ compensation of the carbon added by the pulse. We derive an expression for the IRF(CO$_{2,a}$) at the time when CaCO$_3$ compensation of the pulse is completed - about 10-20 kyr after the pulse input. The following calculations are based on Archer et al. (1998). As

above, we assume the ocean to consist of a single, well-mixed box which is, after CaCO$_3$ compensation, again in equilibrium with the atmosphere. Concentrations of CO$_2$, HCO$_3^-$, and CO$_3^{2-}$ in seawater are related by

$$\frac{\text{pCO}_{2,o} \cdot \left[\text{CO}_3^{2-}\right]}{\left[\text{HCO}_3^-\right]} = \frac{K_2' \cdot K_H}{K_1'} = const. \tag{A31}$$

$K_1'$ and $K_2'$ are the apparent dissociation constants for carbonic acid and $K_H$ is the Henry's Law solubility product for CO$_2$. For simplicity, we assume the ratio of $K_1'$, $K_2'$, and $K_H$, which are temperature and salinity dependent, to be constant over the

course of the experiments. With above assumption and by taking the ratio of initial and final states and by rearranging we get

$$\frac{\text{pCO}_{2,\infty}}{\text{pCO}_{2,0}} \approx \left(\frac{\left[\text{HCO}_{3,\infty}^-\right]}{\left[\text{HCO}_{3,0}^-\right]}\right)^2 \cdot \frac{\left[\text{CO}_{3,0}^{2-}\right]}{\left[\text{CO}_{3,\infty}^{2-}\right]}. \tag{A32}$$

Sedimentary dissolution of CaCO$_3$ in response to the carbon pulse tends to keep the carbonate ion concentration constant over the course of the simulations, yielding

$$\frac{\text{pCO}_{2,\infty}}{\text{pCO}_{2,0}} \approx \left(\frac{\left[\text{HCO}_{3,\infty}^-\right]}{\left[\text{HCO}_{3,0}^-\right]}\right)^2. \tag{A33}$$

$HCO_3^-$ is by far the dominant form of DIC. This allows one to approximate $HCO_3^-$ by DIC in eq. A33 to get:

$$\frac{pCO_{2,\infty}}{pCO_{2,0}} \approx \left(\frac{DIC_{o,0} + \Delta DIC_o}{DIC_{o,0}}\right)^2 = \left(\frac{N_{o,0} + \Delta N_o}{N_{o,0}}\right)^2. \tag{A34}$$

$\Delta$ refers here to the difference between the final and initial state. With pulse sizes substantially smaller than the oceanic DIC inventory, $\Delta N_o \ll N_{o,0}$, we approximate

$$\frac{pCO_{2,\infty}}{pCO_{2,0}} \approx 1 + 2 \cdot \frac{\Delta N_o}{N_{o,0}}. \tag{A35}$$

From Fig. 3a we see that on multi-millennial timescales the majority of the pulse is taken up by the ocean. Further, $CaCO_3$ compensation and weathering-burial imbalances in the $CaCO_3$ cycle (terrestrial neutralization) add (or remove) additional carbon to (from) the ocean. We thus assume $\Delta N_o \approx 1.5 \cdot P$; $P$ is the pulse size in $GtC$. The factor 1.5 is model dependent and slightly different here from the value provided by Archer et al. (1998) or in previous versions of the Bern3D model. This yields

for the relative increase in $pCO_2$:

$$\frac{\Delta pCO_2}{pCO_{2,0}} \approx 3 \cdot \frac{P}{N_{o,0}}. \tag{A36}$$

We recall the definition of the airborne fraction:

$$IRF(CO_{2,a}) = 2.12\,GtC\,\mu atm^{-1} \cdot \frac{\Delta pCO_{2,a}}{P}. \tag{A37}$$

Finally, with eq. A36 and A37 we express the IRF at the time when $CaCO_3$ compensation is completed by

$$IRF_{CaCO_3}(CO_{2,a}) = 3 \cdot 2.12\,GtC\,\mu atm^{-1} \cdot \frac{pCO_{2,0}}{N_{o,0}}. \tag{A38}$$

The remaining atmospheric fraction is thus independent of the pulse size for $P \ll N_{o,0}$. Its magnitude results directly from the carbonate chemistry (Archer et al., 1998). With a pre-industrial ocean DIC inventory of 37,400 $GtC$ and surface ocean $pCO_2$ of 284 $\mu atm$ we get a remaining airborne perturbation of ~5 %. This is in agreement with results from the Bern3D-LPX model after about 10-20 $kyr$ (Fig. 1a).

**Carbonate compensation of the $\delta^{13}C$ perturbation:** Carbonate compensation has almost a negligible influence on the $\delta^{13}C$ perturbation. The carbonate compensation process acts, aside from small isotopic discrimination, equally on the different isotopes. Therefore, $^{12}CO_{2,a}$ and $^{13}CO_{2,a}$ both decrease according to eq. A38, leaving the isotopic ratio in first order approximation unchanged. Additional effects are, here formulated for a positive pulse, as follows. The dissolution of $CaCO_3$ adds an isotopic perturbation to the ocean-atmosphere-land system. The amount of carbon added corresponds to about half of the

pulse size, $P$. $CaCO_3$ has a $\delta^{13}C$ signature of about +2.9 ‰ compared to -24 ‰ of the pulse. We reference the $CaCO_3$ isotopic perturbation relative to the mean signature of the ocean in the model (~0.8 ‰ ) as this additional carbon remains mainly in the ocean, while the mean signature that applies for the pulse is according to eq. A27 about -3 ‰ for pulse sizes between -250 $GtC$ and 500 $GtC$. Thus the isotopic perturbation by $CaCO_3$ dissolution is about 20 times smaller than that of the pulse $(0.5 \times P \times$ (2.9 ‰ - 0.8 ‰ ) : $P \times (-24$ ‰ + 3 ‰ ) = -1:20). The dissolution of $CaCO_3$ enlarges the carbon reservoir by

which the initial atmospheric $\delta^{13}C$ is diluted. However, the amount of added carbon is, except for the 5,000 GtC pulse, small compared the total inventory of the system of about 40,000 GtC. In summary, the influence of $CaCO_3$ compensation on $\delta^{13}C_a$ is small and eq. A29 is still approximately valid for the time scale of $CaCO_3$ compensation.

**Removal of the remaining $CO_2$ perturbation by silicate weathering:** Finally, on timescales of hundreds of thousands

of years imbalances in the weathering and burial fluxes remove any remaining perturbation in carbon from the atmosphere-ocean-land system. Silicate rock weathering leads to a net removal of carbon from the atmosphere by the following simplified reaction (see Colbourn et al., 2013, for more details):

$$2CO_{2(aq)} + H_2O_{(l)} + CaSiO_{3,(s)} \rightarrow Ca^{2+}_{(aq)} + 2HCO^-_{3(aq)} + SiO_{2(aq)}. \tag{A39}$$

The timescale for this silicate weathering feedback has been determined to be on the order of 240 kyr (Colbourn et al., 2015)

and 270 kyr (Lord et al., 2016), however with a large spread. Silicate rock weathering removes the remaining atmospheric perturbation ($\sim$8 %). Further, the additional alkalinity added to the ocean deepens the saturation horizon, resulting in higher carbon burial in marine sediments and thus also removes the DIC perturbation (see Fig. 3a).

**Removal of the remaining $\delta^{13}C$ perturbation by burial of organic matter and $CaCO_3$:** In the case of $\delta^{13}C$, silicate rock weathering has no direct effect on the $\delta^{13}C$ signature. The perturbation is removed on these long timescales by the replacement

of carbon through burial of organic matter and $CaCO_3$ and weathering inputs:

$$\frac{d}{dt}(N_S \cdot \Delta\delta^{13}C_S) = -F_{\text{burial}} \cdot \Delta\delta^{13}C_{burial}, \tag{A40}$$

with $N_S$ and $\Delta\delta^{13}C_S$ equal the total mass and mean isotopic perturbation in the atmosphere-ocean-land system and $F_{burial}$ the burial flux of carbon leaving the system and $\Delta\delta^{13}C_{burial}$ its isotopic signature. In simulations with constant weathering rates, the residence time of carbon in the atmosphere-ocean-land system is on the order of $\tau = N/F_{burial} \sim$ 90 kyr (burial/weathering

input is 0.46 GtC yr$^{-1}$, total amount: $\sim$ 40,000 GtC)). Surprisingly, in the factorial simulation with sediments and constant weathering fluxes ($SED_{500}$), the perturbation is removed much faster than expected from the mean residence time (Fig. 1b, dotted line), because the isotopic perturbation of the burial flux is larger than that of the ocean-atmosphere-land system.

The long-term removal rate of the $\delta^{13}C$ perturbation is sensitive to $\Delta\delta^{13}C$ gradients in the ocean. The isotopic signature of the burial flux is mainly determined by the upper ocean signature. Changes in the gross exchange fluxes of carbon between the

atmosphere and the ocean in response to perturbed $CO_2$ and changes in the isotopic fractionation of air-sea fluxes in response to perturbed temperatures increase the $\Delta\delta^{13}C$ perturbation in the upper ocean relative to the deep (Fig. A1). In case of enabled weathering feedbacks, the additional flux of isotopically heavy carbon from excess weathering partly mitigates surface-to-deep $\Delta\delta^{13}C$ gradients. In turn, the perturbation is removed faster in simulations with constant weathering compared to simulations with enabled weathering feedbacks. Taken together, this highlights the role of spatial patterns of the $\delta^{13}C$ perturbation even on

these very long timescales.

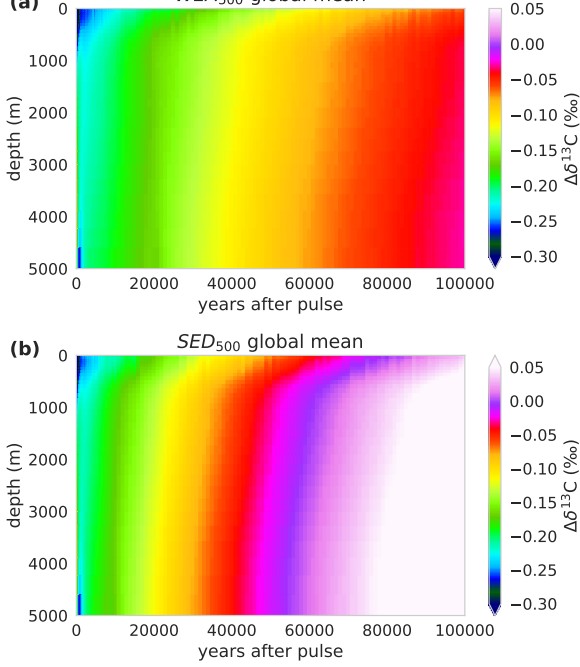

**Figure A1.** Hovmoeller-type diagram showing the temporal evolution of the global-mean vertical profile of $\Delta\delta^{13}C_{DIC}$ for experiment (a) $WEA_{500}$ and (b) $SED_{500}$.

## Appendix B: Sediment module and weathering feedback

**Parameterization of weathering feedbacks on land:** We use the 0-D version of the Rock Geochemical Model v0.9 (Colbourn et al., 2013). Weathering of $CaCO_3$ and $CaSiO_3$ are parameterized as functions of temperature ($T$), runoff ($R$) and productivity ($P$) in the following form for $CaCO_3$:

$$F_{CaCO_3} = F_{CaCO_3,0}(1 + k_{Ca}(T - T_0))\frac{R}{R_0}\frac{P}{P_0} \tag{B1}$$

and for $CaSiO_3$:

$$F_{CaSiO_3} = F_{CaSiO_3,0} \cdot e^{\frac{1000 E_0}{RT_0^2}(T-T_0)} \cdot \left(\frac{R}{R_0}\right)^{\beta}\frac{P}{P_0}. \tag{B2}$$

indicates the initial value diagnosed prior to the pulse, $k_{Ca}$ stems from a correlation between temperature and $HCO_3^-$ ion concentration of groundwater with a value of 0.049, $E_a$ is the activation energy for silicate weathering (63 kJ mol$^{-1}$), $\beta$ is taken as 0.65, and $R$ indicates run-off and is parameterized as a function of temperature:

$$\frac{R}{R_0} = \max\{0, 1 + k_{run}(T - T_0)\} \tag{B3}$$

**Table B1.** Parameters used for the sediment module, similar to what is shown in the appendix A of Tschumi et al. (2011) and with updates from Roth et al. (2014) and Jeltsch-Thömmes et al. (2019).

| Parameter | Value | Unit | Reference |
|---|---|---|---|
| Number of time steps per year | 96 | - | Roth et al. (2014) |
| Density of solid sediment components | 2.6 | $g\,cm^{-3}$ | Heinze et al. (1999) |
| Molar weight of POM | 32.74 | $g\,mol^{-1}$ | Heinze et al. (1999) |
| Molar weight of $CaCO_3$ | 100.0 | $g\,mol^{-1}$ | Heinze et al. (1999) |
| Molar weight of opal | 67.2 | $g\,mol^{-1}$ | Heinze et al. (1999) |
| Molar weight of clay | 430.51 | $g\,mol^{-1}$ | Heinze et al. (1999) |
| Terrestrial clay flux | 0.8 | $g\,m^{-2}yr^{-1}$ | Roth et al. (2014) |
| Rate constant for $CaCO_3$ redissolution | 800 | $l\,mol^{-1}yr^{-1}$ | Jeltsch-Thömmes et al. (2019) |
| Rate constant for opal redissolution | 20 | $l\,mol^{-1}yr^{-1}$ | Tschumi et al. (2011) |
| Rate constant for oxydation | 100 | $l\,mol^{-1}yr^{-1}$ | Roth et al. (2014) |
| Saturation concentration for pore water silicic acid | 800 | $\mu mol\,l^{-1}$ | Heinze et al. (1999) |
| Pore water diffusion coefficient | $5.5\times10^{-6}$ | $cm^2s^{-1}$ | Jeltsch-Thömmes et al. (2019) |
| Bioturbation diffusion coefficient | $15\times10^{-3}$ | $cm^2yr^{-1}$ | Heinze et al. (1999) |
| Redfield coefficient for phosphate | 1 | - | Anderson and Sarmiento (1994) |
| Redfield coefficient for oxygen | -170 | - | Anderson and Sarmiento (1994) |
| Redfield coefficient for nitrate | 16 | - | Anderson and Sarmiento (1994) |
| Redfield coefficient for carbon | 117 | - | Anderson and Sarmiento (1994) |
| Redfield coefficient for alkalinity | -17 | - | Roth et al. (2014) |

with $k_{run}$ = 0.025. In the case of experiments $4box_{500}$ and $LGM_{500}$, LPX-Bern was not coupled and instead productivity was parameterized, again following equations given in Colbourn et al. (2013), as:

$$\frac{P}{P_0} = \left( \frac{2\frac{C}{C_0}}{1+\frac{C}{C_0}} \right)^{0.4} \tag{B4}$$

with $C$ being atmospheric $CO_2$ in ppm. Reference values (0) for temperature, runoff, and productivity are diagnosed in the 100 yr prior to the pulse. For further details and discussion of parameter value choices, the reader is referred to Colbourn et al. (2013).

*Competing interests.* The authors declare that they have no conflict of interest.

*Acknowledgements.* This study was supported by the Swiss National Science Foundation (# 200020_172476) and the Oeschger Centre for Climate Change Research. We thank S. Lienert, Daniel Baggenstos and James Menking for helpful comments on the manuscript.

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
