# Peer review of "Modeling the evolution of pulse-like perturbations in atmospheric carbon and carbon isotopes: the role of weathering-sedimentation imbalances"

_Climate of the Past, 2019_

## Referee Comment (RC1) · Anonymous Referee #1 · 4 Sep 2019

This paper describes the long-term evolution of $CO_2$-pulse experiments in the Bern3D EMIC on $CO_2$, $\delta^{13}$C, $\Delta^{14}$C. For this aim, especially the airborne fraction of these three variables are analysed, e.g. by a fit of a sum of 5 exponential functions. Furthermore, principal components of EOF are used to investigate the distribution of the signal in the ocean.

The main target is the understanding of atmospheric $\delta^{13}$C of $CO_2$ as is so far measured in ice cores covering the last 150 kyr. There, an offset in $\delta^{13}$C between the penultimate and last glacial maximum (PGM, LGM) still lacks an explanation (Schneider et al., 2013; Eggleston et al., 2016), but also $\delta^{13}$C during millennial-scale dynamics are also

briefly discussed (Bauska et al., 2016, 2018).

The paper is on a good way. Its focus is on the evolution of the isotopic signatures following a carbon pulse are to my knowledge new, while the long-term evolution of $CO_2$ pulses has already been analysed elsewhere (Colbourn et al., 2015; Lord et al., 2016), although with a different model. However, I still have a list of (partly major) points, in which I suggest the draft is revised in order to clarify open issues or sharpen its message. They are in detail:

1. The main aim of this study is to better understand the observed evolution of $\delta^{13}$C in the Earth system (as such explicitly written at the beginning of section 3) with special focus on the atmosphere. I therefore suggest to sharpen (and shorten) the introduction focusing on the available atmospheric $\delta^{13}$C data (Schneider et al., 2013; Eggleston et al., 2016; Bauska et al., 2016, 2018), and delete the lengthly citations/discussion of $CO_2$ over the last 800 kyr. Maybe, also add a figure, which shows the relevant $\delta^{13}$C data, both on glacial/interglacial and millennial-time scales, to give the reader an idea about the magnitudes of changes and the unsolved problems.

2. At the end of the abstract (and maybe once in the discussion) volcanism is mentioned as a likely cause for the $\delta^{13}$C offset between PGM and LGM. However, I believe no details on volcanism as assumed in the model are given. Is there a (constant?) volcanic $CO_2$ flux and what is its $\delta^{13}$C signature? This final sentence of the introduction should be backed up with a more in-depth discussion, how this conclusion has been drawn. Right now on page 20 all examples given seemed to be not enough to explain the data, so an imbalance of weathering, volcanic, and burial fluxes seems to be the only possible solution, however the scenario mentioned to be the most likely one is not been investigated in detail. Maybe your insights on $\delta^{13}$C can guide you to prescribe necessary changes in boundary conditions in such a way that the simulated $\delta^{13}$C explains the offset between

PGM and LGM (e.g. how needs volcanism change (in terms of its $\delta^{13}$C signature and/or in its $CO_2$ release?) to generate the observed offset). This would be a real breakthrough.

3. My understanding of the results here and of Lord et al. (2016) is, that the pulse size (labeled $P$ here and $E$ in the other paper) is also important for the time-dependent airborne-fractions. At least, this is the case for $CO_2$. For that reason $E$ has been included in the sum of 5 exponential functions that fit to the model results in Lord et al. (2016) (their Eq 3), but $E$ is missing in the fit used here (Eq 4). I believe this needs to be revised. I acknowledge that there is a subsection on the role of pulse size on page 14, but I am wondering why this is not included in the fitting equations. Maybe also compare airborne fraction of $CO_2$ with results shown in Colbourn et al. (2015); Lord et al. (2016). Any stricking difference?

4. While Lord et al. (2016) and Colbourn et al. (2015) had only future emission in mind (and therefore the starting point has always been the modern carbon cycle), here changs in the past are in focus. One section is missing, which discusses, if (and at best how) everything said here depends on the background state of the carbon cycle. Especially, these detailed changes in $\delta^{13}$C discussed (PGM vs LGM; millennial-scale) happened during more glacial conditions. I understand all pulse experiments have been analysed starting from pre-industrial conditions. At best, everything needs to be repeated from LGM background, but saying that, I realize, that this might double all efforts, and might be too much for the paper at hand. However, it should be shown with at least one perturbation experiment, how things are different when starting from glacial background, in order to give the reader a feeling of the size of such a potential background dependency.

5. Results (Table 2): I would expect the fit parameters also for $\Delta^{14}$C, and for all scenarios and not only for p500. Maybe they can also generalized by the consideration of $E$ in the fitting functions, see point 3 above.

6. When explaining why the airborne-fraction of $\delta^{13}$C sinks at first more rapidly than that of $CO_2$, I believe this can be explained with the gross gas exchange (atmosphere-ocean), while for the $CO_2$ the net gas exchange (net oceanic uptake) is relevant, which is, as correctly written, slowed down by the carbonate chemistry (buffer or Revelle factor).

7. Eq 5: Without saying that Eq 5 is wrong, I was intuitively expecting that the pulse size $P$ might be relevant here.

8. I find the detailed discussion of C budget changes in GtC over time for one specific scenario p250 (page 13-14) not that interesting. It would be better, this is expressed in fractional changes (eg expressed as airborne fraction for the atmosphere), and the paper would especially benefit if it can be generalize from one scenario to more (all?), e.g. maybe by including the dependency on the pulse size.

9. Page 14, last sentence (on radiocarbon) does not make sense to me, e.g. the first part seems to say the opposite of the second part, maybe extend with details or revise.

10. Discussion of millennial-scale changes (page 20, line 29ff): My reading of the papers of Bauska et al did not come to the conclusions, that they argue that the changes in $CO_2$ by 10 ppm and a decrease in $\delta^{13}$C by 0.2‰ can be solely explained by carbon release from terrestrial sources, but they suggest alternative processes. Please revise this discussion of Fig 6 carefully.

11. Fig 1 has in some scenarios some abrupt (and unexpected) changes around 20k in all three variables. I believe, this was shortly mentioned, but I do not think it is entirely explained, especially not for the isotopes.

12. Fig 5: The colorbar on the right hand side is from -30 to +30‰ in changes in $\delta^{13}$C. I hope there is a typo and this is wrong by some order of magnitude, otherwise it

implies that on the local scale the reconstructions are complete off the target and not useful.

Technical issues:

1. Subsection 2.4 Results is empty. Probably this is just the start of section 3 and Discussion is section 4.

2. Fig 2: Hatched area is "sed + wea" in the legend, but "lithosphere" in the caption. Please combine both somehow (e.g. merge to the same). Does this also include volcanism?

3. Fig 3: grey line is called "data", but this is "modeled".

4. Fig 5: Please add in the caption the time units when describing the subfigures ("1 (c,d)" into '"1000 yr (c,d)" and "10 (e,f)" into "10 kyrs (e,f)" and "50 (g,h) kyrs" into "50 kyrs (g,h)".

5. page 11, line 1, change "fast" into "faster"

6. page 12, line 6. "additional addition" is a bit too much adding, delete one.

7. page 12, line 19, "long-term removal time scale " of what?

8. Throughout: I believe figure should be ordered by when they are refered to, but on page 2 a reference to Fig 5g shows up before any reference to Figs. 2-4.

9. Appendix: Maybe consider replacing "$->$" in you subscripts by "2" or "$\rightarrow$" (command rightarrow in LaTeX), e.g. in Eq. A17: change $\alpha_{a->b}$ into $\alpha_{a2b}$ or $\alpha_{a\rightarrow b}$.

10. Reference list needs careful revision, since sub- and superscripts seems to be partly wrong, e.g. CO2 instead of $CO_2$ etc. Futhermore, AGU paper from the

time without page numbers need to have their paper numbers included, e.g. necessary for Parekh et al. (2008) where the missing paper number is PA4202, and the page number 1–14 are useless. This is probably the case for all papers from GBC, GRL, P, and from online only journals such as Nature Communcations which start with page number 1. Check Eggleston et al 2016; Köhler et al 2010; Menviel et al. 2012; Ridgwell and Hargreaves 2007; Skinner et al. 2017; Tarnocai et al 2009 Check also, when two links exist and reduce to one (DOI). Lord et al 2015 is incomplete. Last page number is missing in Wanninkhof 1992.

**References**

Bauska, T. K., Baggenstos, D., Brook, E. J., Mix, A. C., Marcott, S. A., Petrenko, V. V., Schaefer, H., Severinghaus, J. P., and Lee, J. E.: Carbon isotopes characterize rapid changes in atmospheric carbon dioxide during the last deglaciation, Proceedings of the National Academy of Sciences, 113, 3465–3470, doi:10.1073/pnas.1513868113, 2016.

Bauska, T. K., Brook, E. J., Marcott, S. A., Baggenstos, D., Shackleton, S., Severinghaus, J. P., and Petrenko, V. V.: Controls on Millennial-Scale Atmospheric CO2 Variability During the Last Glacial Period, Geophysical Research Letters, 45, 7731–7740, doi:10.1029/2018GL077881, 2018.

Colbourn, G., Ridgwell, A., and Lenton, T. M.: The time scale of the silicate weathering negative feedback on atmospheric CO2, Global Biogeochemical Cycles, 29, 583–596, doi:10.1002/2014GB005054, 2015.

Eggleston, S., Schmitt, J., Bereiter, B., Schneider, R., and Fischer, H.: Evolution of the stable carbon isotope composition of atmospheric $CO_2$ over the last glacial cycle, Paleoceanography, 31, 434–452, doi:10.1002/2015PA002874, 2016.

Lord, N. S., Ridgwell, A., Thorne, M. C., and Lunt, D. J.: An impulse response function for the long tail of excess atmospheric CO2 in an Earth system model, Global Biogeochemical Cycles, 30, 2–17, doi:10.1002/2014GB005074, 2016.

Parekh, P., Joos, F., and Müller, S. A.: A modeling assessment of the interplay between aeolian iron fluxes and iron-binding ligands in controlling carbon dioxide fluctuations during Antarctic warm events, Paleoceanography, 23, PA4202, doi: 10.1029/2007PA001 531, 2008.

Schneider, R., Schmitt, J., Köhler, P., Joos, F., and Fischer, H.: A reconstruction of atmospheric carbon dioxide and its stable carbon isotopic composition from the penultimate glacial maximum to the last glacial inception, Climate of the Past, 9, 2507—2523, doi: 10.5194/cp-9-2507-2013, 2013.

---

## Short Comment (SC1) · 16 Oct 2019

Hi Aurich and Fortunat,

Great paper! In my opinion, this is already well-written and clearly presented. The work is relevant to anyone interested in the carbon cycle, especially to those of us interested in d13C.

I'm curious about two things:

(1) One conclusion of the paper is that you cannot get a 0.2 ‰ depletion in atmospheric d13C-CO2 and a 10 ppm rise in CO2 concentration from a terrestrial pulse,

as is suggested by Bauska 2018 as a possible cause of the variations observed at the onset of HS4. Your conclusion is based on your Figure 6, which shows that terrestrial pulses of carbon reduce d13C-CO2 only ∼0.1 ‰ per 10 ppm increase in CO2. But you state in the caption that the results in Figure 6 are the means of the model output for the 30 years immediately following the perturbations. So the atmospheric data during the perturbations (i.e. while CO2 is increasing and d13C-CO2 decreasing) are not considered? In my own modeling experiments with the OSU 14-box model (model described in Bauska 2016), fitting d13C-CO2 v. CO2 output during a land carbon pulse, not just the recovery after it, significantly decreases the slope in the Keeling plot and makes a -0.2 ‰ change in d13C-CO2 per 10 ppm increase in CO2 due to land carbon seem more feasible. Do you think that the perturbations themselves are too fast to be recorded in ice cores? Surely this is true for extremely fast pulses, but the results in Figure 6 include perturbations up to 400 years duration, which I think should be captured by ice cores even with relatively large gas age distributions.

Put another way, I observe that land carbon pulses in our box model plot as quasi-ellipses in a Keeling plot with the slope of the d13C-CO2 v. CO2 being more negative on the way "up" versus on the way "down," even for multi-centennial carbon releases. So I guess the question is - to what extent is the "up" variability recorded in ice cores? - because it may argue for the Bauska 2018 interpretation. The conclusion in your paper seems particularly strong without elaborating more on this point.

(2) Another conclusion in the paper is that the PGM-LGM d13C-CO2 difference observed in the Schneider/ Schmitt/ Eggleston work cannot be due to changes in terrestrial carbon storage, or internal reorganizations in the marine carbon cycle without considering burial. But I had a difficult time understanding how the results of your experiments argue for these points. Regarding terrestrial carbon storage, is your point that the PGM-LGM land carbon difference would have been too large because the atmospheric imprint of any smaller land carbon transfer would have been attenuated too much to cause the observed PGM-LGM d13C-CO2 difference? If so, the connection

between the model results and the stated conclusion could be clearer. I thought it was also difficult to follow your point about the marine carbon cycle. Perhaps showing the marine and atmospheric d13C data for the PGM v. LGM would help, and stating more clearly exactly how your experimental results support the conclusions.

I'm very keen to hear your thoughts. Thanks, and good luck with the rest of the submission!

- Andy Menking
* * *

---

## Referee Comment (RC2) · Anonymous Referee #2 · 2 Nov 2019

The authors explore the interactive responses of carbon and carbon isotopes in the land-ocean-atmosphere system to a suite of atmospheric perturbations on timescales spanning from decades to 100,000 years. By considering feedbacks from CaCO3 compensation and weathering as well as short term feedbacks from the land and ocean in an Earth system modeling framework, this study provides an important insight regarding changes in the global carbon cycle on a wide range of timescales. Another novel aspect of this study is that the authors developed and used analytical expressions to better understand the different responses in atmospheric CO2 and the d13C of CO2, simulated from the numerical model. This study is worth being published in Climate of the Past after clarifying the following few points:

[Figure]

1. One of the main finding is that atmospheric CO2 perturbation is relatively gradually removed over a 100,000-year timescale, due to non-negligible effects of carbonate compensation and rock weathering feedbacks on atmospheric CO2. On the other hand, the atmospheric d13C-CO2 perturbation is relatively rapidly removed because the oceanic biological pump and the subsequent burial into sediments play a dominant role in removing atmospheric perturbation (as stated in Abstract). It is interesting to see that the biological pump and the burial into sediments can effectively remove the atmospheric d13C-CO2 perturbation on an e-folding timescale of 6 years (Table 2), while oceanic d13C-DIC is removed on much longer timescales (Figure 2b). What is the mechanisms by which the oceanic d13C-DIC is removed? If the burial of organic carbon from the biological pump is responsible for the long-term removal from the land-ocean-atmosphere system, I feel that the sedimentation model deserves a more detailed description than just pointing to a previous study. I also feel that the results are too briefly discussed in the manuscript. For example, how much organic carbon is buried in time with which d13C signatures?

2. It is also possible that the dissolution and weathering fluxes of CaCO3 and the subsequent mixing with seawater might be responsible for the long-term removal of the d13C-DIC perturbation from the ocean. Furthermore, weathering fluxes to the ocean, with a d13C signature of -9.2 permil (as stated in the introduction, if I understood it correctly), might be important for the oceanic d13C-DIC budget, although this may not be directly important for the atmospheric d13C-CO2. How does the contribution from weathering fluxes to the d13C budget compare with the contribution from the burial fluxes to marine sediments? How does the weathering fluxes in the model change in time and why? It would help if the authors discuss it in more detail. In any case, it is interesting that these slow, yet persistent, carbon burial or weathering fluxes over 100,000 years can remove about 80% of the total d13C perturbation in 100,000 years. A related question is the carbon isotopes budgets for the simulations SED and CLO (i.e., Figure 2 for SED and CLO simulations)?

3. Another major finding from this study is that the burial flux of d13C is influenced by the spatial (vertical?) d13C gradients. It would be convincing if the authors show the temporal changes in the burial fluxes of d13C along with an index of the spatial d13C gradient for the suite of simulations.

4. The analytical expressions for the impulse responses indeed help to better understand the different behavior for CO2 and d13C-CO2, yet require some calcifications. The authors assumed 2 ocean-atmosphere boxes for the equation related to CO2 while assuming 3 ocean-atmosphere-land boxes for the equation related to d13C-CO2. Is it because for CO2 budget the land carbon storage can be neglected? I can follow the derivation except (A1). From (A1), I can infer that atmospheric carbon concentration is equal to DIC0 at time = 0. I think that a scaling factor might be missing here, as only 1 mol of CO2 is equilibrated with every 200 moles of DIC. The scaling factor might be included in the authors' definition for ha, and would be canceled out eventually, resulting in the same analytical expression as equation (5). Yet it should be clarified. The scaling factor depends on the perturbation P when P is large, hence we would end up arriving at an equation (5) that depends on P when P is large, which might be a reason that we see a different response for the p5000 simulation in Figure 1a (?).

---

## Author Comment (AC1) · 24 Dec 2019

**Reply to review comments**

We thank the reviewers and James Menking for assessing this manuscript and for their time and effort. The useful comments by the referees are much appreciated and helped to improve the presentation of our results.

The original review comments are given below in black, our reply in blue, and quotes from the revised manuscript in olive. All page and line numbers given below refer to those given in the originally submitted MS. During revisions, we also slightly adjusted the design of the figures. Further we adjusted the names of the experiments to better reflect changes in the model set-up. Please find attached to this reply a revised manuscript where proposed changes are highlighted.

**Reviewer 1**

This paper describes the long-term evolution of $CO_2$ -pulse experiments in the Bern3D EMIC on $CO_2$, $\delta^{13}C$, $\Delta^{14}C$. For this aim, especially the airborne fraction of these three variables are analysed, e.g. by a fit of a sum of 5 exponential functions. Furthermore, principal components of EOF are used to investigate the distribution of the signal in the ocean. The main target is the understanding of atmospheric $\delta^{13}C$ of $CO_2$ as is so far measured in ice cores covering the last 150 kyr. There, an offset in $\delta^{13}C$ between the penultimate and last glacial maximum (PGM, LGM) still lacks an explanation (Schneider et al., 2013; Eggleston et al., 2016), but also $\delta^{13}C$ during millennial-scale dynamics are also briefly discussed (Bauska et al., 2016, 2018). The paper is on a good way. Its focus is on the evolution of the isotopic signatures following a carbon pulse are to my knowledge new, while the long-term evolution of $CO_2$ pulses has already been analysed elsewhere (Colbourn et al., 2015; Lord et al., 2016), although with a different model. However, I still have a list of (partly major) points, in which I suggest the draft is revised in order to clarify open issues or sharpen its message. They are in detail:
We thank the reviewer for the support and thoughtful comments.

1. The main aim of this study is to better understand the observed evolution of $\delta^{13}C$ in the Earth system (as such explicitly written at the beginning of section 3) with special focus on the atmosphere. I therefore suggest to sharpen (and shorten) the introduction focusing on the available atmospheric $\delta^{13}C$ data (Schneider et al., 2013; Eggleston et al., 2016; Bauska et al., 2016, 2018), and delete the lengthly citations/discussion of $CO_2$ over the last 800 kyr. Maybe, also add a figure, which shows the relevant $\delta^{13}C$ data, both on glacial/interglacial and millennial-time scales, to give the reader an idea about the magnitudes of changes and the unsolved problems.
We deleted the sentence concerned with the $CO_2$ over the last 800 kyr. Concerning an additional figure we argue that in the literature plenty of such figures can be found (e.g. Schneider et al., 2013; Eggleston et al., 2016; Bauska et al., 2016, 2018). In order to not overload the manuscript we refrain from adding another figure.

2. At the end of the abstract (and maybe once in the discussion) volcanism is mentioned as a likely cause for the $\delta^{13}C$ offset between PGM and LGM. However, I believe no details on volcanism as assumed in the model are given. Is there a (constant?) volcanic $CO_2$ flux and what is its $\delta^{13}C$ signature? This final sentence of the introduction should be backed up with a more in-depth discussion, how this conclusion has been drawn. Right now on page 20 all examples given seemed to be not enough to explain the data, so an imbalance of weathering, volcanic, and burial fluxes seems to be the only possible solution, however the scenario mentioned to be the most likely one is not been investigated in detail. Maybe your insights on $\delta^{13}C$ can guide you to prescribe necessary changes in boundary conditions in such a way that the simulated $\delta^{13}C$ explains the offset between PGM and LGM (e.g. how needs volcanism change (in terms of its $\delta^{13}C$ signature and/or in its $CO_2$ release?) to generate the observed offset). This would be a real breakthrough.
We included information in the methods section (p.4, l.14) to illustrate how volcanic weathering in the model is implemented. It now reads:

Input fluxes by "weathering" of P, ALK, DIC, $DI^{13}C$, and Si are added uniformly to the coastal surface ocean. At the beginning of transient simulations, the global input fluxes of these compounds are set equal to the burial fluxes diagnosed at the end of the model spin-up. These input fluxes are jointly denoted as "weathering flux". These

fluxes are further attributed to weathering of organic material, of $CaCO_3$, of $CaSiO_3$, and to volcanic $CO_2$ out gassing. The flux of P is assigned to weathering of organic material and related C and ALK fluxes are computed by multiplication of the P flux with the Redfield ratios P:C:Alk = 1:117:-17 for organic material. Similarly, the Si flux is assigned to $CaSiO_3$-weathering and the related ALK flux is computed using Si:ALK=1:2 based on the simplified equation for $CaSiO_3$-weathering: $2CO_2 + H_2O + CaSiO_3 \rightarrow Ca^{2+} + 2HCO_3^- + SiO_2$ (Colbourn et al., 2013). The remaining ALK flux is attributed to $CaCO_3$-weathering with the stoichiometric ratio C:ALK=1:2 following from $CO_2 + H_2O + CaCO_3 \rightarrow Ca^{2+} + 2HCO_3^-$. The volcanic flux is the remaining flux needed to balance the C input flux. The diagnosed fluxes at the end of the spin up are 0.24 GtC $yr^{-1}$ for weathering of organic material, 0.14 GtC $yr^{-1}$ for $CaCO_3$ weathering, 0.08 GtC $yr^{-1}$ for volcanic $CO_2$ outgassing, and 6.96 Tmol Si $yr^{-1}$ for $CaSiO_3$-weathering. The isotopic signature of the weathering carbon corresponds to the respective signature of the burial fluxes and amounts to $\delta^{13}C$ = -9.2 ‰, intermediate between isotopically light organic carbon ($\delta^{13}C$ = -20.5 ‰) and heavier $CaCO_3$ ($\delta^{13}C$ = 2.9 ‰).

We changed the part of the discussion about long-term imbalances (p.20), please see our reply to the short-comment by James Menking for details. We are currently conducting additional experiments to investigate how volcanism would need to change to generate the observed offset in $\delta^{13}C$. These simulations are not yet done and including them here would be beyond the scope of this manuscript.

3. My understanding of the results here and of Lord et al. (2016) is, that the pulse size (labeled P here and E in the other paper) is also important for the time-dependent airborne-fractions. At least, this is the case for $CO_2$. For that reason E has been included in the sum of 5 exponential functions that fit to the model results in Lord et al. (2016) (their Eq 3), but E is missing in the fit used here (Eq 4). I believe this needs to be revised. I acknowledge that there is a subsection on the role of pulse size on page 14, but I am wondering why this is not included in the fitting equations. Maybe also compare airborne fraction of $CO_2$ with results shown in Colbourn et al. (2015); Lord et al. (2016). Any stricking difference?

The difference between Lord et al. (2016) eq. 3 and the IRF fit in this study eq. 4 is that Lord et al. (2016) fit the $CO_2$ concentration at time $t$ in ppmv whereas we fit the remaining fraction (IRF) of the pulse at time $t$. Combining the definition of the IRF in eq. 1 and the IRF fit in eq. 4 in this study yields the same expression as shown in Lord et al. (2016) eq. 3 with the difference that in our case we have a fraction ($a_0$) which remains constantly in the atmosphere. In Lord et al. (2016) experiments were run long enough (1 mio years) so that all emissions have been removed from the atmosphere (see e.g. Fig. 4 in Lord et al. (2016)).

In the results section we added a comparison to airborne fractions of $CO_2$ in Colbourn et al. (2015) (p.9, l.5):

In comparison to others, our results show, up to a few percent, lower remaining fractions for the 5000 GtC pulse (p5000) than results presented by Colbourn et al. (2015) and Lord et al. (2016). The remaining airborne fraction of the $CO_2$ perturbation is 5 % and less after 100 kyrs in all studies.

4. While Lord et al. (2016) and Colbourn et al. (2015) had only future emission in mind (and therefore the starting point has always been the modern carbon cycle), here changs in the past are in focus. One section is missing, which discusses, if (and at best how) everything said here depends on the background state of the carbon cycle. Especially, these detailed changes in $\delta^{13}C$ discussed (PGM vs LGM; millennial-scale) happened during more glacial conditions. I understand all pulse experiments have been analysed starting from pre-industrial conditions. At best, everything needs to be repeated from LGM background, but saying that, I realize, that this might double all efforts, and might be too much for the paper at hand. However, it should be shown with at least one perturbation experiment, how things are different when starting from glacial background, in order to give the reader a feeling of the size of such a potential background dependency.

This is an important question which we did not address in detail in the manuscript. Running all experiments again under glacial conditions is beyond the scope of this manuscript, as already forcing the model into a sensible glacial state is not trivial. However, we add results from two existing additional runs and split Fig. 1 into two figures to illustrate the state dependence of our results. The new figure becomes Fig. 2 in the revised manuscript. We list the two runs in Table 1 and introduce them in the methods section by adding the following on p5, l.23:

We also assess the influence of glacial climate boundary conditions and of a different land biosphere model. In two additional simulations, $LGM_{500}$ and $4box_{500}$, the Bern3D setup and spin-up is as in $WEA_{500}$, except that an atmospheric $CO_2$ concentration of 278 ppm instead of 284.7 ppm is used and the pulse is released in the first timestep of simulation year 100 instead of distributed over the year. The model configuration includes reactive sediments and weathering feedbacks. A four-box terrestrial biosphere model (Siegenthaler and Oeschger, 1987), that allows for $CO_2$ fertilization, instead of LPX-Bern is coupled to the Bern3D in these runs. In $LGM_{500}$ the model is forced by glacial boundary conditions instead of preindustrial conditions as in $4box_{500}$ and in $WEA_{500}$. $CO_2$ is set to 180 ppm and northern hemisphere ice sheets are set to Last Glacial Maximum coverage (Peltier, 1994). It has to be noted, however, that by forcing the model with 180 ppm during spin-up leads to less carbon stored in the

ocean under LGM than PI conditions and results from $LGM_{500}$ should be treated with some caution. Differences in results between $WEA_{500}$ and $4box_{500}$ are due to differences in the two land biosphere models. Differences between $4box_{500}$ and $LGM_{500}$ are exclusively due to differences in climatic boundary conditions.

In the results section we add on p12, l.34:

Next, we compare $WEA_{500}$, where LPX-Bern is used as land model, with $4box_{500}$, where a 4-box land biosphere is used. Differences in IRF($CO_{2,a}$) between $WEA_{500}$ and $4box_{500}$ are due to the different land biosphere models. The four-box land biosphere model takes up less of the $CO_2$ perturbation than LPX-Bern. The root mean squared deviation in IRF($CO_{2,a}$) from the two simulations is less than 1 % of the perturbation. Differences are most pronounced between simulation year 1 and 10,000 (Fig. 6a). Differences in IRF($\delta^{13}C_a$) (Fig. 6b) are largest in the first decades after the pulse and the root mean square deviation in IRF($\delta^{13}C_a$) amounts to less than 1 % between the two simulations. Finally, we compare the simulations $LGM_{500}$, forced by glacial boundary conditions, and $B3D_{500}$, forced by preindustrial conditions. Differences between the two simulations are solely due to the difference in climate and $CO_2$ forcing. IRF($CO_{2,a}$) is lower by a few percent in $LGM_{500}$ compared to $4box_{500}$. The difference is largest in the first 10 kyrs and up to 12 % of the perturbation. The RMSE between the two simulations amounts to ~2 %. The larger oceanic uptake in $LGM_{500}$ than in $4box_{500}$ can be explained by the higher alkalinity simulated for LGM than for preindustrial conditions. A higher alkalinity implies lower values of the Revelle factor $\xi$ and a larger carbon uptake capacity of the ocean. IRF($\delta^{13}C_a$) shows a similar pattern with lower values for the LGM background compared to the preindustrial background (cf. black dotted and dark violet dotted lines in Fig. 5b). The difference can be understood with the help of eq. 6; inserting the small glacial value for the atmospheric carbon inventory (381.6 GtC) in the nominator of the first term yields IRF$_\infty$($\delta^{13}C_a$)=0.027 compared to IRF$_\infty$($\delta^{13}C_a$)=0.032 when using the preindustrial value (589.4 GtC) in the equation. These estimates of IRF($\delta^{13}C_a$) are in good agreement with the results from $LGM_{500}$ and $4box_{500}$, respectively (Fig. 2b). In summary, the influence of glacial compared to preindustrial boundary conditions on IRF($\delta^{13}C_a$ appears modest in our model.

In the discussion we add on p21 the following (see also our reply to the short-comment by James Menking:

We acknowledge that most of the idealized experiments presented here were run under preindustrial boundary conditions whereas cold glacial conditions prevailed during HS1 and HS4. A limited set of sensitivity simulations suggest that the IRFs for $CO_2$ and $\delta^{13}C$ are similar under preindustrial and glacial $CO_2$ in our model. The direct influence of lower atmospheric $CO_2$ on IRF($\delta^{13}C$) appears limited. Further work is needed to better quantify the dependency of IRFs on the climate state.

Nevertheless, our results suggest that a release of terrestrial carbon to the atmosphere, as sole mechanism for the $CO_2$ and $\delta^{13}C_{atm}$ excursion during HS4 and HS1 seems not very likely. Other mechanisms in addition to a terrestrial carbon release likely contributed to the reconstructed variations.

Further, references in the text are changed to account for the new Fig. 2. See attached diff for details on the changes.

5. Results (Table 2): I would expect the fit parameters also for $\Delta^{14}C$, and for all scenarios and not only for p500. Maybe they can also generalized by the consideration of E in the fitting functions, see point 3 above.

We now also provide fits for $\Delta\Delta^{14}C_{only}$ and $\Delta\Delta^{14}C_{dead}$. A generalization becomes difficult as soon as pulse sizes differ substantially (i.e. $WEA_{500}$ vs. $WEA_{5000}$). This non-linearity is also visible in Fig. 4 and 5 of Lord et al. (2016).

6. When explaining why the airborne-fraction of $\delta^{13}C$ sinks at first more rapidly than that of $CO_2$, I believe this can be explained with the gross gas exchange (atmosphere-ocean), while for the $CO_2$ the net gas exchange (net oceanic uptake) is relevant, which is, as correctly written, slowed down by the carbonate chemistry (buffer or Revelle factor).

We prefer to keep our quantitative analysis which is derived in the appendix. We believe that the reviewer is referring to earlier work where the atmospheric budget for $^{13}C$ is solved by expressing change and fluxes in units of $GtC$ ‰ $yr^{-1}$. The net air-sea flux for the isotopic perturbation is then be expressed by adding the product of the gross sea-to-air $CO_2$ flux times the air-sea isotopic disequilibrium and the product of the net air-to-sea $CO_2$ flux times the isotopic perturbation in the atmosphere (see for example eq. 16 and Tab. 3a in Joos and Bruno (1998)). The first term with the gross $CO_2$ flux and the air-sea isotopic disequilibrium dominates over the term with the net $CO_2$ flux during the first years after a perturbation. However, to understand why this is the case a similar analysis as given in the appendix is necessary.

7. Eq 5: Without saying that Eq 5 is wrong, I was intuitively expecting that the pulse size P might be relevant here. The buffer factor depends on the pulse size. This is explained on page 14, line 8ff under the heading "Influence of pulse size". In the main text we added a reference to the appendix, where eq. 5 is derived. Further, we removed 'P denotes again the pulse size' on p.10, l.9 which was mistakenly stated there. The sentence on page 10 line 11 is modified to read:

$\xi$ varies with environmental conditions and increases with the size of the $CO_2$ perturbation. $\xi$ is on the order of

10 for small pulse sizes. The dependency of IRF(CO$_2$) on the size of the carbon pulse is implicitly captured in the variable $\xi$.

8. I find the detailed discussion of C budget changes in GtC over time for one specific scenario p250 (page 13-14) not that interesting. It would be better, this is expressed in fractional changes (eg expressed as airborne fraction for the atmosphere), and the paper would especially benefit if it can be generalize from one scenario to more (all?), e.g. maybe by including the dependency on the pulse size.

Please note that Figure 2 (now Fig. 3) already shows the results for two simulations. We modified the text on p13 to make this clear to the reader:

Next, we discuss the budgets of the carbon and $^{13}$C perturbations. Fig. 3 shows results for simulation $WEA_{250}$, with a pulse release of 250 GtC, and for simulation $WEA_{-250}$, with a pulse removal of 250 GtC. Differences between these two runs are clearly visible in Fig. 3, but the general evolution of the budgets is similar. In the following, we give numerical results for simulation $WEA_{250}$.

As the reviewer seems very interested in this issue, we added the figure below (Fig. 1) which illustrates the differences in responses for a range of simulations with different pulse sizes.

[Figure]

Figure 1: Changes in carbon and $\delta^{13}$C reservoirs for pulse sizes -250 GtC to 500 GtC as fractional change. Pink area indicates the range from the pulse sizes.

9. Page 14, last sentence (on radiocarbon) does not make sense to me, e.g. the first part seems to say the opposite of the second part, maybe extend with details or revise.

We changed the sentence to read:

IRF($\Delta^{14}$C$_a$) is larger for $^{14}$C$_{-500}$ than for $^{14}$C$_{dead}$ on these very long time scales. This is because the perturbation in CO$_2$ and, in turn, the long-term perturbation in $\Delta^{14}$C$_a$ is the same for both experiments. However, the initial perturbation in $\Delta^{14}$C - the denominator of the response function (see eq. 2) - is smaller in $^{14}$C$_{-500}$ than $^{14}$C$_{dead}$.

10. Discussion of millennial-scale changes (page 20, line 29ff): My reading of the papers of Bauska et al did not come to the conclusions, that they argue that the changes in CO$_2$ by 10 ppm and a decrease in $\delta^{13}$C by 0.2 ‰ can be solely explained by carbon release from terrestrial sources, but they suggest alternative processes. Please revise this discussion of Fig 6 carefully.

We revised the sentence (p.20, l.32) to read:

A release of terrestrial carbon has been discussed as a possible cause of these events (Bauska et al., 2016, 2018). This is in accordance with what is stated in the discussion in Bauska et al. (2018) on page 7738

Further, we added a second panel to Fig. 6 (Fig. 7 in the revised manuscript) and expanded the discussion on this issue. Please see our answer to the comment by James Menking

11. Fig 1 has in some scenarios some abrupt (and unexpected) changes around 20k in all three variables. I believe, this was shortly mentioned, but I do not think it is entirely explained, especially not for the isotopes.

The change, as stated in the manuscript (p.9, l.3) occurs in the control run and results from a sea ice-albedo feedback. The change in CO$_2$ is less than 1 ppm and less than 0.03 ‰ in $\delta^{13}$CO$_2$. As the control run is subtracted from all experiments, the effect is visible in the different experiments with different relative importance, depending

on the size of the perturbation. For example, in the case of experiment $WEA_{100}$ with an initial $\delta^{13}CO_2$ perturbation of about -2.5 ‰ (100 GtC with $\delta^{13}$C=-24 ‰), after 20 kyrs only about 1.5 % of the perturbation (-0.038 ‰) are left airborne. This is of the same order of magnitude as the effect of the sea ice-albedo feedback and very small. We added the information of the size of change in the control run to the text which now reads (p.9, l.3):

The jumps visible at around 20 kyrs in Fig. 1 arise from a sea ice-albedo feedback in the control run. They appear most pronounced in the experiments with small perturbations. The change in the control run in $CO_2$ is less than 1 ppm and less than 0.03 ‰ in $\delta^{13}CO_2$. These jumps are not of further relevance for our discussion.

12. Fig 5: The colorbar on the right hand side is from -30 to +30 ‰ in changes in $\delta^{13}$C. I hope there is a typo and this is wrong by some order of magnitude, otherwise it implies that on the local scale the reconstructions are complete off the target and not useful.

As stated in the figure caption, the colorbar on the right hand side shows the difference between modeled and reconstructed $\delta^{13}C_{DIC}$ using the first three principal components in % of the modeled perturbation and not the difference itself in ‰. We made this clearer in the figure caption by changing the first sentence to read:

Modeled $\delta^{13}C_{DIC}$ perturbation in ‰ (a,c,e,g) and difference between modeled and reconstructed $\delta^{13}C_{DIC}$ perturbation using the first three principal components. These differences are shown in % of the modeled perturbation (b,d,f,h).

Technical issues:

1. Subsection 2.4 Results is empty. Probably this is just the start of section 3 and Discussion is section 4.
Done.

2. Fig 2: Hatched area is "sed + wea" in the legend, but "lithosphere" in the caption. Please combine both somehow (e.g. merge to the same). Does this also include volcanism?
'sed + wea.' was changed to lithosphere' in the legend. Volcanism in the model is a constant flux of 0.08 $GtC\ yr^{-1}$ and part of the weathering input diagnosed after the spin-up. This information is now also included in the methods section (see reply to point 2 above).

3. Fig 3: grey line is called "data", but this is "modeled".
Done.

4. Fig 5: Please add in the caption the time units when describing the subfigures ("1 (c,d)" into "'1000 yr (c,d)" and "10 (e,f)" into "10 kyrs (e,f)" and "50 (g,h) kyrs" into "50 kyrs (g,h)".
Done.

5. page 11, line 1, change "fast" into "faster"
Done.

6. page 12, line 6. "additional addition" is a bit too much adding, delete one.
Done.

7. page 12, line 19, "long-term removal time scale " of what?
Of $\delta^{13}C_{atm}$; added to the text.

8. Throughout: I believe figure should be ordered by when they are refered to, but on page 2 a reference to Fig 5g shows up before any reference to Figs. 2-4.
Done. We removed the reference to Fig. 5 on p12, l20 and changed the order of Fig. 4 and Fig. 5. Further, we modified the caption of Fig. 3 to better conform with the text flow of the main text:

Figure 3: Modeled global-average perturbation of the isotopic signature of dissolved inorganic carbon in the ocean ($\delta^{13}C_{DIC}$; thick gray line) in response to a carbon pulse to the atmosphere of 500 GtC with a $\delta^{13}$C signature of -24 ‰. The thin colored lines show the perturbation as reconstructed with one (blue), two (green, dash), and three (red) principal components.

9. Appendix: Maybe consider replacing "->" in you subscripts by "2" or "→" (command rightarrow in LaTeX), e.g. in Eq. A17: change $\alpha_{a->b}$ into $\alpha_{a2b}$ or $\alpha_{a \to b}$.
Done.

10. Reference list needs careful revision, since sub- and superscripts seems to be partly wrong, e.g. $CO_2$ instead of $CO_2$ etc. Futhermore, AGU paper from the time without page numbers need to have their paper numbers included, e.g. necessary for Parekh et al. (2008) where the missing paper number is PA4202, and the page number 1–14 are useless. This is probably the case for all papers from GBC, GRL, P, and from online only journals such as Nature Communcations which start with page number 1. Check Eggleston et al 2016; Köhler et al 2010; Menviel et al. 2012; Ridgwell and Hargreaves 2007; Skinner et al. 2017; Tarnocai et al 2009 Check also, when two links exist and reduce to one (DOI). Lord et al 2015 is incomplete. Last page number is missing in Wanninkhof 1992.
Done.

There appears to be a misunderstanding. The biological pump and sediment burial are not responsible for the initial reduction of the atmospheric perturbations. Rather, air-sea gas exchange and physical ocean transport as well as land biosphere processes are responsible for the initial reduction of the perturbations. We added a corresponding sentence in the abstract and the text reads now (p.1, l.5):

On timescales from years to many centuries the atmospheric perturbations in $CO_2$ and $\delta^{13}CO_2$ are reduced by

air-sea gas exchange and physical transport from the surface to the deep ocean and by the land biosphere. Isotopic perturbations are initially removed much faster from the atmosphere than perturbations in $CO_2$ as explained by aquatic carbonate chemistry. On longer time scales, the $CO_2$ perturbation is removed by carbonate compensation and silicate rock weathering. In contrast, the $\delta^{13}C$ perturbation is removed by the relentless flux of organic and calcium carbonate particles buried in sediments.

The fast decrease in IRF($\delta^{13}C$) in the first decade is due to the fast redistribution of the perturbation in the atmosphere-ocean-land system. The main difference to IRF($CO_2$) is that the isotopic ratio is not affected by the chemical buffering from the acid-base reactions. The six year timescale from the IRF($\delta^{13}C$) fit is to the time scales of air-sea gas exchange and upper ocean mixing. This is discussed in the manuscript starting on p.9, l.16. We modified the text for the initial response at p.11, l.4 to read:

Further, the top 300 m with a carbon inventory of 2,600 GtC are approximately ventilated within a decade through air-sea gas exchange and upper ocean physical transport processes (circulation, mixing).

On long time scales, the isotopic perturbation in the ocean-atmosphere-land system are indeed jointly removed by the burial flux to the lithosphere. The isotopic perturbation in the ocean, the land biosphere and atmosphere are coupled by air-sea and air-land gas exchange. The burial flux therefore reduces the perturbations in all three reservoirs as evident in Fig. 2b. The sediment module of Bern3D is introduced on p.4, l3-22 in the methods section. The equations, parameters and evaluation of the module have been published earlier. For convenience and further reference, we add a table with all parameters in the appendix. We added the following text on p.4, l.4 to point the interested reader to the details of the model:

The governing equations of the sediment model are given in the appendix A of Tschumi et al. (2011). A comparison of simulated versus observation-based sediment composition for the setup used in this study is given in Appendix B of (Jeltsch-Thömmes et al., 2019). A table with all parameters applied here and by (Jeltsch-Thömmes et al., 2019) is given in the appendix (Table C1).

The following text is added in addition on p.4, l.12 to inform the reader on simulated burial fluxes and sedimentary stocks:

The burial flux at preindustrial steady state is 0.22 GtC yr$^{-1}$ in the form of $CaCO_3$, 0.24 GtC yr$^{-1}$ in the form of POC, and 6.96 TmolSi yr$^{-1}$ in the form of opal, all within the observational range. The global inventories in the interactive sediment layers amount to 916 GtC for $CaCO_3$, to 510 GtC for POC, and to 21,460 Tmol for opal.

Further, we add the equations of the weathering feedback parameterization as documented in (Colbourn et al., 2013) in the appendix.

The marine $\Delta\delta^{13}C_{DIC}$ perturbation is removed by the burial flux of organic carbon and $CaCO_3$. We now add further information about how much organic carbon and $CaCO_3$ is buried over time and with which signature in the section where the carbon and isotopic perturbation budgets are discussed. The new text on p.14, l. 7 reads:

We further illustrate the contribution of the burial and weathering fluxes to the carbon and isotope budgets for the simulation with ($WEA_{500}$) and without weathering feedbacks ($SED_{500}$) in Fig. 4. The inventory in the relatively fast exchanging pools – atmosphere, ocean, reactive sediments, land biosphere - increases by 350 GtC in addition to the pulse input of 500 GtC over the course of simulation $SED_{500}$. This additional increase is mainly due to a reduction in the burial of $CaCO_3$. In contrast, $CaCO_3$ burial is enhanced in simulation $WEA_{500}$ and ~430 Gt of carbon are removed from the fast exchanging reservoirs. This removal is only partly compensated by enhanced weathering (~380 GtC), leading to a reduction in the inventory of the fast exchanging pools from 500 GtC immediately after the pulse to 453 GtC at 100 kyr. Changes in POC burial fluxes are negligible for the carbon budget.

Turning to the isotopic pertubation budget, the initial perturbation due to the pulse input of -12,000 GtC ‰ is mainly removed by anomalies in the POC burial flux both in $SED_{500}$ (~10,000 GtC ‰) and $WEA_{500}$ (~7,000 GtC ‰). Changes in $CaCO_3$ burial and weathering mitigate about a quarter (~2,900 GtC ‰) of the initial isotopic perturbation in the two simulations. This leaves a relatively small isotopic inventory perturbation of +1,100 and -1,600 GtC ‰ in the fast exchanging reservoirs at 100 kyr in simulations $SED_{500}$ and $WEA_{500}$.

The contributions of the anomalies in the POC and $CaCO_3$ burial fluxes may be approximately attributed to changes in the signature ($\Delta\delta$) and to changes in flux ($\Delta F$) relative to the initial flux, $F_0$, and signature, $\delta_0$, before the perturbation:

$$\Delta(F \cdot \delta) = F_0 \cdot \Delta\delta + \Delta F \cdot \delta_0 \qquad (1)$$

The isotopic perturbation of the POC burial flux is mainly mediated by a change in the signature of the organic carbon buried ($\Delta\delta^{13}C(POC)$). For example, the signature of the total POC burial flux is on average 0.29 ‰ more negative than the corresponding input from weathering in simulation $WEA_{500}$. This change in signature of the total POC burial flux leads to an effective reduction of the isotopic perturbation. The change in POC burial flux ($\Delta F(POC)$) is, as mentioned above, negligible. Both changes in the signature of the total $CaCO_3$ burial flux and

changes in CaCO₃ burial contribute significantly to the isotopic perturbation. At the end of simulation $WEA_{500}$, the mean $\delta^{13}$C signature of CaCO₃ burial is on average 0.14 ‰ more negative than the input from weathering. In turn, the term $F_0 \cdot \Delta\delta$ for CaCO₃ burial contributes $\sim$ -3,100 GtC·‰ to the mitigation of the initial isotopic perturbation. This is partly counteracted by excess burial of CaCO₃, with $\Delta F \cdot \delta_0$ equal $\sim$1,300 GtC·‰. This yield an overall burial contribution of $\sim$ -1,800 GtC·‰. The $\delta^{13}$C perturbation is further removed by increased weathering input of CaCO₃ (see methods), contributing about 1,100 $GtC \cdot$ ‰ over 100 kyrs in $WEA_{500}$.

In summary, the isotopic perturbation is mainly mitigated by a change in the signature of the mean POC burial flux. The smaller contributions from the CaCO₃ cycle to the isotopic budget is related to both a change in the signature of the mean CaCO₃ burial flux and to changes in burial and weathering carbon fluxes.

Further, we add an additional figure (Fig. 2 in the reply) to the revised manuscript, in conjunction with the new text above.

[Figure]

Figure 2: **New Fig. 4 for the revised manuscript:** Budget plot of the anomaly fluxes in (a) GtC and (b) GtC·‰ split up in the atmosphere-ocean-land, atmosphere-ocean-land-reactive sediment, CaCO₃ burial, POC burial, and CaCO₃ weathering contribution. Solid lines show results from experiment $WEA_{500}$, dashed lines from experiment $SED_{500}$.

2. It is also possible that the dissolution and weathering fluxes of CaCO3 and the subsequent mixing with seawater might be responsible for the long-term removal of the d13C-DIC perturbation from the ocean. Furthermore, weathering fluxes to the ocean,with a d13C signature of -9.2 permil (as stated in the introduction, if I understood it correctly), might be important for the oceanic d13C-DIC budget, although this may not be directly important for the atmospheric d13C-CO2. How does the contribution from weathering fluxes to the d13C budget compare with the contribution from the burial fluxes to marine sediments? How does the weathering fluxes in the model change in time and why? It would help if the authors discuss it in more detail. In any case, it is interesting that these slow, yet persistent, carbon burial or weathering fluxes over 100,000 years can remove about 80% of the total d13C perturbation in 100,000 years. A related question is the carbon isotopes budgets for the simulations SED and CLO (i.e., Figure 2 for SED and CLO simulations)?
See also our reply to the point above. Further, we now add a table (Table 1 in the reply) to the revised manuscript that gives the carbon and carbon isotopic budgets at year 100,000 (year 90,000 for experiment $CLO_{500}$) in GtC and

Table 1: Carbon and carbon isotopic budget at year 100,000 (year 90,000 for exp. $CLO_{500}$) for atmosphere, land, ocean, sediment, excess organic carbon burial, excess $CaCO_3$ burial, and $CaCO_3$ weathering for the three experiments $WEA_{500}$, $SED_{500}$, and $CLO_{500}$. Values are given in GtC for the carbon and GtC·‰ for the carbon isotopic perturbation.

|  | exp. | atm. | land | ocean | sediment | excess burial $CaCO_3$ | excess burial POC | $CaCO_3$ weathering |
|---|---|---|---|---|---|---|---|---|
| GtC | $WEA_{500}$ | 11 | 18 | 394 | 30 | 426 | 7 | 384 |
|  | $SED_{500}$ | 38 | 71 | 741 | -1 | -338 | -7 | - |
|  | $CLO_{500}$ | 73 | 123 | 308 | - | - | - | - |
| GtC·‰ | $WEA_{500}$ | -98 | -662 | -1,028 | 164 | -1,801 | -7,006 | 1,106 |
|  | $SED_{500}$ | -205 | -1,761 | 3,503 | -25 | -2,852 | -9,664 | - |
|  | $CLO_{500}$ | -580 | -3,629 | -6,343 | - | - | - | - |

GtC·‰for the contributions from: atmosphere, land, ocean, sediment, lithosphere (split into excess organic carbon burial, excess $CaCO_3$ burial, and excess $CaCO_3$ weathering). The table is shown below and added together with the new text (see point above) on p.13.

3. Another major finding from this study is that the burial flux of d13C is influenced by the spatial (vertical?) d13C gradients. It would be convincing if the authors show the temporal changes in the burial fluxes of d13C along with an index of the spatial d13C gradient for the suite of simulations.
Please see our response to your point 2. We now show the evolution of the burial flux in a new figure. In Fig. 3 below (added to the appendix of the MS), we show the evolution of horizontally-averaged $\delta^{13}C$ of DIC in a Hovmoeller-type diagram and replace '*(not shown)*' on p.12, l.14 with the reference to the new figure in the appendix.

[Figure]

Figure 3: **New Fig. B1 for the revised manuscript:** Hovmoeller-type diagram of global $\Delta\delta^{13}C_{DIC}$ for experiment (a) $WEA_{500}$ and (b) $SED_{500}$.

4. The analytical expressions for the impulse responses indeed help to better understand the different behavior for CO2 and d13C-CO2, yet require some calcifications.The authors assumed 2 ocean-atmosphere boxes for the equation related to CO2 while assuming 3 ocean-atmosphere-land boxes for the equation related to d13C-CO2. Is it because for CO2 budget the land carbon storage can be neglected? I can follow the derivation except (A1). From

(A1), I can infer that atmospheric carbon concentration is equal to DIC0 at time = 0. I think that a scaling factor might be missing here, as only 1 mol of CO2 is equilibrated with every 200 moles of DIC. The scaling factor might be included in the authors' definition for ha, and would be canceled out eventually, resulting in the same analytical expression as equation (5). Yet it should be clarified. The scaling factor depends on the perturbation P when P is large, hence we would end up arriving at an equation (5) that depends on P when P is large, which might be a reason that we see a different response for the p5000 simulation in Figure 1a (?).

We add the following sentences from the appendix to the main text to justify our choice of a 2-box model for the $CO_2$ and a 3-box model for the $\delta^{13}C$ perturbation. The text on p.10, l.6 now reads:

Changes in land biosphere carbon storage can be considerable, in particular during the first century (see Fig. 3a). However, the role of the land carbon stock changes becomes smaller, the more carbon is taken up by the ocean and is thus for simplicity not considered in the analytical expression for $CO_2$. For $CO_2$ the expression reads: [...]

In case of the isotopic perturbation, we add on p.10, l.15:

For $\delta^{13}C_a$, we also consider the initial carbon inventory of the land biosphere ($N_{b,0}$), as a substantial amount of the isotopic perturbation is contained in the land biosphere on multi-centennial to millennial timescales (Fig. 3b). The corresponding expression reads (for the derivation see the appendix):

There is no scaling factor missing in the equations and equation A7 does not explicitly depend on the size of the pulse. The concentration $C$ is specifically defined for our box model and should not be confused with the concentration or mixing ratio of $CO_2$ in the real-world atmosphere. We note now that the Revelle factor depends on the pulse size. We have clarified the text starting on p22, l5 to read:

We consider a two box model of the atmosphere (index: $a$) and ocean (index: $o$) with area A=$A_a$=$A_o$ and heights $h_a$ and $h_o$. All boxes are assumed to be well mixed. Before the pulse addition, the model is in equilibrium and the initial $CO_2$ partial pressure is pCO$_{2,0}$=pCO$_{2,a,0}$=pCO$_{2,o,0}$. The initial concentration of dissolved inorganic carbon (DIC) in the ocean is DIC$_0$ and the carbon inventory in the ocean is given by $N_o$=DIC $\cdot h_o \cdot A$. We relate the atmospheric carbon inventory, $N_a$, to a time-varying atmospheric concentration $C_a$ and to the fixed volume of the model atmosphere: $N_a$=$C_a \cdot h_a \cdot A$. We are free to select the units of the model-specific concentration $C_a$. Here, we express $C_a$ in units of DIC$_0$ and set the initial atmospheric concentration $C_{a,0}$ equal to DIC$_0$. The atmospheric inventory is proportional to the atmospheric partial pressure, pCO$_{2,a}$ and, correspondingly, $C_a$ has to be proportional to pCO$_{2,a}$. This leads to the following definition of $C_a$:

$$C_a = \frac{\text{pCO}_{2,a}}{\text{pCO}_{2,0}}\text{DIC}_0, \tag{2}$$

$C$ is specifically defined for our box model and should not be confused with the concentration or mixing ratio of $CO_2$ in the real-world atmosphere. The atmospheric scale height $h_a$ is given by the requirement that volume times concentration equals inventory:

$$h_a = \frac{N_{a,0}}{A \cdot \text{DIC}_0}. \tag{3}$$

The perturbation ($\Delta$) from the initial equilibrium in pCO$_2$, and similarly for other quantities, is defined by: $\Delta$pCO$_2(t)$= pCO$_2(t)$-pCO$_2(t_0)$. The perturbation in pCO$_{2,o}$ is related to that in DIC by the Revelle factor $\xi$ (Revelle and Suess, 1957):

$$\xi = \frac{\Delta\text{pCO}_{2,o}}{\text{pCO}_{2,0}} \cdot \frac{\text{DIC}_0}{\Delta\text{DIC}} \tag{4}$$

We note that the carbonate chemistry is non-linear and $\xi$ varies with environmental conditions (DIC, alkalinity, temperature, salinity). Assuming a constant Revelle factor $\xi$ is therefore an approximation. In particular, $\xi$ increases with increasing DIC. As a consequence, the "effective" value of $\xi$ increases with increasing carbon emission and is thus the larger the larger the magnitude of the carbon pulses. [This yields for the perturbation in the net air-to-sea carbon flux...]

Equation 5 implicitly includes a dependence on the pulse size in form of the buffer factor $\xi$. The larger the pulse size the larger also the buffer factor and thus also the remaining atmospheric fraction. We state this in the appendix on p.27, l.17-18 and now also include it in the main text on p.11, l.6:

It has to be noted here that we assume $\xi$ constant for small perturbations. With increasing pulse sizes, also $\xi$ increases, leaving larger airborne fractions (c.f. $WEA_{500}$ versus $WEA_{5000}$ in Fig. 1a after 1-2 kyrs). Regarding $\delta^{13}C$, the corresponding [...]

**Short Comment by James Menking**

Hi Aurich and Fortunat, Great paper! In my opinion, this is already well-written and clearly presented. The work is relevant to anyone interested in the carbon cycle, especially to those of us interested in d13C.

Thank you for your support and interesting questions that helped us to improve the presentation of results.

I'm curious about two things:

(1) One conclusion of the paper is that you cannot get a 0.2 ‰ depletion in atmospheric d13C-CO2 and a 10 ppm rise in CO2 concentration from a terrestrial pulse, as is suggested by Bauska 2018 as a possible cause of the variations observed at the onset of HS4. Your conclusion is based on your Figure 6, which shows that terrestrial pulses of carbon reduce d13C-CO2 only ~0.1 ‰ per 10 ppm increase in CO2. But you state in the caption that the results in Figure 6 are the means of the model output for the 30 years immediately following the perturbations. So the atmospheric data during the perturbations (i.e. while CO2 is increasing and d13C-CO2 decreasing) are not considered? In my own modeling experiments with the OSU 14-box model (model described in Bauska 2016), fitting d13C-CO2 v. CO2 output during a land carbon pulse, not just the recovery after it, significantly decreases the slope in the Keeling plot and makes a -0.2 ‰ change in d13C-CO2 per 10 ppm increase in CO2 due to land carbon seem more feasible. Do you think that the perturbations themselves are too fast to be recorded in ice cores? Surely this is true for extremely fast pulses, but the results in Figure 6 include perturbations up to 400 years duration, which I think should be captured by ice cores even with relatively large gas age distributions. Put another way, I observe that land carbon pulses in our box model plot as quasi-ellipses in a Keeling plot with the slope of the d13C-CO2 v. CO2 being more negative on the way "up" versus on the way "down," even for multi-centennial carbon releases. So I guess the question is - to what extent is the "up" variability recorded in ice cores? - because it may argue for the Bauska 2018 interpretation. The conclusion in your paper seems particularly strong without elaborating more on this point.

Thank you for raising this issue. Averaging over different intervals and also including the 'up' variability smears out the rather straight line seen in Fig. 6 to something more like a cone, opening at higher $\Delta CO_2$ values, and also moving $\Delta\delta^{13}C$-$\Delta CO_2$ pairs towards the ice core value. We provide now more background information, add a second panel to Fig. 6 of the manuscript (Fig. 4 below, Fig. 7 in the revised manuscript) and change the corresponding paragraphs starting on p.20, l.29 to now read:

[Figure]

Figure 4: **New Fig. 7 for the revised manuscript:** (a) First 2 kyrs of the $\Delta\delta^{13}C$ to $\Delta CO_2$ ratio of a 100 GtC pulse ($WEA_{100}$, $LGM_{100}$) plotted over time of single pulses (teal crosses) and series of pulses up to the "year after pulse" where plotted (purple crosses). Dashed horizontal line indicates a $\Delta\delta^{13}C$ to $\Delta CO_2$ ratio of 10 ppm to -0.2 ‰ as seen in ice core records for HS1 and HS4. (b) Response in $\Delta CO_2$ and $\Delta\delta^{13}C_{atm}$ to a uniform release of 20, 40, 80, 120, and 160 GtC (colors) of isotopically light carbon ($\delta^{13}C$ = -24 ‰) to the atmosphere over 100, 200, 300, and 400 years (numbers). Shown is the mean over 30 years starting in the last year of the release. Red cross indicates the approx. $\Delta\delta^{13}C$ to $\Delta CO_2$ change as seen in ice core records for HS1 and HS4 (Bauska et al., 2016, 2018).

Further, the different behavior of $CO_2$ and $\delta^{13}C_{atm}$ perturbations has also consequences for centennial scale $CO_2$ and $\delta^{13}C_{atm}$ variations such as during Heinrich Stadial (HS) 4 and 1. Variations in $CO_2$ and $\delta^{13}C_{atm}$ during HS4 and HS1 have been measured on Antarctic ice cores. The data show an increase in $CO_2$ on the order of ~10 ppm over 200-300 years, accompanied by a decrease in $\delta^{13}C_{atm}$ of ~0.2 ‰. This results in a value of roughly -0.02 ‰ per ppm for the ratio of the change in $\delta^{13}C$ to the change in $CO_2$ ($r$= $\Delta\delta^{13}C$ / $\Delta CO_2$). These changes have been attributed to a release of terrestrial carbon to the atmosphere (Bauska et al., 2016, 2018).

We utilize our pulse response simulations under PI and LGM boundary conditions to estimate the changes in $r$ in response to a transient terrestrial carbon input with an isotopic signature of -24 ‰. The results for $r$ will depend on the evolution of the emissions into the atmosphere. For a pulse-like input at time zero the evolution of $r$ after the pulse is shown in Fig. 4 for the first 2,000 years. $r$ is about -0.08 (-0.06) ‰ ppm$^{-1}$ immediately after the release under LGM (PI) boundary conditions. The difference between LGM and PI values of $r$ comes from the difference in the atmospheric C and $^{13}$C inventory to which the pulse is added. The magnitude of $r$ is reduced rapidly within the first 25 years and remains approximately constant at -0.015 (-0.01) ‰ ppm$^{-1}$ thereafter. Thus, $r$ recorded in ice after a hypothetical pulse release of carbon would be closer to -0.01 than -0.02 ‰ ppm$^{-1}$ if the age distribution of the ice core samples is wider than a few decades. However, emissions may occur more gradually. For simplicity, we assume an emission pulse sustained over period $T$ according to a two-sided Heaviside function; terrestrial emissions, $e(t)$, suddenly increase to unity, remain elevated for period $T$, and are then immediately reduced again to zero. $r$ is then readily calculated from the IRFs ($r = \int_0^T dt \cdot e(t) \cdot \text{IRF}(\delta^{13}C)(T-t)/\int_0^T dt \cdot e(t) \cdot \text{IRF}(CO_2)(T-t)$). The results for $r$ are again plotted in Fig. 4. $r$ is decreasing with the duration $T$ of sustained emissions and falls below -0.02 ‰ ppm$^{-1}$ for $T$ larger than about 90 (30) years for LGM (PI) boundary conditions. Finally, we also performed simulations with the Bern3D-LPX under PI boundary conditions to account for non-linearity that may not be captured in the IRF representation. The releases of varying amounts of terrestrial carbon with $\delta^{13}C$ = -24 ‰ over 100 to 400 years yield a linear response in $\Delta CO_2$ and $\Delta\delta^{13}C_{atm}$ (Fig. 4) with a slope $r$ of about -0.01 ‰ ppm$^{-1}$.

There are uncertainties in these estimates of $r$. We assumed that the emitted carbon was about -18 ‰ depleted with respect to the atmosphere. The relative uncertainty in this value translates directly to the relative uncertainty in $r$. The release of material that is isotopically even more depleted would bring our model results in better agreement with the ice core estimate of $r$. Observations and land model simulations suggest that the $\delta^{13}C$ depletion in vegetation and soils may vary between -24 ‰, for heavily discriminated C3 plant material, to -5 indicative of C4 plant material Keller et al. (2018). C4 plants were likely more abundant during glacials than interglacials (Collatz et al., 1998). Also, the values shown in Fig. 4 are 30-year means. A wider gas age distribution would reduce the perturbation in $\delta^{13}C_{atm}$ more than in $CO_2$ (see e.g. Köhler et al., 2011). On the other hand, the increase in While there are uncertainties in our model estimates as well as in the determination of $r$ from the ice core measurements, our analysis suggests that other mechanisms in addition to a terrestrial release appear to have contributed to the $CO_2$ and $\delta^{13}C_{atm}$ excursions during HS4 and HS1.

We acknowledge that most of the idealized simulations presented in this study were run under preindustrial boundary conditions. A limited set of sensitivity simulations suggest that the IRFs for $CO_2$ and $\delta^{13}C$ are similar under preindustrial and glacial $CO_2$ in our model. The direct influence of lower atmospheric $CO_2$ on IRF($\delta^{13}C$) appears limited. Further work is needed to better quantify the dependency of IRFs on the climate state.

We also adjusted the wording in the abstract to better reflect these uncertainties:
Our results may suggest that changes in terrestrial carbon storage were not the sole cause for the abrupt, centennial $CO_2$ and $\delta^{13}C$ variations recorded in ice during Heinrich Stadials HS1 and HS4 of the last glacial period. However, model and data uncertainties prevent a firm conclusion.

(2) Another conclusion in the paper is that the PGM-LGM d13C-CO2 difference observed in the Schneider/ Schmitt/ Eggleston work cannot be due to changes in terrestrial carbon storage, or internal reorganizations in the marine carbon cycle without considering burial. But I had a difficult time understanding how the results of your experiments argue for these points. Regarding terrestrial carbon storage, is your point that the PGM-LGM land carbon difference would have been too large because the atmospheric imprint of any smaller land carbon transfer would have been attenuated too much to cause the observed PGM-LGM d13C-CO2 difference? If so, the connection between the model results and the stated conclusion could be clearer. I thought it was also difficult to follow your point about the marine carbon cycle. Perhaps showing the marine and atmospheric d13C data for the PGM v. LGM would help, and stating more clearly exactly how your experimental results support the conclusions.

In light of your comments we modified the text for clarification. The paragraph (p.20, l. 9) reads now:
Our results have consequences for the interpretation of the difference in $\delta^{13}C$ between similar climate states such as the Penultimate Glacial Maximum (PGM) and the Last Glacial Maximum (LGM). Substantial temporal $\delta^{13}C$ differences between these periods are recorded in ice cores (Schneider et al., 2013; Eggleston et al., 2016) and in marine sediments (Hoogakker et al., 2006; Oliver et al., 2010). Different mechanisms, such as changes in the $\delta^{13}C$ signature of weathering and burial fluxes, varying contribution of volcanic outgassing of $CO_2$, and changes in the amount of carbon stored in the land biosphere, especially in yedoma and permafrost soils, have been discussed for the PGM-LGM $\delta^{13}C$ offset (e.g. Lourantou et al., 2010; Schneider et al., 2013). An internal reorganization

[revised manuscript text omitted]

Joos, F. and Bruno, M. (1998). Long-term variability of the terrestrial and oceanic carbon sinks and the budgets of the carbon isotopes 13 C and 14 C. *Global Biogeochemical Cycles*, 12(2):277–295.

Keller, D. P., Lenton, A., Scott, V., Vaughan, N. E., Bauer, N., Ji, D., Jones, C. D., Kravitz, B., Muri, H., and Zickfeld, K. (2018). The Carbon Dioxide Removal Model Intercomparison Project (CDRMIP): Rationale and experimental protocol for CMIP6. *Geoscientific Model Development*, 11(3):1133–1160.

Köhler, P., Knorr, G., Buiron, D., Lourantou, A., and Chappellaz, J. (2011). Abrupt rise in atmospheric CO2 at the onset of the Bølling/Allerød: in-situ ice core data versus true atmospheric signals. *Climate of the Past*, 7(2):473–486.

Lindgren, A., Hugelius, G., and Kuhry, P. (2018). Extensive loss of past permafrost carbon but a net accumulation into present-day soils. *Nature*, 560(7717):219–222.

Lord, N. S., Ridgwell, A., Thorne, M. C., and Lunt, D. J. (2016). An impulse response function for the âlong tailâ of excess atmospheric CO2 in an Earth system model. *Global Biogeochemical Cycles*, 30(1):2–17.

Lourantou, A., Chappellaz, J., Barnola, J. M., Masson-Delmotte, V., and Raynaud, D. (2010). Changes in atmospheric CO2 and its carbon isotopic ratio during the penultimate deglaciation. *Quaternary Science Reviews*, 29(17-18):1983–1992.

Oliver, K. I. C., Hoogakker, B. A. A., Crowhurst, S., Henderson, G. M., Rickaby, R. E. M., Edwards, N. R., and Elderfield, H. (2010). A synthesis of marine sediment core d13C data over the last 150 000 years. *Climate of the Past*, 6(5):645–673.

Peltier, W. R. (1994). Ice Age Paleotopography. *Science*, 265(5169):195 – 201.

Revelle, R. and Suess, H. E. (1957). Carbon Dioxide Exchange Between Atmosphere and Ocean and the Question of an Increase of Atmospheric CO2 during the Past Decades. *Tellus*, 9(1):18–27.

Schneider, R., Schmitt, J., Köhler, P., Joos, F., and Fischer, H. (2013). A reconstruction of atmospheric carbon dioxide and its stable carbon isotopic composition from the penultimate glacial maximum to the last glacial inception. *Climate of the Past*, 9(6):2507–2523.

Siegenthaler, U. and Oeschger, H. (1987). Biospheric CO2 emissions during the past 200 years reconstructed by deconvolution of ice core data. *Tellus B: Chemical and Physical Meteorology*, 39(1-2):140–154.

Tarnocai, C., Canadell, J. G., Schuur, E. A. G., Kuhry, P., Mazhitova, G., and Zimov, S. (2009). Soil organic carbon pools in the northern circumpolar permafrost region. *Global Biogeochemical Cycles*, 23(2):GB2023.

Tschumi, T., Joos, F., Gehlen, M., and Heinze, C. (2011). Deep ocean ventilation, carbon isotopes, marine sedimentation and the deglacial CO2 rise. *Climate of the Past*, 7(3):771–800.

[Figure]

Figure 5: **New Fig. 1 for the revised manuscript:** (a) $CO_2$,(b) $\delta^{13}C$, and (c) $\Delta^{14}C$ perturbation in the atmosphere for different simulations with different pulse sizes and model configurations. The isotopic signature is $\delta^{13}C$=-24 ‰ for all pulse sizes except the 5000 GtC pulse with $\delta^{13}C$=-28 ‰. Dashed and dotted brown lines in panel (a) and (b) show results from a 500 GtC pulse with no weathering feedback and no ocean sediments, respectively. See Table 1 and section 2.2 for details of the experimental setup of the runs. Note that the scaling of the second y-axis is different for the three panels. Data are filtered with a moving average of 1000 years in the interval 1 kyr - 10 kyrs and 5000 years in the interval 10 kyr - 100 kyrs for better visibility. Factorial simulations ($WEA_{500}$, $SED_{500}$, $CLO_{500}$, $4box_{500}$, and $LGM_{500}$) for the time interval 1 kyr to 100 kyrs are shown in Fig. 6 for better visibility.

[revised manuscript text omitted]

---

## Author Comment (AC2) · 5 Jan 2020

The comment was uploaded in the form of a supplement:
https://www.clim-past-discuss.net/cp-2019-107/cp-2019-107-AC2-supplement.pdf

---

## Author Comment (AC3) · 5 Jan 2020

**Reply to review comments**

We thank the reviewers and James Menking for assessing this manuscript and for their time and effort. The useful comments by the referees are much appreciated and helped to improve the presentation of our results.

The original review comments are given below in black, our reply in blue, and quotes from the revised manuscript in olive. All page and line numbers given below refer to those given in the originally submitted MS. During revisions, we also slightly adjusted the design of the figures. Further, we adjusted the names of the experiments to better reflect changes in the model set-up. Please find attached to this reply a revised manuscript where proposed changes are highlighted.

**Reviewer 1**

This paper describes the long-term evolution of $CO_2$-pulse experiments in the Bern3D EMIC on $CO_2$, $\delta^{13}C$, $\Delta^{14}C$. For this aim, especially the airborne fraction of these three variables are analysed, e.g. by a fit of a sum of 5 exponential functions. Furthermore, principal components of EOF are used to investigate the distribution of the signal in the ocean. The main target is the understanding of atmospheric $\delta^{13}C$ of $CO_2$ as is so far measured in ice cores covering the last 150 kyr. There, an offset in $\delta^{13}C$ between the penultimate and last glacial maximum (PGM, LGM) still lacks an explanation (Schneider et al., 2013; Eggleston et al., 2016), but also $\delta^{13}C$ during millennial-scale dynamics are also briefly discussed (Bauska et al., 2016, 2018). The paper is on a good way. Its focus is on the evolution of the isotopic signatures following a carbon pulse are to my knowledge new, while the long-term evolution of $CO_2$ pulses has already been analysed elsewhere (Colbourn et al., 2015; Lord et al., 2016), although with a different model. However, I still have a list of (partly major) points, in which I suggest the draft is revised in order to clarify open issues or sharpen its message. They are in detail:

We thank the reviewer for the support and thoughtful comments.

1. The main aim of this study is to better understand the observed evolution of $\delta^{13}C$ in the Earth system (as such explicitly written at the beginning of section 3) with special focus on the atmosphere. I therefore suggest to sharpen (and shorten) the introduction focusing on the available atmospheric $\delta^{13}C$ data (Schneider et al., 2013; Eggleston et al., 2016; Bauska et al., 2016, 2018), and delete the lengthly citations/discussion of $CO_2$ over the last 800 kyr. Maybe, also add a figure, which shows the relevant $\delta^{13}C$ data, both on glacial/interglacial and millennial-time scales, to give the reader an idea about the magnitudes of changes and the unsolved problems.

We deleted the sentence concerned with the $CO_2$ over the last 800 kyr. Concerning an additional figure we argue that in the literature plenty of such figures can be found (e.g. Schneider et al., 2013; Eggleston et al., 2016; Bauska et al., 2016, 2018). In order to not overload the manuscript we refrain from adding another figure.

2. At the end of the abstract (and maybe once in the discussion) volcanism is mentioned as a likely cause for the $\delta^{13}C$ offset between PGM and LGM. However, I believe no details on volcanism as assumed in the model are given. Is there a (constant?) volcanic $CO_2$ flux and what is its $\delta^{13}C$ signature? This final sentence of the introduction should be backed up with a more in-depth discussion, how this conclusion has been drawn. Right now on page 20 all examples given seemed to be not enough to explain the data, so an imbalance of weathering, volcanic, and burial fluxes seems to be the only possible solution, however the scenario mentioned to be the most likely one is not been investigated in detail. Maybe your insights on $\delta^{13}C$ can guide you to prescribe necessary changes in boundary conditions in such a way that the simulated $\delta^{13}C$ explains the offset between PGM and LGM (e.g. how needs volcanism change (in terms of its $\delta^{13}C$ signature and/or in its $CO_2$ release?) to generate the observed offset). This would be a real breakthrough.

We included information in the methods section (p.4, l.14) to illustrate how volcanic weathering in the model is implemented. It now reads:

Input fluxes by "weathering" of P, ALK, DIC, DI$^{13}$C, and Si are added uniformly to the coastal surface ocean. At the beginning of transient simulations, the global input fluxes of these compounds are set equal to the burial fluxes diagnosed at the end of the model spin-up. These input fluxes are jointly denoted as "weathering flux". These fluxes

are further attributed to weathering of organic material, of $CaCO_3$, of $CaSiO_3$, and to volcanic $CO_2$ outgassing. The flux of P is assigned to weathering of organic material and related C and ALK fluxes are computed by multiplication of the P flux with the Redfield ratios P:C:Alk = 1:117:-17 for organic material. Similarly, the Si flux is assigned to $CaSiO_3$-weathering and the related ALK flux is computed using Si:ALK=1:2 based on the simplified equation for $CaSiO_3$-weathering: $2CO_2 + H_2O + CaSiO_3 \rightarrow Ca^{2+} + 2HCO_3^- + SiO_2$ (Colbourn et al., 2013). The remaining ALK flux is attributed to $CaCO_3$-weathering with the stoichiometric ratio C:ALK=1:2 following from $CO_2 + H_2O + CaCO_3 \rightarrow Ca^{2+} + 2HCO_3^-$. The volcanic flux is the remaining flux needed to balance the C input flux. The diagnosed fluxes at the end of the spin up are 0.24 GtC $yr^{-1}$ for weathering of organic material, 0.14 GtC $yr^{-1}$ for $CaCO_3$ weathering, 0.08 GtC $yr^{-1}$ for volcanic $CO_2$ outgassing, and 6.96 Tmol Si $yr^{-1}$ for $CaSiO_3$-weathering. The isotopic signature of the weathering carbon corresponds to the respective signature of the burial fluxes and amounts to $\delta^{13}$C = -9.2 ‰, intermediate between isotopically light organic carbon ($\delta^{13}$C = -20.5 ‰) and heavier $CaCO_3$ ($\delta^{13}$C = 2.9 ‰).

We changed the part of the discussion about long-term imbalances (p.20), please see our reply to the short-comment by James Menking for details. We are currently conducting additional experiments to investigate how volcanism would need to change to generate the observed offset in $\delta^{13}$C. These simulations are not yet done and including them here would be beyond the scope of this manuscript.

3. My understanding of the results here and of Lord et al. (2016) is, that the pulse size (labeled P here and E in the other paper) is also important for the time-dependent airborne-fractions. At least, this is the case for $CO_2$. For that reason E has been included in the sum of 5 exponential functions that fit to the model results in Lord et al. (2016) (their Eq 3), but E is missing in the fit used here (Eq 4). I believe this needs to be revised. I acknowledge that there is a subsection on the role of pulse size on page 14, but I am wondering why this is not included in the fitting equations. Maybe also compare airborne fraction of $CO_2$ with results shown in Colbourn et al. (2015); Lord et al. (2016). Any stricking difference?

The difference between Lord et al. (2016) eq. 3 and the IRF fit in this study eq. 4 is that Lord et al. (2016) fit the $CO_2$ concentration at time $t$ in ppmv whereas we fit the remaining fraction (IRF) of the pulse at time $t$. Combining the definition of the IRF in eq. 1 and the IRF fit in eq. 4 in this study yields the same expression as shown in Lord et al. (2016) eq. 3 with the difference that in our case we have a fraction ($a_0$) which remains constantly in the atmosphere. In Lord et al. (2016) experiments were run long enough (1 mio years) so that all emissions have been removed from the atmosphere (see e.g. Fig. 4 in Lord et al. (2016)).

In the results section we added a comparison to airborne fractions of $CO_2$ in Colbourn et al. (2015) (p.9, l.5):

In comparison to others, our results show, up to a few percent, lower remaining fractions for the 5000 GtC pulse ($WEA_{5000}$) than results presented by Colbourn et al. (2015) and Lord et al. (2016). The remaining airborne fraction of the $CO_2$ perturbation is 5 % and less after 100 kyr in all studies.

4. While Lord et al. (2016) and Colbourn et al. (2015) had only future emission in mind (and therefore the starting point has always been the modern carbon cycle), here changs in the past are in focus. One section is missing, which discusses, if (and at best how) everything said here depends on the background state of the carbon cycle. Especially, these detailed changes in $\delta^{13}$C discussed (PGM vs LGM; millennial-scale) happened during more glacial conditions. I understand all pulse experiments have been analysed starting from pre-industrial conditions. At best, everything needs to be repeated from LGM background, but saying that, I realize, that this might double all efforts, and might be too much for the paper at hand. However, it should be shown with at least one perturbation experiment, how things are different when starting from glacial background, in order to give the reader a feeling of the size of such a potential background dependency.

This is an important question which we did not address in detail in the manuscript. Running all experiments again under glacial conditions is beyond the scope of this manuscript, as already forcing the model into a sensible glacial state is not trivial. However, we add results from existing additional runs and split Fig. 1 into two figures to illustrate the state dependence of our results. The new figure becomes Fig. 2 in the revised manuscript. We list the additional runs in Table 1 and introduce them in the methods section by adding the following on p.5, l.23:

We also assess the influence of glacial climate boundary conditions and of a different land biosphere model, in two additional simulations, $LGM_{500}$ and $4box_{500}$. In $4box_{500}$ the Bern3D setup and spin-up is as in $WEA_{500}$, except that an atmospheric $CO_2$ concentration of 278 ppm instead of 284.7 ppm is used and the pulse is released in the first timestep of simulation year 100 instead distributed over the year. The model configuration includes reactive sediments and weathering feedbacks. A four-box terrestrial biosphere model (Siegenthaler and Oeschger, 1987), that allows for $CO_2$ fertilization, instead of LPX-Bern is coupled to the Bern3D in these runs. In $LGM_{500}$ the model is forced by glacial boundary conditions instead of preindustrial conditions as in $4box_{500}$ and in $WEA_{500}$. $CO_2$ is set to 180 ppm and northern hemisphere ice sheets are set to Last Glacial Maximum coverage (Peltier, 1994). It has to be noted, however, that by forcing the model with 180 ppm during spin-up leads to less carbon stored in the

ocean under LGM than PI conditions and results from $LGM_{500}$ should be treated with some caution. Differences in results between $WEA_{500}$ and $4box_{500}$ are due to differences in the two land biosphere models. Differences between $4box_{500}$ and $LGM_{500}$ are exclusively due to differences in climatic boundary conditions.

In the results section we add on p12, l.34:

Next, we compare $WEA_{500}$, where LPX-Bern is used as land model, with $4box_{500}$, where a 4-box land biosphere is used. Differences in IRF($CO_{2,a}$) between $WEA_{500}$ and $4box_{500}$ are due to the different land biosphere models. The four-box land biosphere model takes up less of the $CO_2$ perturbation than LPX-Bern. The root mean squared deviation in IRF($CO_{2,a}$) from the two simulations is less than 1 % of the perturbation. Differences are most pronounced between simulation year 1 and 10,000 (Fig. 2a). Differences in IRF($\delta^{13}C_a$) (Fig. 2b) are largest in the first decades after the pulse and the root mean square deviation in IRF($\delta^{13}C_a$) amounts to less than 1 % between the two simulations.

Finally, we compare the simulations $LGM_{500}$, forced by glacial boundary conditions, and $4box_{500}$, forced by preindustrial conditions. Differences between the two simulations are solely due to the difference in climate and $CO_2$ forcing. IRF($CO_{2,a}$) is lower by a few percent in $LGM_{500}$ compared to $4box_{500}$ (Fig. 2a). The difference is largest in the first 10 kyr and up to 12 % of the perturbation. The root mean square deviation between the two simulations amounts to ~2 %. The larger oceanic uptake in $LGM_{500}$ than in $4box_{500}$ can be explained by the higher alkalinity simulated for LGM than for preindustrial conditions. A higher alkalinity implies lower values of the Revelle factor $\xi$ and a larger carbon uptake capacity of the ocean. IRF($\delta^{13}C_a$) shows a similar pattern with lower values for the LGM background compared to the preindustrial background (cf. black dotted and dark violet dotted lines in Fig. 1b). The difference can be understood with the help of eq. 6; inserting the small glacial value for the atmospheric carbon inventory (381.6 GtC) in the nominator of the first term yields IRF$_\infty$($\delta^{13}C_a$)=0.027 compared to IRF$_\infty$($\delta^{13}C_a$)=0.032 when using the preindustrial value (589.4 GtC) in the equation. These estimates of IRF($\delta^{13}C_a$) are in good agreement with the results from $LGM_{500}$ and $4box_{500}$, respectively (Fig. 2b). In summary, the influence of glacial compared to preindustrial boundary conditions on IRF($\delta^{13}C_a$) appears modest in our model.

In the discussion we add on p21 the following (see also our reply to the short-comment by James Menking:

We acknowledge that most of the idealized experiments presented here were run under preindustrial boundary conditions A limited set of sensitivity simulations suggest that the IRFs for $CO_2$ and $\delta^{13}C$ are similar under preindustrial and glacial $CO_2$ in our model. The direct influence of lower atmospheric $CO_2$ on IRF($\delta^{13}C$) appears limited. Further work is needed to better quantify the dependency of IRFs on the climate state.

Further, references in the text are changed to account for the new Fig. 2. See attached diff for details on the changes.

5. Results (Table 2): I would expect the fit parameters also for $\Delta^{14}C$, and for all scenarios and not only for p500. Maybe they can also generalized by the consideration of E in the fitting functions, see point 3 above.

We now also provide fits for $\Delta\Delta^{14}C_{only}$ and $\Delta\Delta^{14}C_{dead}$. A generalization becomes difficult as soon as pulse sizes differ substantially (i.e. $WEA_{500}$ vs. $WEA_{5000}$). This non-linearity is also visible in Fig. 4 and 5 of Lord et al. (2016).

6. When explaining why the airborne-fraction of $\delta^{13}C$ sinks at first more rapidly than that of $CO_2$ , I believe this can be explained with the gross gas exchange (atmosphere-ocean), while for the $CO_2$ the net gas exchange (net oceanic uptake) is relevant, which is, as correctly written, slowed down by the carbonate chemistry (buffer or Revelle factor).

We prefer to keep our quantitative analysis which is derived in the appendix. We believe that the reviewer is referring to earlier work where the atmospheric budget for $^{13}C$ is solved by expressing change and fluxes in units of $GtC$ ‰ $yr^{-1}$. The net air-sea flux for the isotopic perturbation is then expressed by adding the product of the gross sea-to-air $CO_2$ flux times the air-sea isotopic disequilibrium and the product of the net air-to-sea $CO_2$ flux times the isotopic perturbation in the atmosphere (see for example eq. 16 and Tab. 3a in Joos and Bruno (1998)). The first term with the gross $CO_2$ flux and the air-sea isotopic disequilibrium dominates over the term with the net $CO_2$ flux during the first years after a perturbation. However, to understand why this is the case a similar analysis as given in the appendix is necessary.

7. Eq 5: Without saying that Eq 5 is wrong, I was intuitively expecting that the pulse size P might be relevant here. The buffer factor depends on the pulse size. This is explained on page 14, line 8ff under the heading "Influence of pulse size". In the main text we added a reference to the appendix, where eq. 5 is derived. Further, we removed 'P denotes again the pulse size' on p.10, l.9 which was mistakenly stated there. The sentence on page 10 line 11 is modified to read:

$\xi$ varies with environmental conditions and increases with the size of the $CO_2$ perturbation. $\xi$ is on the order of 10 for small pulse sizes. The dependency of IRF($CO_2$) on the size of the carbon pulse is implicitly captured in the variable $\xi$.

8. I find the detailed discussion of C budget changes in GtC over time for one specific scenario p250 (page 13-14) not that interesting. It would be better, this is expressed in fractional changes (eg expressed as airborne fraction for the atmosphere), and the paper would especially benefit if it can be generalize from one scenario to more (all?), e.g. maybe by including the dependency on the pulse size.

Please note that Figure 2 (now Fig. 3) already shows the results for two simulations. We modified the text on p.13 to make this clear to the reader:

Next, we discuss the budgets of the carbon and $^{13}$C perturbations. Figure 3 shows results for simulation $WEA_{250}$, with a pulse release of 250 GtC, and for simulation $WEA_{-250}$, with a pulse removal of 250 GtC. Differences between these two runs are clearly visible in Fig. 3, but the general evolution of the budgets is similar. In the following, we give numerical results for simulation $WEA_{250}$.

As the reviewer seems very interested in this issue, we added the figure below (Fig. 1) which illustrates the differences in responses for a range of simulations with different pulse sizes.

[Figure]

Figure 1: Changes in carbon and $\delta^{13}$C reservoirs for pulse sizes -250 GtC to 500 GtC as fractional change. Pink area indicates the range from the pulse sizes.

9. Page 14, last sentence (on radiocarbon) does not make sense to me, e.g. the first part seems to say the opposite of the second part, maybe extend with details or revise.

We changed the sentence to read:

IRF($\Delta^{14}C_a$) is larger for $^{14}C_{-500}$ than for $^{14}C_{dead}$ on these very long time scales. This is because the perturbation in $CO_2$ and, in turn, the long-term perturbation in $\Delta^{14}C_a$ is the same for both experiments. However, the initial perturbation in $\Delta^{14}C$ - the denominator of the response function (see eq. 2) - is smaller in $^{14}C_{-500}$ than $^{14}C_{dead}$.

10. Discussion of millennial-scale changes (page 20, line 29ff): My reading of the papers of Bauska et al did not come to the conclusions, that they argue that the changes in $CO_2$ by 10 ppm and a decrease in $\delta^{13}$C by 0.2 ‰ can be solely explained by carbon release from terrestrial sources, but they suggest alternative processes. Please revise this discussion of Fig 6 carefully.

We revised the sentence (p.20, l.32) to read:

A release of terrestrial carbon has been discussed as a possible cause of these events (Bauska et al., 2016, 2018).

This is in accordance with what is stated in the discussion in Bauska et al. (2018) on page 7738.

Further, we added a second panel to Fig. 6 (Fig. 8 in the revised manuscript) and expanded the discussion on this issue. Please see our answer to the comment by James Menking

11. Fig 1 has in some scenarios some abrupt (and unexpected) changes around 20k in all three variables. I believe, this was shortly mentioned, but I do not think it is entirely explained, especially not for the isotopes.

The change, as stated in the manuscript (p.9, l.3) occurs in the control run and results from a sea ice-albedo feedback. The change is less than 1 ppm in $CO_2$ and less than 0.03 ‰ in $\delta^{13}CO_2$. As the control run is subtracted from all experiments, the effect is visible in the different experiments with different relative importance, depending on the size of the perturbation. For example, in the case of experiment $WEA_{100}$ with an initial $\delta^{13}CO_2$ perturbation of about -2.5 ‰ (100 GtC with $\delta^{13}$C=-24 ‰), after 20 kyr only about 1.5 % of the perturbation (-0.038 ‰) are left

airborne. This is of the same order of magnitude as the effect of the sea ice-albedo feedback and very small. We added the information of the size of change in the control run to the text which now reads (p.9, l.3):
The jumps visible at around 20 kyr in Fig. 1 arise from a sea ice-albedo feedback in the control run. They appear most pronounced in the experiments with small perturbations. The change in the control run in $CO_2$ is less than 1 ppm and less than 0.03 ‰ in $\delta^{13}CO_2$. These jumps are not of further relevance for our discussion.

12. Fig 5: The colorbar on the right hand side is from -30 to +30 ‰ in changes in $\delta^{13}C$. I hope there is a typo and this is wrong by some order of magnitude, otherwise it implies that on the local scale the reconstructions are complete off the target and not useful.
As stated in the figure caption, the colorbar on the right hand side shows the difference between modeled and reconstructed $\delta^{13}C_{DIC}$ using the first three principal components in % of the modeled perturbation and not the difference itself in ‰. We made this clearer in the figure caption by changing the first sentence to read:
Modeled $\delta^{13}C_{DIC}$ perturbation in ‰ (a,c,e,g) and difference between modeled and reconstructed $\delta^{13}C_{DIC}$ perturbation using the first three principal components. These differences are shown in % of the modeled perturbation (b,d,f,h).

Technical issues:
1. Subsection 2.4 Results is empty. Probably this is just the start of section 3 and Discussion is section 4.
Done.
2. Fig 2: Hatched area is "sed + wea" in the legend, but "lithosphere" in the caption. Please combine both somehow (e.g. merge to the same). Does this also include volcanism?
'sed + wea.' was changed to lithosphere' in the legend. Volcanism in the model is a constant flux of 0.08 $GtC\ yr^{-1}$ and part of the weathering input diagnosed after the spin-up. This information is now also included in the methods section (see reply to point 2 above).
3. Fig 3: grey line is called "data", but this is "modeled".
Done.
4. Fig 5: Please add in the caption the time units when describing the subfigures ("1 (c,d)" into '"1000 yr (c,d)" and "10 (e,f)" into "10 kyr (e,f)" and "50 (g,h) kyr" into "50 kyr (g,h)".
Done.
5. page 11, line 1, change "fast" into "faster"
Done.
6. page 12, line 6. "additional addition" is a bit too much adding, delete one.
Done.
7. page 12, line 19, "long-term removal time scale " of what?
Of $\delta^{13}C_{atm}$; added to the text.
8. Throughout: I believe figure should be ordered by when they are refered to, but on page 2 a reference to Fig 5g shows up before any reference to Figs. 2-4.
Done. We removed the reference to Fig. 5 on p.12, l.20 and changed the order of Fig. 4 and Fig. 5. Further, we modified the caption of Fig. 3 (Fig. 5 in the revised MS) to better conform with the text flow of the main text:
Figure 5: Modeled global-average perturbation of the isotopic signature of dissolved inorganic carbon in the ocean ($\delta^{13}C_{DIC}$; thick gray line) in response to a carbon pulse to the atmosphere of 500 GtC with a $\delta^{13}C$ signature of -24 ‰. The thin colored lines show the perturbation as reconstructed with one (blue), two (green, dash), and three (red) principal components.

9. Appendix: Maybe consider replacing "->" in you subscripts by "2" or "→" (command rightarrow in LaTeX), e.g. in Eq. A17: change $\alpha_{a->b}$ into $\alpha_{a2b}$ or $\alpha_{a\to b}$.
Done.
10. Reference list needs careful revision, since sub- and superscripts seems to be partly wrong, e.g. $CO_2$ instead of $CO_2$ etc. Futhermore, AGU paper from the time without page numbers need to have their paper numbers included, e.g. necessary for Parekh et al. (2008) where the missing paper number is PA4202, and the page number 1–14 are useless. This is probably the case for all papers from GBC, GRL, P, and from online only journals such as Nature Communcations which start with page number 1. Check Eggleston et al 2016; Köhler et al 2010; Menviel et al. 2012; Ridgwell and Hargreaves 2007; Skinner et al. 2017; Tarnocai et al 2009 Check also, when two links exist and reduce to one (DOI). Lord et al 2015 is incomplete. Last page number is missing in Wanninkhof 1992.
Done.

**References**

Bauska, T. K., Baggenstos, D., Brook, E. J., Mix, A. C., Marcott, S. A., Petrenko, V. V., Schaefer, H., Severinghaus,

J. P., and Lee, J. E.: Carbon isotopes characterize rapid changes in atmospheric carbon dioxide during the last deglaciation, Proceedings of the National Academy of Sciences, 113, 3465–3470, doi: 10.1073/pnas.1513868113, 2016.

Bauska, T. K., Brook, E. J., Marcott, S. A., Baggenstos, D., Shackleton, S., Severinghaus, J. P., and Petrenko, V. V.: Controls on Millennial-Scale Atmospheric $CO_2$ Variability During the Last Glacial Period, Geophysical Research Letters, 45, 7731–7740, doi: 10.1029/2018GL077881, 2018.

Colbourn, G., Ridgwell, A., and Lenton, T. M.: The time scale of the silicate weathering negative feedback on atmospheric $CO_2$, Global Biogeochemical Cycles, 29, 583–596, doi: 10.1002/2014GB005054, 2015.

Eggleston, S., Schmitt, J., Bereiter, B., Schneider, R., and Fischer, H.: Evolution of the stable carbon isotope composition of atmospheric $CO_2$ over the last glacial cycle, Paleoceanography, 31, 434–452, doi: 10.1002/2015PA002874, 2016.

Lord, N. S., Ridgwell, A., Thorne, M. C., and Lunt, D. J.: An impulse response function for the long tail of excess atmospheric $CO_2$ in an Earth system model, Global Biogeochemical Cycles, 30, 2–17, doi: 10.1002/2014GB005074, 2016.

Parekh, P., Joos, F., and Müller, S. A.: A modeling assessment of the interplay between aeolian iron fluxes and iron-binding ligands in controlling carbon dioxide fluctuations during Antarctic warm events, Paleoceanography, 23, PA4202, doi: 10.1029/2007PA001 531, 2008.

Schneider, R., Schmitt, J., Köhler, P., Joos, F., and Fischer, H.: A reconstruction of atmospheric carbon dioxide and its stable carbon isotopic composition from the penultimate glacial maximum to the last glacial inception, Climate of the Past, 9, 2507—2523, doi: 10.5194/cp-9-2507-2013, 2013.

**Reviewer 2**

The authors explore the interactive responses of carbon and carbon isotopes in the land-ocean-atmosphere system to a suite of atmospheric perturbations on timescales spanning from decades to 100,000 years. By considering feedbacks from CaCO3 compensation and weathering as well as short term feedbacks from the land and ocean in an Earth system modeling framework, this study provides an important insight regarding changes in the global carbon cycle on a wide range of timescales. Another novel aspect of this study is that the authors developed and used analytical expressions to better understand the different responses in atmospheric CO2 and the d13C of CO2,simulated from the numerical model. This study is worth being published in Climate of the Past after clarifying the following few points:

We thank the reviewer for the support and thoughtful comments.

1. One of the main finding is that atmospheric CO2 perturbation is relatively gradually removed over a 100,000-year timescale, due to non-negligible effects of carbonate compensation and rock weathering feedbacks on atmospheric CO2. On the other hand, the atmospheric d13C-CO2 perturbation is relatively rapidly removed because the oceanic biological pump and the subsequent burial into sediments play a dominant role in removing atmospheric perturbation (as stated in Abstract). It is interesting to see that the biological pump and the burial into sediments can effectively remove the atmospheric d13C-CO2 perturbation on an e-folding timescale of 6 years (Table 2),while oceanic d13C-DIC is removed on much longer timescales (Figure 2b). What is the mechanisms by which the oceanic d13C-DIC is removed? If the burial of organic carbon from the biological pump is responsible for the long-term removal from the land-ocean-atmosphere system, I feel that the sedimentation model deserves a more detailed description than just pointing to a previous study. I also feel that the results are too briefly discussed in the manuscript. For example, how much organic carbon is buried in time with which d13C signatures?

There appears to be a misunderstanding. The biological pump and sediment burial are not responsible for the initial reduction of the atmospheric perturbations. Rather, air-sea gas exchange and physical ocean transport as well as land biosphere processes are responsible for the initial reduction of the perturbations. We added a corresponding sentence in the abstract and the text reads now (p.1, l.5):

On timescales from years to many centuries the atmospheric perturbations in $CO_2$ and $\delta^{13}CO_2$ are reduced by air-sea gas exchange and physical transport from the surface to the deep ocean and by the land biosphere. Isotopic perturbations are initially removed much faster from the atmosphere than perturbations in $CO_2$ as explained by

aquatic carbonate chemistry. On longer time scales, the $CO_2$ perturbation is removed by carbonate compensation and silicate rock weathering. In contrast, the $\delta^{13}C$ perturbation is removed by the relentless flux of organic and calcium carbonate particles buried in sediments.

The fast decrease in IRF($\delta^{13}C$) in the first decade is due to the fast redistribution of the perturbation in the atmosphere-ocean-land system. The main difference to IRF($CO_2$) is that the isotopic ratio is not affected by the chemical buffering from the acid-base reactions. The six year timescale from the IRF($\delta^{13}C$) fit is to the time scales of air-sea gas exchange and upper ocean mixing. This is discussed in the manuscript starting on p.9, l.16. We modified the text for the initial response at p.11, l.4 to read:

Further, the top 300 m with a carbon inventory of 2,600 GtC are approximately ventilated within a decade through air-sea gas exchange and upper ocean physical transport processes (circulation, mixing).

On long time scales, the isotopic perturbation in the ocean-atmosphere-land system are indeed jointly removed by the burial flux to the lithosphere. The isotopic perturbation in the ocean, the land biosphere and atmosphere are coupled by air-sea and air-land gas exchange. The burial flux therefore reduces the perturbations in all three reservoirs as evident in Fig. 2b. The sediment module of Bern3D is introduced on p.4, l3-22 in the methods section. The equations, parameters and evaluation of the module have been published earlier. For convenience and further reference, we add a table with all parameters in the appendix. We added the following text on p.4, l.4 to point the interested reader to the details of the model:

The governing equations of the sediment model are given in the appendix A of Tschumi et al. (2011). A comparison of simulated versus observation-based sediment composition for the setup used in this study is given in appendix B of Jeltsch-Thömmes et al. (2019). A table with all parameters applied here and by Jeltsch-Thömmes et al. (2019) is given in the appendix (Table B1).

The following text is added in addition on p.4, l.12 to inform the reader on simulated burial fluxes and sedimentary stocks:

The burial flux at preindustrial steady state is 0.22 GtC yr$^{-1}$ in the form of $CaCO_3$, 0.24 GtC yr$^{-1}$ in the form of POC, and 6.96 TmolSi yr$^{-1}$ in the form of opal, all within the observational range. The global inventories in the interactive sediment layers amount to 916 GtC for $CaCO_3$, to 510 GtC for POC, and to 21,460 Tmol for opal.

Further, we add the equations of the weathering feedback parameterization as documented in Colbourn et al. (2013) in the appendix.

The marine $\Delta\delta^{13}C_{DIC}$ perturbation is removed by the burial flux of organic carbon and $CaCO_3$. We now add further information about how much organic carbon and $CaCO_3$ is buried over time and with which signature in the section where the carbon and isotopic perturbation budgets are discussed. The new text on p.14, l. 7 reads:

We further illustrate the contribution of the burial and weathering fluxes to the carbon and isotope budgets for the simulation with ($WEA_{500}$) and without weathering feedbacks ($SED_{500}$) in Fig. 4. The inventory in the relatively fast exchanging pools – atmosphere, ocean, reactive sediments, land biosphere - increases by ∼350 GtC in addition to the pulse input of 500 GtC over the course of simulation $SED_{500}$. This increase is mainly due to a reduction in the burial of $CaCO_3$. In contrast, $CaCO_3$ burial is enhanced in simulation $WEA_{500}$ and ∼430 Gt of carbon are removed from the fast exchanging reservoirs. This removal is only partly compensated by enhanced weathering (∼380 GtC), leading to a reduction in the inventory of the fast exchanging pools from 500 GtC immediately after the pulse to 453 GtC at 100 kyr. Changes in POC burial fluxes are negligible for the carbon budget.

Turning to the isotopic pertubation budget, the initial perturbation due to the pulse input of -12,000 GtC ‰ is mainly removed by anomalies in the POC burial flux both in $SED_{500}$ (∼10,000 GtC ‰) and $WEA_{500}$ (∼7,000 GtC ‰). Changes in $CaCO_3$ burial and weathering mitigate about a quarter (∼2,900 GtC ‰) of the initial isotopic perturbation in the two simulations. This leaves a relatively small isotopic inventory perturbation of about +1,100 and -1,600 GtC ‰ in the fast exchanging reservoirs at 100 kyr in simulations $SED_{500}$ and $WEA_{500}$.

The contributions of the anomalies in the POC and $CaCO_3$ burial fluxes may be approximately attributed to changes in the signature ($\Delta\delta$) and to changes in flux ($\Delta F$) relative to the initial flux, $F_0$, and signature, $\delta_0$, before the perturbation:

$$\Delta(F \cdot \delta) = F_0 \cdot \Delta\delta + \Delta F \cdot \delta_0 \tag{1}$$

The isotopic perturbation of the POC burial flux is mainly mediated by a change in the signature of the organic carbon buried ($\Delta\delta^{13}C(POC)$). For example, the signature of the total POC burial flux is on average 0.29 ‰ more negative than the corresponding input from weathering in simulation $WEA_{500}$. This change in signature of the total POC burial flux leads to an effective reduction of the isotopic perturbation. The change in POC burial flux ($\Delta F(POC)$) is, as mentioned above, negligible. Both changes in the signature of the total $CaCO_3$ burial flux and changes in $CaCO_3$ burial contribute significantly to the isotopic perturbation. At the end of simulation $WEA_{500}$, the mean $\delta^{13}C$ signature of $CaCO_3$ burial is on average 0.14 ‰ more negative than the input from

weathering. In turn, the term $F_0 \cdot \Delta\delta$ for CaCO$_3$ burial contributes $\sim$ -3,100 GtC·‰ to the mitigation of the initial isotopic perturbation. This is partly counteracted by excess burial of CaCO$_3$, with $\Delta F \cdot \delta_0$ equal $\sim$1,300 GtC·‰. This yield an overall burial contribution of $\sim$ -1,800 GtC·‰. The $\delta^{13}$C perturbation is further removed by increased weathering input of CaCO$_3$ (see methods), contributing about 1,100 $GtC \cdot$ ‰ over 100 kyr in $WEA_{500}$.

In summary, the isotopic perturbation is mainly mitigated by a change in the signature of the mean POC burial flux. The smaller contributions from the CaCO$_3$ cycle to the isotopic budget is related to both a change in the signature of the mean CaCO$_3$ burial flux and to changes in burial and weathering carbon fluxes.

Further, we add an additional figure (Fig. 2 in the reply) to the revised manuscript, in conjunction with the new text above.

[Figure]

Figure 2: **New Fig. 4 for the revised manuscript:** The perturbation budget for (a) carbon and (b) carbon isotopes. Solid lines show results from experiment $WEA_{500}$, dashed lines from experiment $SED_{500}$. In both panels the pulse (500 GtC, -12,000 $GtC \cdot$ ‰) is subtracted from the combined atmosphere-ocean-land-reactive sediment inventory. Burial fluxes are plotted inversely and the sum of burial (green, orange lines) and weathering (black) fluxes yields the combined inventory (pink).

2. It is also possible that the dissolution and weathering fluxes of CaCO3 and the subsequent mixing with seawater might be responsible for the long-term removal of the d13C-DIC perturbation from the ocean. Furthermore, weathering fluxes to the ocean,with a d13C signature of -9.2 permil (as stated in the introduction, if I understood it correctly), might be important for the oceanic d13C-DIC budget, although this may not be directly important for the atmospheric d13C-CO2. How does the contribution from weathering fluxes to the d13C budget compare with the contribution from the burial fluxes to marine sediments? How does the weathering fluxes in the model change in time and why? It would help if the authors discuss it in more detail. In any case, it is interesting that these slow, yet persistent, carbon burial or weathering fluxes over 100,000 years can remove about 80% of the total d13C perturbation in 100,000 years. A related question is the carbon isotopes budgets for the simulations SED and CLO (i.e., Figure 2 for SED and CLO simulations)?
Please see our reply to the point above.

3. Another major finding from this study is that the burial flux of d13C is influenced by the spatial (vertical?) d13C

gradients. It would be convincing if the authors show the temporal changes in the burial fluxes of d13C along with an index of the spatial d13C gradient for the suite of simulations.

Please see our response to your point 2. We now show the evolution of the burial flux in a new figure. In Fig. 3 below (added to the appendix of the MS), we show the evolution of horizontally-averaged $\delta^{13}$C of DIC in a Hovmoeller-type diagram and replace '*(not shown)*' on p.12, l.14 with the reference to the new figure in the appendix.

[Figure]

Figure 3: **New Fig. A1 for the revised manuscript:** Hovmoeller-type diagram showing the temporal evolution of the global-mean vertical profile of $\Delta\delta^{13}C_{DIC}$ for experiment (a) $WEA_{500}$ and (b) $SED_{500}$.

4. The analytical expressions for the impulse responses indeed help to better understand the different behavior for CO2 and d13C-CO2, yet require some calcifications. The authors assumed 2 ocean-atmosphere boxes for the equation related to CO2 while assuming 3 ocean-atmosphere-land boxes for the equation related to d13C-CO2. Is it because for CO2 budget the land carbon storage can be neglected? I can follow the derivation except (A1). From (A1), I can infer that atmospheric carbon concentration is equal to DIC0 at time = 0. I think that a scaling factor might be missing here, as only 1 mol of CO2 is equilibrated with every 200 moles of DIC. The scaling factor might be included in the authors' definition for ha, and would be canceled out eventually, resulting in the same analytical expression as equation (5). Yet it should be clarified. The scaling factor depends on the perturbation P when P is large, hence we would end up arriving at an equation (5) that depends on P when P is large, which might be a reason that we see a different response for the p5000 simulation in Figure 1a (?).

We add the following sentences from the appendix to the main text to justify our choice of a 2-box model for the $CO_2$ and a 3-box model for the $\delta^{13}C$ perturbation. The text on p.10, l.6 now reads:

Changes in land biosphere carbon storage can be considerable, in particular during the first century (see Fig. 3a). However, the role of the land carbon stock changes becomes smaller, the more carbon is taken up by the ocean and is thus for simplicity not considered in the analytical expression for $CO_2$. For $CO_2$ the expression reads: [...]

In case of the isotopic perturbation, we add on p.10, l.15:

For $\delta^{13}C_a$, we also consider the initial carbon inventory of the land biosphere ($N_{b,0}$), as a substantial amount of the isotopic perturbation is contained in the land biosphere on multi-centennial to millennial timescales (Fig. 3b). The corresponding expression reads (for the derivation see the appendix):

There is no scaling factor missing in the equations and equation A7 does not explicitly depend on the size of the pulse. The concentration $C$ is specifically defined for our box model and should not be confused with the concentration or mixing ratio of $CO_2$ in the real-world atmosphere. We note now that the Revelle factor depends on the pulse size. We have clarified the text starting on p22, l5 to read:

We consider a two box model of the atmosphere (index: $a$) and ocean (index: $o$) with area A=$A_a$=$A_o$ and heights $h_a$ and $h_o$. All boxes are assumed to be well mixed. Before the pulse addition, the model is in equilibrium and the initial $CO_2$ partial pressure is $pCO_{2,0}$=$pCO_{2,a,0}$=$pCO_{2,o,0}$. The initial concentration of dissolved inorganic carbon (DIC) in the ocean is $DIC_0$ and the carbon inventory in the ocean is given by $N_o$=$DIC \cdot h_o \cdot A$. We relate the atmospheric carbon inventory, $N_a$, to a time-varying atmospheric concentration $C_a$ and to the fixed volume of the model atmosphere: $N_a$=$C_a \cdot h_a \cdot A$. We are free to select the units of the model-specific concentration $C_a$. Here, we express $C_a$ in units of $DIC_0$ and set the initial atmospheric concentration $C_{a,0}$ equal to $DIC_0$. The atmospheric inventory is proportional to the atmospheric partial pressure, $pCO_{2,a}$ and, correspondingly, $C_a$ has to be proportional to $pCO_{2,a}$. This leads to the following definition of $C_a$:

$$C_a = \frac{pCO_{2,a}}{pCO_{2,0}} DIC_0, \qquad (2)$$

$C_a$ is specifically defined for our box model and should not be confused with the concentration or mixing ratio of $CO_2$ in the real-world atmosphere. The atmospheric scale height $h_a$ is given by the requirement that volume times concentration equals inventory:

$$h_a = \frac{N_{a,0}}{A \cdot DIC_0}. \qquad (3)$$

The perturbation ($\Delta$) from the initial equilibrium in $pCO_2$, and similarly for other quantities, is defined by: $\Delta pCO_2(t)$= $pCO_2(t)$-$pCO_2(t_0)$. The perturbation in $pCO_{2,o}$ is related to that in DIC by the Revelle factor $\xi$ (Revelle and Suess, 1957):

$$\xi = \frac{\Delta pCO_{2,o}}{pCO_{2,0}} \cdot \frac{DIC_0}{\Delta DIC} \qquad (4)$$

We note that the carbonate chemistry is non-linear and $\xi$ varies with environmental conditions (DIC, alkalinity, temperature, salinity). Assuming a constant Revelle factor $\xi$ is therefore an approximation. In particular, $\xi$ increases with increasing DIC. As a consequence, the "effective" value of $\xi$ increases with increasing carbon emission and is thus the larger the larger the magnitude of the carbon pulses. [This yields for the perturbation in the net air-to-sea carbon flux...]

Equation 5 implicitly includes a dependence on the pulse size in form of the buffer factor $\xi$. The larger the pulse size the larger also the buffer factor and thus also the remaining atmospheric fraction. We state this in the appendix on p.27, l.17-18 and now also include it in the main text on p.11, l.6:

It has to be noted here that we assume $\xi$ constant for small perturbations. With increasing pulse sizes, also $\xi$ increases, leaving larger airborne fractions (c.f. $WEA_{500}$ versus $WEA_{5000}$ in Fig. 1a after 1-2 kyr). Regarding $\delta^{13}$C, the corresponding [...]

**Short Comment by James Menking**

Hi Aurich and Fortunat, Great paper! In my opinion, this is already well-written and clearly presented. The work is relevant to anyone interested in the carbon cycle, especially to those of us interested in d13C.
Thank you for your support and interesting questions that helped us to improve the presentation of results.

I'm curious about two things:

(1) One conclusion of the paper is that you cannot get a 0.2 ‰ depletion in atmospheric d13C-CO2 and a 10 ppm rise in CO2 concentration from a terrestrial pulse, as is suggested by Bauska 2018 as a possible cause of the variations observed at the onset of HS4. Your conclusion is based on your Figure 6, which shows that terrestrial pulses of carbon reduce d13C-CO2 only ~0.1 ‰ per 10 ppm increase in CO2. But you state in the caption that the results in Figure 6 are the means of the model output for the 30 years immediately following the perturbations. So the atmospheric data during the perturbations (i.e. while CO2 is increasing and d13C-CO2 decreasing) are not considered? In my own modeling experiments with the OSU 14-box model (model described in Bauska 2016), fitting d13C-CO2 v. CO2 output during a land carbon pulse, not just the recovery after it, significantly decreases the slope in the Keeling plot and makes a -0.2 ‰ change in d13C-CO2 per 10 ppm increase in CO2 due to land carbon seem more feasible. Do you think that the perturbations themselves are too fast to be recorded in ice cores? Surely this is true for extremely fast pulses, but the results in Figure 6 include perturbations up to 400 years duration, which I think should be captured by ice cores even with relatively large gas age distributions. Put another way, I observe that land carbon pulses in our box model plot as quasi-ellipses in a Keeling plot with the slope of the d13C-CO2 v. CO2 being more negative on the way "up" versus on the way "down," even for multi-centennial carbon releases.

So I guess the question is - to what extent is the "up" variability recorded in ice cores? - because it may argue for the Bauska 2018 interpretation. The conclusion in your paper seems particularly strong without elaborating more on this point.

Thank you for raising this issue. Averaging over different intervals and also including the 'up' variability smears out the rather straight line seen in Fig. 6 to something more like a cone, opening at higher $\Delta CO_2$ values, and also moving $\Delta\delta^{13}C$-$\Delta CO_2$ pairs towards the ice core value. We provide now more background information, add a second panel to Fig. 6 of the manuscript (Fig. 4 below, Fig. 8 in the revised manuscript) and change the corresponding paragraphs starting on p.20, l.29 to now read:

[Figure]

Figure 4: **New Fig. 8 for the revised manuscript:** (a) Temporal evolution of the ratio of $\Delta\delta^{13}C$ to $\Delta CO_2$ in the atmosphere. Results are from simulations $WEA_{100}$ forced by preindustrial boundary conditions (solid lines) and $LGM_{100}$ forced by Last Glacial Maximum boundary conditions (thin dashed lines). Carbon is released as a single pulse at year 0 (green lines) or assumed to be released at a uniform rate from year 0 to the year plotted (purple lines; see main text). The dashed horizontal line indicates a $\Delta\delta^{13}C$ to $\Delta CO_2$ ratio of -0.2 ‰ to 10 ppm as seen in ice core records for HS1 and HS4. (b) Response in $\Delta CO_2$ and $\Delta\delta^{13}C_{atm}$ to a uniform release of 20, 40, 80, 120, and 160 GtC (colors) of isotopically light carbon ($\delta^{13}C$ = -24 ‰) to the atmosphere over 100, 200, 300, and 400 years (numbers). Shown is the mean over 30 years starting in the last year of the release. Red cross indicates the approx. $\Delta\delta^{13}C$ to $\Delta CO_2$ change as seen in ice core records for HS1 and HS4 (Bauska et al., 2016, 2018).

Further, the different behavior of $CO_2$ and $\delta^{13}C_{atm}$ perturbations has also consequences for centennial scale $CO_2$ and $\delta^{13}C_{atm}$ variations such as during Heinrich Stadial (HS) 4 and 1. Variations in $CO_2$ and $\delta^{13}C_{atm}$ during HS4 and HS1 have been measured on Antarctic ice cores. The data show an increase in $CO_2$ on the order of ~10 ppm over 200-300 years, accompanied by a decrease in $\delta^{13}C_{atm}$ of ~0.2 ‰. This results in a value of roughly -0.02 ‰ per ppm for the ratio of the change in $\delta^{13}C$ to the change in $CO_2$ ($r = \Delta\delta^{13}C / \Delta CO_2$). A release of terrestrial carbon has been discussed as a possible cause of these events (Bauska et al., 2016, 2018).

We utilize our pulse response simulations under PI and LGM boundary conditions to estimate the changes in $r$ in response to a transient terrestrial carbon input with an isotopic signature of -24 ‰. The results for $r$ will depend on the evolution of the emissions into the atmosphere. For a pulse-like input at time zero the evolution of $r$ after the pulse is shown in Fig. 8a for the first 2,000 years. $r$ is about -0.08 (-0.06) ‰ ppm$^{-1}$ immediately after the release under LGM (PI) boundary conditions. The difference between LGM and PI values of $r$ comes from the difference in the atmospheric C and $^{13}$C inventory to which the pulse is added. The magnitude of $r$ is reduced rapidly within the first 25 years and remains approximately constant at -0.015 (-0.01) ‰ ppm$^{-1}$ thereafter. Thus, $r$ recorded in ice after a hypothetical pulse release of carbon would be above -0.02 ‰ ppm$^{-1}$ if the age distribution of the ice core samples is wider than a few decades. However, emissions may occur more gradually. For simplicity, we assume an emission pulse sustained over period $T$ according to a two-sided Heaviside function; terrestrial emissions, $e(t)$, suddenly increase to unity, remain elevated for period $T$, and are then immediately reduced again to zero. $r$ is then readily calculated from the IRFs ($r = \int_0^T dt \cdot e(t) \cdot IRF(\delta^{13}C)(T-t) / \int_0^T dt \cdot e(t) \cdot IRF(CO_2)(T-t)$). The results for $r$ are again plotted in Fig. 8a. $r$ is decreasing with the duration $T$ of sustained emissions and falls below -0.02 ‰ ppm$^{-1}$ for $T$ larger than about 90 (30) years for LGM (PI) boundary conditions. Finally, we also performed simulations with the Bern3D-LPX under PI boundary conditions to account for non-linearity that may not be captured in the IRF representation. The releases of varying amounts of terrestrial carbon with $\delta^{13}C$ = -24

‰ over 100 to 400 years yield a linear response in $\Delta CO_2$ and $\Delta\delta^{13}C_{atm}$ (Fig. 8b) with a slope $r$ of about -0.01 ‰ ppm$^{-1}$.

There are uncertainties in these estimates of $r$. We assumed that the emitted carbon was about -18 ‰ depleted with respect to the atmosphere. The relative uncertainty in this value translates directly to the relative uncertainty in $r$. The release of material that is isotopically even more depleted would bring our model results in better agreement with the ice core estimate of $r$. Observations and land model simulations suggest that the $\delta^{13}C$ depletion in vegetation and soils may vary between -24 ‰, for heavily discriminated C3 plant material, to -5 indicative of C4 plant material Keller et al. (2017). C4 plants were likely more abundant during glacials than interglacials (Collatz et al., 1998). Also, the values shown in Fig. 8b are 30-year means. A wider gas age distribution would reduce the perturbation in $\delta^{13}C_{atm}$ more than in $CO_2$ (see e.g. Köhler et al., 2011). On the other hand, the $CO_2$ increase may have occurred in less than 100 years (Bauska et al., 2018) and correspondingly a more negative value of $r$ is expected for such a fast compared to a centennial scale increase. Given these uncertainties in our model estimates as well as in the determination of $r$ from the ice core measurements, we are not in a position to firmly conclude whether a terrestrial carbon release was the sole mechanism for the HS1 and HS4 $CO_2$ events or not. Other mechanisms in addition to a terrestrial release may have contributed to the $CO_2$ and $\delta^{13}C_{atm}$ excursions during HS4 and HS1.

We acknowledge that most of the idealized simulations presented in this study were run under preindustrial boundary conditions. A limited set of sensitivity simulations suggest that the IRFs for $CO_2$ and $\delta^{13}C$ are similar under preindustrial and glacial $CO_2$ in our model. The direct influence of lower atmospheric $CO_2$ on IRF($\delta^{13}C$) appears limited. Further work is needed to better quantify the dependency of IRFs on the climate state.

We also adjusted the wording in the abstract to better reflect these uncertainties:
Our results may suggest that changes in terrestrial carbon storage were not the sole cause for the abrupt, centennial $CO_2$ and $\delta^{13}C$ variations recorded in ice during Heinrich Stadials HS1 and HS4 of the last glacial period. However, model and data uncertainties prevent a firm conclusion.

(2) Another conclusion in the paper is that the PGM-LGM d13C-CO2 difference observed in the Schneider/ Schmitt/ Eggleston work cannot be due to changes in terrestrial carbon storage, or internal reorganizations in the marine carbon cycle without considering burial. But I had a difficult time understanding how the results of your experiments argue for these points. Regarding terrestrial carbon storage, is your point that the PGM-LGM land carbon difference would have been too large because the atmospheric imprint of any smaller land carbon transfer would have been attenuated too much to cause the observed PGM-LGM d13C-CO2 difference? If so, the connection between the model results and the stated conclusion could be clearer. I thought it was also difficult to follow your point about the marine carbon cycle. Perhaps showing the marine and atmospheric d13C data for the PGM v. LGM would help, and stating more clearly exactly how your experimental results support the conclusions.

In light of your comments we modified the text for clarification. The paragraph (p.20, l. 9) reads now:
Our results have consequences for the interpretation of the difference in $\delta^{13}C$ between similar climate states such as the Penultimate Glacial Maximum (PGM) and the Last Glacial Maximum (LGM). Substantial temporal $\delta^{13}C$ differences between these periods are recorded in ice cores (Schneider et al., 2013; Eggleston et al., 2016) and in marine sediments (Hoogakker et al., 2006; Oliver et al., 2010). Different mechanisms, such as changes in the $\delta^{13}C$ signature of weathering and burial fluxes, varying contribution of volcanic outgassing of $CO_2$, and changes in the amount of carbon stored in the land biosphere, especially in yedoma and permafrost soils, have been discussed for the PGM-LGM $\delta^{13}C$ offset (e.g. Lourantou et al., 2010; Schneider et al., 2013). An internal reorganization of the marine carbon cycle without considering changes in burial may not explain the offset. With no changes in the weathering-burial balance, the mass in the atmosphere-ocean-land system remains constant. Then, a change in atmospheric $\delta^{13}C$ would require an opposing $\delta^{13}C$ change in the ocean. This appears in conflict with marine and ice core records which suggest that $\delta^{13}C$ increased both in the ocean and in atmosphere by about 0.3 ‰ between the PGM and LGM (Hoogakker et al., 2006; Oliver et al., 2010; Schneider et al., 2013; Eggleston et al., 2016). However, internal marine carbon cycle reorganizations, e.g., due to changes in circulation, may have altered $\delta^{13}C$ in the surface ocean, and in turn $\delta^{13}C$ of the burial flux, and thereby the balance between weathering input and burial.

The Bern3D results suggest that changes in organic carbon storage on land (or in the ocean) are not a likely explanation of the isotopic offset. The $\delta^{13}C$ signal of a transfer of organic carbon of plausible magnitude to the atmosphere and ocean would have been attenuated too much over time to cause the observed PGM-LGM $\delta^{13}C$ difference. Schneider et al. (2013) estimated required changes in land carbon storage to match the ice core and

marine temporal offsets in $\delta^{13}$C using preliminary Bern3D results for pulse release simulations. Our results show, in agreement with these earlier results, that the isotopic perturbation associated with a terrestrial carbon release is attenuated to less than 15 % within two millennia for pulse sizes of up to 5,000 GtC and declines thereafter (see black line in Fig. 5b). This would require several thousand GtC to have been stored additionally in the land biosphere during the LGM compared to the PGM in order to explain the $\delta^{13}$C offset (see also Schneider et al., 2013). This amount seems large in light of estimated total carbon stocks of ~1500 GtC in perennially frozen soils in Northern Hemisphere permafrost regions today (Tarnocai et al., 2009) and an even smaller inventory at the LGM (Lindgren et al., 2018).

Taken together, these ad-hoc considerations suggest that long-term imbalances in the weathering (including volcanic emissions) and burial fluxes appear to be a plausible cause for the $\delta^{13}$C differences between PGM and LGM. However, further work, which considers the transient evolution of $CO_2$, $\delta^{13}$C, and other proxies, is required to gain further insight into the contributions of individual mechanisms to long term $\delta^{13}$C changes recorded in ice and marine cores.

I'm very keen to hear your thoughts. Thanks, and good luck with the rest of the submission! - Andy Menking

[revised manuscript text omitted]

$$\frac{dN_o^i}{dt} = F^i_{a->o\,a\to o} - F^i_{o->a\,o\to a} = F^i_{a->o,net\,a\to o,net}$$

$$\frac{dN_b^i}{dt} = F^i_{a->b\,a\to b} - F^i_{b->a\,b\to a} = F^i_{a->b,net\,a\to b,net}$$

$$\frac{dN_a^i}{dt} = -F^i_{a->o,net\,a\to o,net} - F^i_{a->b,net\,a\to b,net} + \delta(t) \cdot P^i \tag{A11}$$

$N$ is the reservoir size (e.g. in GtC), indices $a$, $o$, and $b$ denote the atmosphere, ocean, and land biosphere box, respectively. Index $i$ refers either to carbon or $^{13}$C. $F$ denotes fluxes between reservoirs. $\delta$ is the Kronecker symbol and $P^i$ the pulse released at time $t = t_0 = 0$ to the atmosphere.

*The initial atmospheric $\delta^{13}$C perturbation:* The IRF($t$) for $\delta^{13}$C$_a$ is given by the perturbation in the isotopic signature, $\Delta\delta^{13}$C$_a$(t), divided by the initial ($ini$) perturbation, $\Delta\delta^{13}$C$_a^{ini}$. $\Delta\delta^{13}$C$_a^{ini}$ is the perturbation immediately after a carbon input of amount $P$ and with signature $^{13}$R$_P$ ($\delta^{13}$C$_P$). Mass balance implies that the amount of $^{13}$C before and after the pulse release is equal:

$$N_{a,0} \cdot {}^{13}R_{a,0} + P \cdot {}^{13}R_P = (N_{a,0} + P) \cdot {}^{13}R_a^{ini}. \tag{A12}$$

We convert the ratios to $\delta$-units and subtract on both sides $(N_{a,0} + P) \cdot \delta^{13}$C$_{a,0}$ to get the initial perturbation:

$$\Delta\delta^{13}\text{C}_a^{ini} = \frac{P}{N_{a,0} + P} \cdot (\delta^{13}\text{C}_P - \delta^{13}\text{C}_{a,0}). \tag{A13}$$

$\Delta\delta^{13}C_a^{ini}$ is proportional to the difference between the signature of the pulse and the initial atmospheric signature. Thus, the initial perturbation is relative to the atmospheric $\delta^{13}C$ signature.

*The relationship between $\delta^{13}C_a$, $\delta^{13}C_o$, and $\delta^{13}C_b$ at equilibrium:* In this subsection, relationships between the isotopic signatures are developed by considering equilibrium between the three boxes. This will allow us to simplify the isotopic mass balance equations further below .

The gross air-to-sea flux per unit area is:

$$^{13}f_{a->o\underset{\sim}{a\to o}} = {^{13}k_g} \cdot {^{13}pCO_{2,a}} = k_g \cdot pCO_{2,a} \cdot {^{13}\alpha_{a->o\underset{\sim}{a\to o}}} \cdot {^{13}R(pCO_{2,a})}, \tag{A14}$$

where $^{13}\alpha_{\overline{a->s}} {^{13}\alpha_{\underset{\sim}{a\to o}}} = {^{13}k_g}/k_g$ is the discrimination factor for the gross air-to-sea transfer. The gross sea-to-air flux is:

$$
\begin{aligned}
^{13}f_{o->a\underset{\sim}{o\to a}} = {^{13}k_g} \cdot {^{13}pCO_{2,o}} &= k_g \cdot pCO_{2,o} \left( \frac{^{13}k_g}{k_g} \frac{^{13}R(pCO_{2,o})}{^{13}R(DIC)} \right) \cdot {^{13}R(DIC)} \\
&= k_g \cdot pCO_{2,o} \cdot {^{13}\alpha_{o->a\underset{\sim}{o\to a}}} \cdot {^{13}R(DIC)}.
\end{aligned}
\tag{A15}
$$

The term in parentheses is the the discrimination factor for the transfer of carbon from the DIC pool to the atmosphere, $^{13}\alpha_{\overline{o->a}}{^{13}\alpha_{\underset{\sim}{o\to a}}}$. At equilibrium (*eq*), the two gross fluxes cancel each other. It follows with $pCO_{2,a,eq} = pCO_{2,o,eq}$ and $^{13}pCO_{2,a,eq} = {^{13}pCO_{2,o,eq}}$:

$$^{13}R_{eq}(pCO_{2,a}) = \frac{^{13}\alpha_{o->a}\,{^{13}\alpha_{o\to a}}}{^{13}\alpha_{a->o}\,{^{13}\alpha_{\underset{\sim}{a\to o}}}} \cdot {^{13}R_{eq}(DIC)} = \frac{1}{\alpha_{a,o}} \cdot {^{13}R_{eq}(DIC)}. \tag{A16}$$

This equation gives the equilibrium relationship between the atmospheric and oceanic signature. $\alpha_{a,o}$ denotes the equilibrium discrimination factor between $pCO_{2,a}$ and DIC. $\alpha_{a,o}$ depends on temperature and somewhat on carbonate chemistry (Mook, 1986) and is about 1.008 and thus close to 1.

A similar relationship is readily developed for the isotopic ratio of the total carbon in the land biosphere. The gross fluxes between the land and atmosphere are:

$$
\begin{aligned}
^{13}F_{a->b\underset{\sim}{a\to b}} &= F_{a->b\underset{\sim}{a\to b}} \cdot {^{13}\alpha_{a->b\underset{\sim}{a\to b}}} \cdot {^{13}R(pCO_{2,a})} \\
^{13}F_{b->a\underset{\sim}{b\to a}} &= F_{b->a\underset{\sim}{b\to a}} \cdot {^{13}\alpha_{b->a\underset{\sim}{
[revised manuscript text omitted]